



# Scanning Polarization Lidar LOSA-M3: Opportunity for Research of Crystalline Particle Orientation in the Clouds of Upper Layers

Grigorii P. Kokhanenko, Yurii S. Balin, Marina G. Klemasheva, Sergei V. Nasonov, Mikhail M. Novoselov, Iogannes E. Penner, Svetlana V. Samoilova

5  V.E.Zuev Institute of Atmospheric Optics, SB RAS, Tomsk, Russia

*Correspondence to*: Grigorii Kokhanenko (kokh@iao.ru)

**Abstract.** The article describes a scanning polarization lidar LOSA-M3, developed at the Institute of Atmospheric Optics, the Siberian Branch of Russian Academy of Sciences (IAO SB RAS). The first results of studying the crystalline particles orientation by means of this lidar are presented herein. The main features of LOSA-M3 lidar are the following: 1) an automatic scanning device, which allows to change the sounding direction in the upper hemisphere at the speed up to 1.5 degrees per second with the accuracy of angle measurement setting at least 1 arc minute; 2) separation of polarization components of the received radiation is carried out directly behind the receiving telescope, without installing the elements distorting polarization, such as dichroic mirrors and beamsplitters; and 3) continuous alternation of the initial polarization state (linear - circular) from pulse to pulse that makes it possible to evaluate some elements of the scattering matrix.

Several series of measurements of the ice cloud structure of the upper layers in the zenith scan mode were carried out in Tomsk in April-October 2018. The results show that the degree of horizontal orientation of particles can vary significantly in different parts of the cloud. The dependence of signal intensity on the tilt angle reflects the distribution of particle deflection relative to the horizontal plane, and is well described by the exponential dependence. The values of cross-polarized component in most cases show a weak decline of intensity with the angle. However, these variations are smaller than the measurement errors. We can conclude that it is practically independent of the tilt angle. In most cases the scattering intensity at the wavelength of 532 nm has a wider distribution than at 1064 nm.

## 1 Introduction

Cirrus clouds cover a significant part of the earth's surface. Thus, they have a significant impact on the radiation balance and climate, primarily due to the effects of radiation attenuation and reflection (Liou, 1986; Sassen et al., 1989). In many cases, crystalline particles of cirrus clouds have a pronounced orientation in space. This leads to optical anisotropy, manifested in various forms of the solar halo. Anisotropy affects the passage and reflection of radiation from clouds and, for example, leads to dependence of reflectivity on the zenith angle of the sun (Lavigne et al., 2008; Klotzsche and Macke, 2006).

The most well-known phenomenon is a predominant orientation of crystals in the horizontal plane. It can be caused by aerodynamic forces arising from the free fall of particles (Kaul and Samokhvalov, 2005). Sections with the horizontal





orientation of particles manifest in occurrence of sun glare when observing cloud cover from space (Chepfer et al., 1999;

Masuda and Ishimoto, 2004). Analysis of the glare width shows the correspondence to the Gaussian distribution of crystal

inclination with the half-width about 0.4 degrees (Lavigne et al., 2008; Breon and Dubrulle, 2004). Presence of horizontally

oriented crystals is also detected when observing specular spots from the spotlight on the clouds (Borovoi et al., 2008);

flutter width is also estimated at 0.4°. However, lidar observations of the ice clouds provide the basic information about

particle orientation (Platt et al., 1978; Noel and Sassen, 2005; Chen et al., 2002).

Unlike water clouds, ice clouds cause greater depolarization of backscattered radiation (Sassen and Benson, 2001). Most

often, the depolarization ratio is within $\delta = 0.3$–$0.6$ in the cloud areas with randomly oriented particles (Noel et al., 2002;

You et al., 2006). The magnitude of depolarization is undoubtedly related to the shape of particles and the phase composition

of cloud, which are taken into account when analyzing the observations of cirrus clouds (Noel et al., 2002; Hoareau et al.,

2013; Stillwell et al., 2018; Haarig et al., 2016; Campbell et al., 2015).

Starting from the works of Platt et al., 1978 and Sassen, 1977, numerous observations show that with a vertical

orientation of the lidar, the horizontally oriented particles cause a specular reflection, manifested in the absence of

depolarization and increased backscattering (Sassen and Benson, 2001; Sassen, 1991; Platt, 1978; Thomas et al., 1990). Data

analysis of the polarization lidar in the CALIPSO experiment (Cho et al., 2008; Noel and Chepfer, 2010; Yoshida et al.,

2010) shows that a significant fraction of horizontally oriented particles in middle latitudes is observed in the temperature

range from -35°C to -5°C. Deviation of the lidar from the vertical position eliminates the effect of mirror reflection. For

example, the CALIPSO lidar is 3° deviated from the nadir to eliminate this effect (Hunt et al., 2009). The angular

dependence of depolarization may vary for clouds with different temperatures (Noel and Sassen, 2005; Sassen and Benson,

2001). According to data from these works, the dependence of signal amplitude on the lidar tilt angle corresponds to the

Gaussian distribution.

Another effect caused by the horizontally oriented columns is the corner reflection when the lidar is tilted 30° from the

zenith (Konoshonkin et al., 2016). Experiments with the tilt angles of 30-43° were carried out (Hunt et al., 2009; Del Guasta

et al., 2006; Hayman et al., 2012; Veselovskii et al., 2017; Neely et l., 2013) and showed a high probability of such effect.

Particle orientations are promoted not only by aerodynamic forces, but also by forces of a different nature. A number of

works on polarization sounding of cirrus clouds (Chepfer et al., 1999; Noel and Sassen, 2005; Hayman et al., 2012; Kau et

al., 2004; Balin et al., 2011; Borovoi et al., 2014; Hayman et al., 2014) showed that crystalline particles often demonstrate

not only a preferential orientation relative to the horizon (zenith orientation) but also a preferential orientation in the

horizontal plane (azimuthal orientation). The probable reasons for this orientation are wind shifts and electric fields. It is

obvious that the direction of preferential orientation is connected with the direction of action of these forces (Kaul, 2000).

The basis for such conclusion is the observed non-invariance of the light backscattering phase matrix (BSPM) with respect to

rotation of the coordinate system (or the cloud as a whole). For example, according to a large array of experimentally

measured BSPM (Kaul et al., 2004), zero value of the element $m_{14}$ is the most probable, $m_{14} = 0 \pm 0.05$, whereas



$m_{12} = -0.22 \pm 0.2$. It means that in 30% of cases the measured depolarization and amplitude of the signal depend on the orientation of the lidar reference plane relative to the direction of the preferential orientation of particles. In other words,
when a linear polarized lidar rotates around the vertical axis, characteristics of the backscattered signal (amplitude, depolarization) should change cyclically. However, authors are unaware of such direct measurements

It should be mentioned that in most works the polarization characteristics of signal are measured at the wavelength of 532 nm. Due to technical difficulties, only a small number of works use the first harmonic of the Nd: YAG laser (1064 nm) that is optimal for recording aerosol layers (Haarig et al., 2016; Veselovskii et al., 2017; McGill et al., 2002; Burton et al., 2015;
Haarig et al., 2017). The assumption of independence of crystalline particles scattering of the wavelength is not always justified (Borovoi et al., 2014; Vaughan et al., 2010; Tao et al., 2008). Therefore, comparison of polarization and amplitude (color ratio) of signals at two wavelengths can provide the additional information about properties of crystalline particles.

The article describes a scanning polarization lidar LOSA-M3, developed at the IAO SB RAS. The first results of studying the crystalline particles orientation by means of this lidar are given herein. The main purpose of this lidar is to study
the optical characteristics of crystal clouds of the upper and middle layers at two wavelengths - 532 and 1064 nm. The design, optical scheme and principle of operation of the scanning polarization lidar are described in Sect. 2. Methods of instrument setup and calibration of the polarization channels are described in Sect. 3. Examples of observations of horizontally oriented particles in the cirrus clouds are given in Sect. 4.

## 2 Lidar description

Scanning polarizing lidar LOSA-M3 is a continuation of a series of small-size lidars LOSA-MS and LOSA-M2, developed and operated in the IAO SB RAS (Bairashin et al., 2005, 2011) in the Laboratory of Optical Sensing of the Atmosphere (LOSA). All of these lidars are intended primarily for use in field conditions, which impose certain restrictions on the weight-dimensional characteristics and the energy potential of the device. The main features of LOSA-M3 lidar are the following: 1) automatic scanning device, which allows to change the direction of sounding within the upper hemisphere at
the speed of up to 1.5 degrees per second with the accuracy of angle measurement setting at least 1 arc minute; 2) separation of polarization components of the received radiation - $I_{\parallel}$, which coincides with the original radiation, and the component

$I_{\perp}$ orthogonal to it, is carried out directly behind the receiving telescope, without installing the elements distorting polarization, such as dichroic mirrors and beamsplitters; and 3) continuous alternation of the initial polarization state (linear - circular) from pulse to pulse that makes it possible to evaluate some elements of BSPM. The lidar appearance is shown in
Fig. 1.

The optical scheme of the lidar is shown in Fig. 2. The Q-Smart 850 (Quantel) laser with a fundamental harmonic energy of 850 mJ is used with the repetition frequency of 10 Hz. Radiation is linearly polarized, but the polarization plane of the second harmonic (532 nm) is rotated by 45° relative to the first. For coincidence of the polarization planes, the phase plate



*PP1* with the phase shift of 20 wavelengths is used for λ = 532 nm, and 9.5 wavelengths for λ = 1064 nm. Turning the plate
helps achieve the coincidence of the polarization planes for two harmonics. The Glan prism *GP* improves the polarization
contrast of radiation. The quarter wave λ/4 phase plate *PP2* serves to transform the polarization state (linear-circular).

Rotation of the phase plates (the analogous plate *PP3* is placed in front of the analyzer) is performed by means of the
rotating platform 8RU-M (Standa) in synchronism with the laser pulses frequency. Thus, the phase plate can rotate by 45°
between pulses; at the same time, the state of polarization will consistently change from linear to circular and vice versa.
Immediately, we note that the lidar signals are separately recorded for each position of the plates *PP2* and *PP3* and can be
summed up (accumulated) in further processing for a certain period of time. Usually it takes from ten seconds to minute.
Thus, a synchronous rotation of two plates – *PP2* in the transmission channel and *PP3* in the receiver - allows measuring the
polarization characteristics (e.g. depolarization ratio $\delta = I_\perp / I_\parallel$ ) simultaneously for both linear and circular polarizations.
This makes it possible to exclude the variability of elements of the scattering matrix during the observation time.

The beam is collimated by the 7-fold achromatic expander *BE*, designed at the IAO SB RAS (Kochanenko et al., 2012).
Two receivers are used – an achromatic lens *AL* with the 40 mm diameter and 200 mm focus for the near zone, and
Cassegrain mirror lens *CL* with the 200 mm diameter and 1000 mm focus for the far zone. The Iris diaphragms *FS1*, *FS2* in
the focal plane of each lens determine the telescope field of view. A special feature of the Cassegrain lens design is
installation of a video camera *VC* behind the secondary mirror, which is getting radiation through an annular diaphragm in
the outer area of the secondary mirror (Simonova et al., 2015). The camera has a global shutter and is synchronized along
with laser pulses. This camera setup allows observing the image of the laser spot on the objects around without parallax. It is
especially important for setting the lidar field of view and for excluding the possibility of lidar orientation towards residential
buildings.

The signal from the receiver of the near zone through the optic fiber *Fb* is fed to the mirror shutter *MS*, by means of
which the signals of the near and far zones are alternately switched. The lens *L* forms a quasi-parallel beam that enters
through the phase plate *PP3* (similar to the plate *PP2*) to the Wollaston prism *WP*. The prism divides the beam into two
components with orthogonal polarization, which are further divided along the wavelengths by dichroic beamsplitters. Unlike
the schemes in which the wavelength division is carried out before separation of polarization components, there is no
distortion of the polarization state when reflected from dichroic elements and there is no need to apply laborious calculations
of the instrument vector and correction of the measured polarization (Di et al., 2016).

A beamsplitter *BS1* (Di-757, Semrock) is placed in the cross-polarization channel, transmitting radiation of 1064 nm and
reflecting 532 nm. In the channel, corresponding to initial polarization, beamsplitters *BS2* (transmitting 1064 nm) and *BS3*
(reflecting 532 nm and transmitting 607 nm) are installed. The radiation is detected by photodetectors in the analog mode:
the avalanche photodiodes *APD1*, *APD2* for 1064 nm (C30956EH-TC Perkin Elmer, 3 mm diameter of the receiving area),
photoelectric multipliers *PM1*, *PM2* (H11506 Hamamatsu) for 532 nm. Weak signals of Raman scattering at 607 nm are
recorded with the photomultiplier *PM3* (H11706P Hamamatsu) in the photon counting mode. Part of 532-nm radiation is



removed by the glass plate *BS4* on the photomultiplier *PM4* (H11706P), operating in the photon counting mode, which allows comparing the signals at 607 and 532 nm within one dynamic range.

The signals from photodetectors are processed by 12-bit ADCs LA-n12USB (Rudnev-Shilyaev) in case of analog signals
or by a 200 MHz photon counter (IOA SB RAS) for Raman signals and input to the computer. Peculiarity of lidar is alternation of the transceiver parameters from pulse to pulse: changing of signals from the near and far zones, the angle of the phase plates' rotation, and the rotation angle of the lidar during scanning. This eliminates a possibility of accumulating signals directly after digitizing. In our case, each signal is assigned a digital code, corresponding to the position of lidar elements, and the signals are sorted during computer processing.

## 135 3 Tuning of lidar optical elements

### 3.1 Coinciding the polarization planes

One of the main sources of errors in polarization measurements is a discrepancy between the polarization planes of the emitting and receiving channels. The Glan prism (*GP* in Fig. 2) is installed in a rotating frame and allows aligning the planes of emitter and receiver. Accuracy of installation angle is $34' = 0.0125$ rad. Figure 3 shows the measured value of the cross-
polarized component (normalized to the minimum = 1), depending on the angle of the prism rotation. A signal from a uniform aerosol layer with a constant value of $\delta \sim 1\%$ was recorded with averaging in over 4.2-5 km range and 3 minutes interval.

Let us have the signal components $P_\parallel$ and $P_\perp$, and a true depolarization ratio $\delta = P_\perp / P_\parallel$. Signal at the cross-polarization receiver, when the prism position is inaccurate, will be $P'_\perp = P_\perp \cos^2 \alpha + P_\parallel \sin^2 \alpha \approx P_\perp + P_\parallel \alpha^2$ ($\alpha$ is a setting angle error).
Hence, the measured depolarization ratio will be $\delta' = \dfrac{P'_\perp}{P_\parallel} = \delta + \alpha^2$, and the error of measured depolarization ratio is $\Delta\delta = \alpha^2 = 0.016\%$.

### 3.2 Phase plates setup

Phase quarter-wave plates are mounted on a rotating platform driven by a stepper motor, working 1200 steps per revolution. One step of the platform is 0.3° and takes 0.68 ms of time, which allows the plate to rotate 45 degrees in the time period
between two laser pulses (100 ms). During one revolution of the platform 8 laser shots are made. Positions of the plates A (transmitter) and B (receiver) for four consecutive pulses are shown in Fig. 4a. The bold line shows the direction of a fast axis of the plates. With such an arrangement of axes, the signals of the cross-polarized component with both linear and circular polarization are recorded by the same photodetector.

The rotation angle setting is monitored by the zero position sensors of platforms. When installing the plates in the
platform frame, the plate axis can be shifted relative to the sensor at a certain angle, initially unknown. However, a laser





pulse must be produced at the moment when the axis of the plates coincides with the reference plane of the lidar. Fig. 4b shows a situation, where for plates A and B, from the moment of sensors triggering to the laser pulse, passes 31 ms (45 steps) and 23 ms (34 steps), respectively. The situation repeats every 8 pulses.

To install the plates, a section of a homogeneous atmosphere with small aerosol content is selected. Only one plate is inserted into the channel. Figure 5a shows the lidar signal from two photodetectors, summarized over all positions of plate B (red and green lines). A section from 6 to 9 km was chosen, on which the depolarization ratio (blue line) is constant. Fig. 5b shows the depolarization ratio for each pulse (below) and averaged over a 30 minute record (top). Minimal depolarization is observed at the 34th step of the platform. The accuracy of platform setting is ± 1 step, which corresponds to 0.03% error with respect to depolarization ratio.

**3.3 Calibration of the polarization channels**

Measurements of the polarization of backscattered radiation require careful consideration of polarization distortions in the receiving paths and sensitivity of photodetectors in the channels of original and cross-polarization (Belegante et al., 2018; McCullough et al., 2018; Freudenthaler, 2016). The task of observations in lidar networks is to monitor the optical and microphysical properties of aerosol, which requires restoring not only the backscattering coefficient, but also the lidar ratio

and attenuation. Therefore, a large number of lidars are designed as aerosol-Raman (Reichardt et al., 2012; Madonna et al., 2018; Whiteman et al., 2007; Groß et al., 2015) or multiwave high spectral resolution lidars (HSRL) (Burton et al., 2015; Haarig et al., 2017; Althausen et al., 2000; Eloranta, 2005). Most of these lidars use dichroic beamsplitters as wavelength dividers, and polarizing elements (film polarizers) are installed after the beamsplitters and deflecting mirrors (McCullough et al., 2018; Engelmann et al., 2016; Nott et al., 2012). This leads to distortions in recording of polarization components, which

require complex procedures for determining the eigenvectors of polarization of the lidar and taking them into account when restoring the polarization state of scattered radiation (Haymann et al., 2012; Di et al., 2016; Freudenthaler, 2016; Freudenthaler et al., 2009; Bravo-Aranda et al., 2016; Alvares et al., 2006).

Measurements of polarization characteristics of backscattered radiation simultaneously for several wavelengths were carried out by different groups: 347+697 nm (Pal and Carswell, 1978), 355+532 nm (Groß et al., 2015; Althausen et al.,

2000; Engelmann et al., 2016; Summa et al., 2013), and 355+532+1064 nm (Haarig et al., 2016; Burton et al., 2015). The LOSA-M3 lidar measures polarization components at wavelengths of 532 and 1064 nm. At the same time, the polarization components are distinguished directly behind the receiving telescope, before the radiation is separated according to wavelength. This scheme has no distortions of the polarization state from dichroic mirrors reflection; therefore, there is no need to use laborious calculations for the instrumental vector of the transmitting-receiving channel and correction of the

measured polarization. In this case, it is sufficient to determine the relative sensitivity of detectors in both lidar channels to measure the magnitude of the depolarization ratio. If we do not consider the devices that used a sequential change of polarizations on one photodetector (Platt, 1977; Eloranta and Piironen, 1994; McCullough et al., 2017), the similar problem was solved in all devices for polarization sounding.



The relative sensitivity of photodetectors can be determined by observing the source with known polarization — it can be a non-polarized source (Sassen and Benson, 2001), or an atmospheric layer with purely molecular scattering (Noel and Sassen, 2005; Noel et al., 2002; Kaul et al., 2004; Biele et al., 2000; Volkov et al., 2002). However, the depolarization ratio for molecular scattering depends significantly on the bandwidth of interference filter used, since rotational Raman lines can contribute to the signal (She, 2001; Young, 1980). For pure Rayleigh scattering, the depolarization ratio is δ=0.00365, with a wide filter, δ=0.015. It leads to an ambiguous lidar calibration.

The most common calibration is associated with rotation of the separation plane for polarization components by 90° (Freudenthaler, 2016). In this case, the photodetectors change places with respect to the polarization components. Turning by 90° can be carried out physically by rotating the entire photodetector unit (Yoshida et al., 2010; Strawbridge, 2013), or by rotating the half-wave phase plate, which can be installed both in the transmitter channel (Spinhirne, 1982) and the receiver (McGill et al., 2002; Reichardt et al., 2012). In previous lidars of the LOSA series (Balin et al., 2009), mechanical rotation of the photodetector unit by 90° was used. The intensity ratios in both channels were taken into account and made it possible to eliminate possible changes in the object brightness during turning. When calibrating with this method, you can choose any stable aerosol-cloud formation as a scattering object, which is characterized by a noticeable (>10%) and constant depolarization ratio during the measurement period.

To measure the depolarization ratio of backscattered radiation, the relative sensitivity of detectors must be known. Thus, in our lidar we offer a new method for a continuous rotation of λ/2 phase plate, temporarily installed in the receiver module instead of λ/4 plate *PP3* in Fig. 2. For the light backscattering phase matrix (BSPM), we take a $4\times4$ matrix **M** relating the Stokes vectors of radiation scattered in the direction toward the source $\mathbf{S} = \left[I,Q,U,V\right]^{T}$ with the Stokes vector $\mathbf{S_0}$ of radiation, incident on an ensemble of particles. Polarization components are expressed as $I_{\parallel} = (I+Q)/2$, $I_{\perp} = (I-Q)/2$. Let us assume that laser radiation is linearly polarized (the initial vector is $\mathbf{S}_0^L = \left[1,1,0,0\right]^{T}$). Then the scattered Stokes vector is

$$\mathbf{S} = \mathbf{M}\mathbf{S}_0^L = \left[m_{11}+m_{12}, m_{12}+m_{22}, -m_{13}-m_{23}, m_{14}+m_{24}\right]^{T}.$$ If we install the λ/2 phase plate with matrix

$$\mathbf{L} = \begin{pmatrix} 1 & 0 & 0 & 0 \\ 0 & \cos4\varphi & \sin4\varphi & 0 \\ 0 & \sin4\varphi & -\cos4\varphi & 0 \\ 0 & 0 & 0 & 0 \end{pmatrix}$$ before the polarization prism, where φ is the rotation angle, the Stokes vector will

be $$\mathbf{S} = \mathbf{L}\mathbf{M}\mathbf{S}_0^L = \begin{pmatrix} m_{11}+m_{12} \\ (m_{12}+m_{22})C_4 - (m_{13}+m_{23})S_4 \\ (m_{12}+m_{22})S_4 + (m_{13}+m_{23})C_4 \\ m_{14}+m_{24} \end{pmatrix},$$ where $C_4 = \cos4\varphi$, $S_4 = \sin4\varphi$. Polarization components $I_{\parallel}, I_{\perp}$ are

different $2I_{\perp}^{\parallel} = m_{11}+m_{12} \pm (m_{12}+m_{22})C_4 \mp (m_{13}+m_{23})S_4$ (the upper sign is for $I_{\parallel}$, the lower for $I_{\perp}$). However, the integral



for a complete turn of the plate over the angle φ $\int_{2\pi} I_{\parallel,\perp} d\varphi = \pi(m_{11} + m_{12})$ is the same for each polarization component,

regardless of the values of matrix elements. The ratio of measured values of the integral for two components $K = \int I_\perp d\varphi / \int I_\parallel d\varphi$ gives us the value of relative sensitivity of the polarization channels.

Calibration procedure is done separately for 532 and 1064 nm. The method works for any initial state of polarization and for any BSPM of aerosol layer. Unlike the Δ90 method, there is no need to ensure extremely accurate setting of the plate rotation angle, but during the plate turning (about 20 seconds), change in the scattering properties of the object should be minimal. Fig. 6 shows one of calibration records made on 07 May 2017, demonstrating the errors of the applied method. The upper part (6a) shows the record of a signal integrated by the plate rotation angle, components $I_\parallel$ (green) and $I_\perp$ (blue). Integration was carried out during 4 revolutions of the plate. Fig. 6b shows the value of calibration constant $K$, calculated in the height range of 1800-8000 m. In this case, the value $K=1.91\pm0.05$.

## 4 Observations of ice clouds during lidar zenith scanning

In this paper we present the results of observations of the ice cloud cover under the lidar zenith scanning. Scanning was carried out at the rate of 0.5 degrees per second, which corresponds to 3-minute shift of sounding direction between laser pulses. The angle is measured from the vertical position of radiation beam, which was set with an error of ± 5 minutes. Moreover, in some cases the scanning was carried out with the lidar axis passing through the zenith (ranging from -1 ° to +4°), which made it possible to control the accuracy of the lidar vertical setting.

### 4.1 Zenith scanning at 1064 nm wavelength

Figure 7 shows the sounding data of 6April 2018, 10:25 local time. Fig. 7a gives the component $I_\parallel$ (signal intensity, mV); 7b – the depolarization ratio $\delta^{Circ} = I_\perp / I_\parallel$ (%), measured for the initial circle polarization of laser beam; 7c – the zenith scanning angle (minutes). Data of weather sounding (7d) are taken from the site of the Universty of Wyoming (http://weather.uwyo.edu).

Duration of recording is 300 seconds. In the interval from 120 to 250 seconds, scanning was carried out in the range from 0 to 2 degrees. The behavior of two cloud layers is significantly different. Characteristics of the upper layer (8-10 km) do not change when scanning, which indicates the chaotic orientation of particles in the cloud. In the lower layer (6.5-7 km), a pronounced modulation of signal intensity and polarization is observed, characteristic for a predominantly horizontal orientation of particles. The maximum intensity of signal with the vertical sounding direction corresponds to the minimum of depolarization ratio. An extremely low value $\delta^{Circ}$ in the zenith direction $\delta^{Circ}_{Zen} \approx 4-5\%$ corresponds to the specular reflection.



The other situation is observed on 2 June 2018, 09:55 LT (Fig. 8). Duration of this recording is 550 seconds. The maximum inclination was 4°, the beam passed through the vertical by 1° while scanning (it leads to appearance of double lines of the maximum intensity in Fig.8a). A cirrus cloud with a complex structure occupies the layer of 8 - 11.7 km.

Pronounced modulation of the signal intensity and depolarization ratio are observed throughout the entire height of the cloud. As in the previous case, the maximum intensity of the signal corresponds to the minimum of depolarization ratio. However, the minimum values $\delta_{Zen}^{Circ}$ differ significantly in various parts of the cloud. In the lower part of the cloud, the value $\delta_{Zen}^{Circ} \approx 0.6$ indicates the predominance of chaotically oriented particles and a small proportion of horizontally oriented particles. In the upper part (11.2-11.7 km) $\delta_{Zen}^{Circ} < 5\%$ is characteristic of mirror reflection and indicates a pronounced

horizontal orientation of particles in this part of the cloud. Values outside the vertical are close throughout the entire cloud thickness. This suggests that particles in the cloud differ only in the degree of their horizontal orientation.

**4.2 Dependence of the signal intensity on the lidar tilt angle**

The clouds of the upper layers never constitute a formation uniform in height and constant in time. Pronounced layers with the thickness of hundreds of meters are regularly observed in the structure of ice clouds. They differ in the state of

depolarization and signal intensity from the higher and lower regions and are supposedly homogeneous in composition and degree of particle orientation. However, the signal intensity and the value of the depolarization ratio do not remain constant, but vary with time rather quickly. The height of layers also varies. Weak values of the backscatter signals lead to the need of averaging the signals over the height of the selected layer and over time about 3-5 minutes. As a result, it is necessary to pre-select sections of the cloud, characterized by an approximately constant value of intensity and depolarization ratio, and

lasting for several scan cycles to measure the dependence of echo signal characteristics on the tilt angle. An example of such procedure for selecting the cloud sections to be studied is shown in Fig. 9.

In the given 5-minute recording, 6 sections were selected (marked with rectangles) with the duration of three to seven scan cycles. The rest of cloud portions were not analyzed on this record. In total, about 20 records were selected during observations on June 2, 2018 and October 1, 2018, which have a pronounced dependence on the tilt angle, and there is no

signal overflow when the lidar is oriented to zenith. The following are typical examples of dependences of the intensity of polarization components on the tilt angle.

Figure 10 indicates the dependences of signal intensities of parallel $I_{\parallel}^{circ}$ (circular initial polarization) and cross-polarized $I_{\perp}^{circ}$ components on the tilt angle α. The record in Fig. 10a was registered at $12^{00}$ LT with averaging of characteristics over the layer from 11470m to 11600m. The record 10b was made at $10^{00}$ LT with averaging of characteristics over the layer from

10980 to 11270m. The signal was accumulated during 10 minutes. These measurements and all data, obtained in June and October 2018, are described satisfactorily by the following exponential dependence

$$I(\alpha) = I_0 + A\exp(-|\alpha - \alpha_0|/w) \tag{1}$$





(red line in Fig. 10), where I is the signal intensity, $I_0$ is intensity without the specular component ($\alpha \gg 4°$), A is the constant, depending on the contribution of specular reflection in total intensity, $\alpha$ is the tilt angle, $w$ determines the width of distribution, and $\alpha_0$ indicates the error of lidar targeting to the vertical. For Fig. 10a $w=42$ arc minutes, for Fig. 10b $w=82'$. The Gaussian function used in Noel and Sassen, 2005 (blue line in Fig. 10) poorly describes the observed distribution as it does not take into consideration the intensity peak at $\alpha=0°$.

In most cases the values of $I_\perp$ show a weak decline of intensity with the angle, but these variations are smaller than measurement errors. We can conclude that $I_\perp$ is practically independent of the tilt angle.

Figure 11 gives some selected dependences of intensity angle distributions for component $I_\parallel$. The dependences are normalized to the value $I_0$ obtained by fitting according to the Eq. (1). Squares indicate the tilt angles, corresponding to the distribution of width $w$. For all measurements the value $w$ is within 40-150 arc minutes. The shift of the curve maximum from 0° is less than 2 minutes, which indicates a good lidar orientation to the zenith. Since the signal intensity $I_0$ for $\alpha \gg 4°$ is determined by all particles, the ratio $I_\parallel(0°)/I_0$ shows the proportion of horizontally oriented particles in total mass of crystals in the given section of the cloud.

Previous data related to radiation with initially circular polarization. The degree of depolarization of linearly polarized radiation is generally less than that of circularly polarized radiation. For chaotically oriented particles, the depolarization is less than half (Mishchenko and Hovenier, 1995; Gimmestad, 2008), for oriented particles the difference is smaller, since the BSPM element $m_{12}$ is not equal to zero (Balin et al., 2011). Low value of depolarization results in that the cross-polarized component $I_\perp^{Lin}$ very often does not stand out against the background of the photodetector noise in the specular reflection. Therefore, the dependences $I_\perp(\alpha)$ were investigated for circular polarization. Dependences $I_\parallel(\alpha)$ for linear and circular polarizations for scan angles 0-4° do not differ within the error limits.

**4.3 Angular distributions for green and infrared wavelengths**

The obtained dependence $I_\parallel(\alpha)$ in general terms should correspond to distribution of specularly reflecting particles along the angles of deviation from the horizontal plane (flutter). However, due to the phenomenon of backscattered radiation diffraction on the particle's contour (Borovoi et al., 2008), the distribution is wider than the distribution of particles along the angles of inclination. In addition, the ratio of backscattering coefficients (color ratio) for $\alpha \gg 4°$ (and for randomly oriented particles in any direction) is not equal to unity (Borovoi et al., 2014; Vaughan et al., 2010; Tao et al., 2008). Therefore, the distribution $I_\parallel(\alpha)$ may depend on the radiation wavelength.

In our lidar, measurements for the wavelength of $\lambda = 532$ nm are carried out only for linear polarization of radiation, as the quarter-wave plates in the transceiver are made for the wavelength $\lambda = 1064$ nm. However, in the position where the axis



of the rotating phase plate coincides with the plane of polarization of the emitter, the radiation remains linearly polarized for any wavelength. These pulses are used for measurements at the wavelength of 532 nm.

Two cases with the maximum angular differences are shown in Fig. 12. The relative sensitivity of photodetectors at 532 and 1064 nm was not calibrated; thus, the amplitudes of signals are reduced to ony value. In the left figure the angle distributions are the same, in the right figure the distribution for 532 nm is much wider ($I_{532}\left(4°\right)/I_{1064}\left(4°\right)=1.5$). In all other cases the distribution width $w$ for 532 nm is equal or bigger than for 1064 nm. The reason for such behavior of dependences is not yet clear for the authors.

## 5 Conclusions

Scanning polarizing lidar LOSA-M3 is developed and operated in the Laboratory of Optical Sensing of the Atmosphere (LOSA) at the IAO SB RAS. The main purpose of this lidar is to study the optical characteristics of crystal clouds of the upper and middle layers at two wavelengths - 532 and 1064 nm. Lidar allows a smooth change of the tilt angle from the vertical with simultaneous conical (azimuthal) scanning. Such measurement scheme makes it possible to study the preferential orientation of ice crystals in the clouds in detail. The lidar simultaneously measures the polarization

characteristics of signals for linear and circular initial polarizations, which allows obtaining additional information about the anisotropy of scattering particles, including exploring the azimuthal orientation of the particles. At the same time, the lidar, like all of the LOSA aerosol-Raman lidars, has a Raman scattering channel (607 nm band, for night observations) and a system for combining near and far zones, which allows it to be used for observing aerosol fields in the troposphere distances of 50m - 15km.

During 2018, several series of measurements of the structure of the ice clouds of the upper layers in the zenith scan mode were carried out. The results show that the contribution of horizontally oriented particles, giving a specular reflection, can vary significantly in different parts of the cloud. The dependence of signal intensity on the tilt angle reflects the distribution of the particles deviation relative to the horizontal plane, and is well described by the exponential dependence. Cross-polarized component $I_\perp$ in most cases shows a weak decline with the angle, but its variations are smaller than the

measurement errors. We can conclude that $I_\perp$ is practically independent of the tilt angle. The angle distribution for the radiation for 532 nm in all experiments is equal or wider than for 1064 nm. The reason for such behavior of dependences is not yet clear for the authors.

Author contributions. GP and IP designed the lidar, MN and MK developed the software, GP, YuB, and SN carried out the

lidar measurements, SN and SS carried out the data analysis, GK prepared the manuscript with contributions from all co-authors.



Competing interests. The authors declare that they have no conflict of interest.

Acknowledgements. The reported study was funded by RFBR, according to the Research Project No. 18-55- 81001 and by Project No. AAAA-A17-117021310142-5.

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



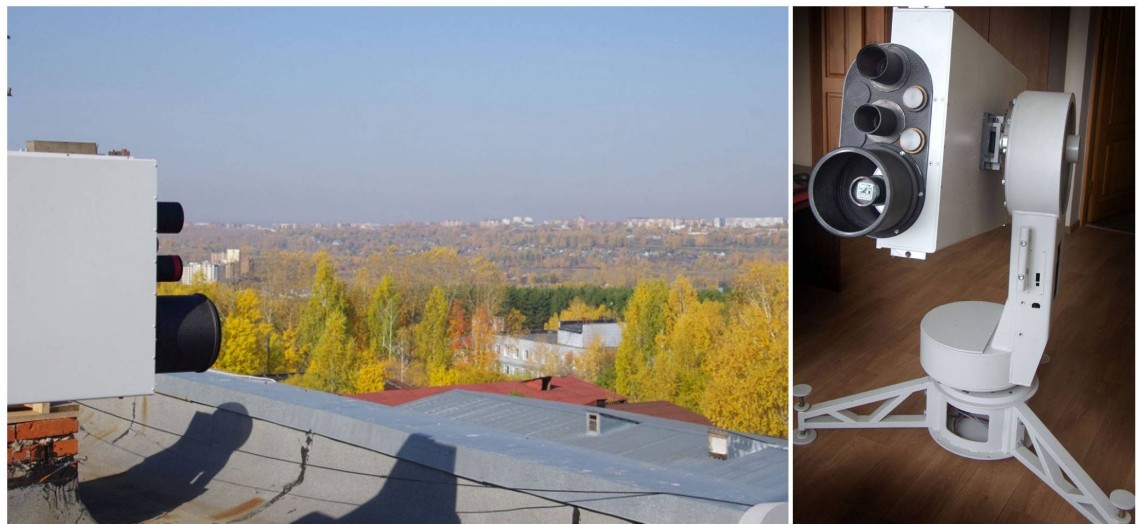

**Figure 1. Appearance of the lidar on a rotary column in the laboratory room (right) and on the institute building roof (left)**

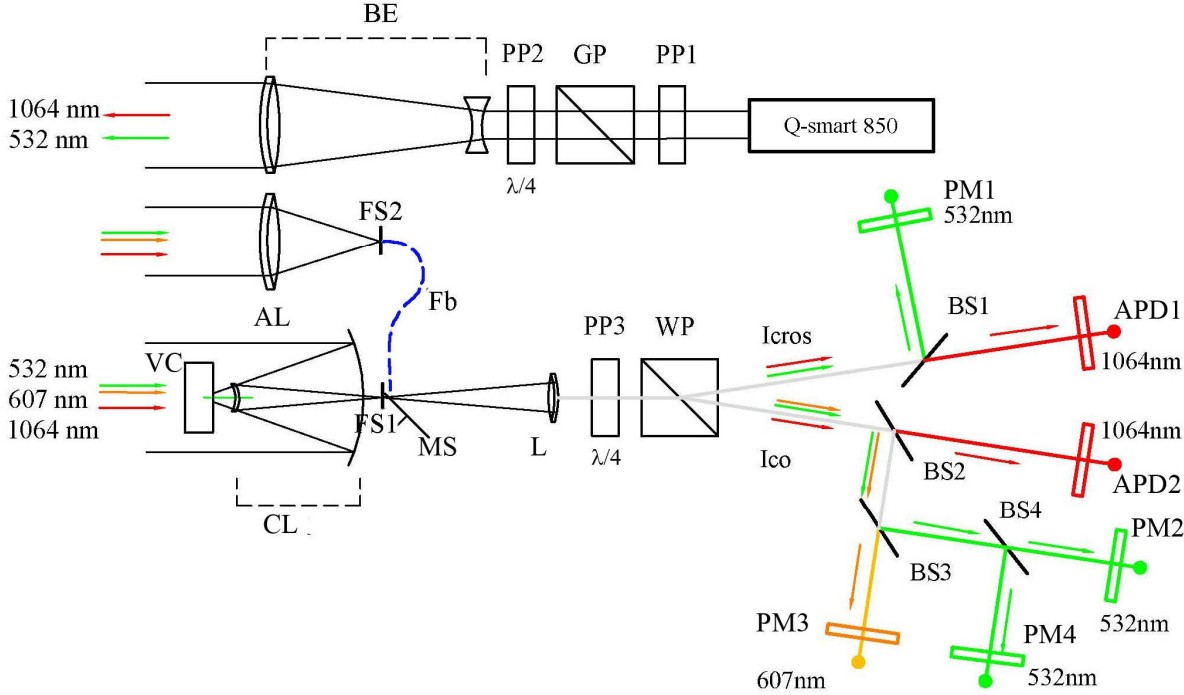


**Figure 2. Optical circuit of the LOSA-M3 lidar**



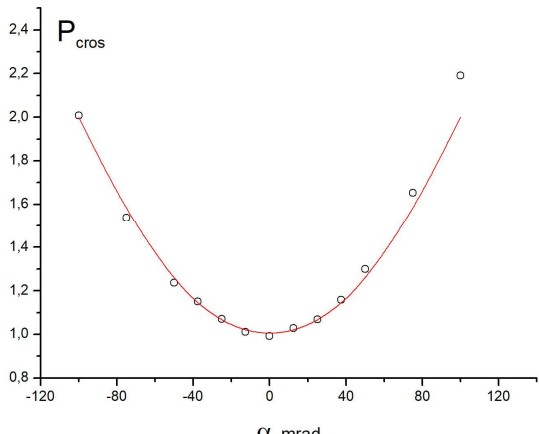

**Figure 3. Adjusting the position of the Glan prism. Measured values $P_\perp$ (circles) and fitting curve (red line)**

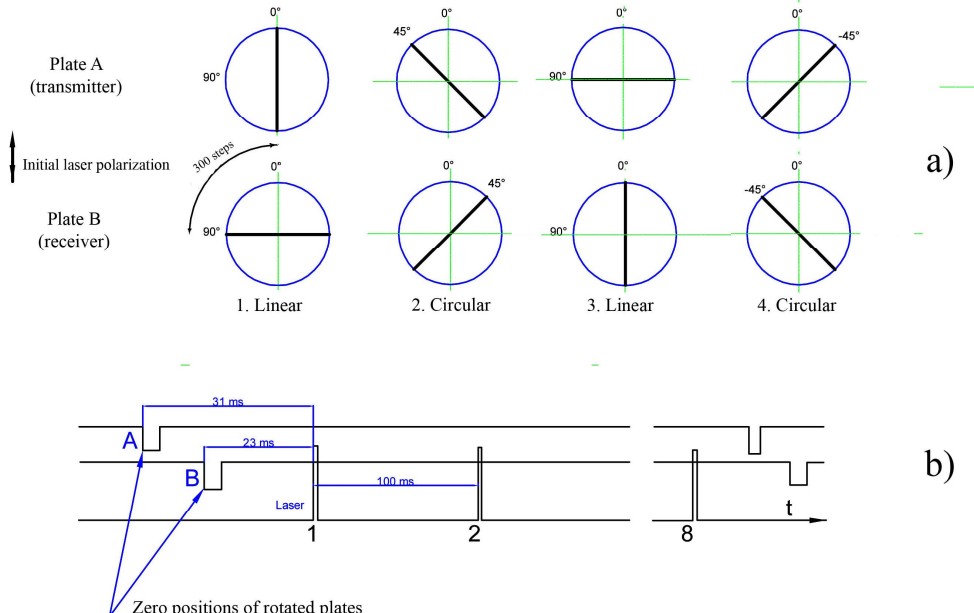


**Figure 4. Diagram of the rotation of the phase plates. a) plate mounting angles for the four consecutive laser pulses; b) a time diagram of laser pulses and rotation sensors.**





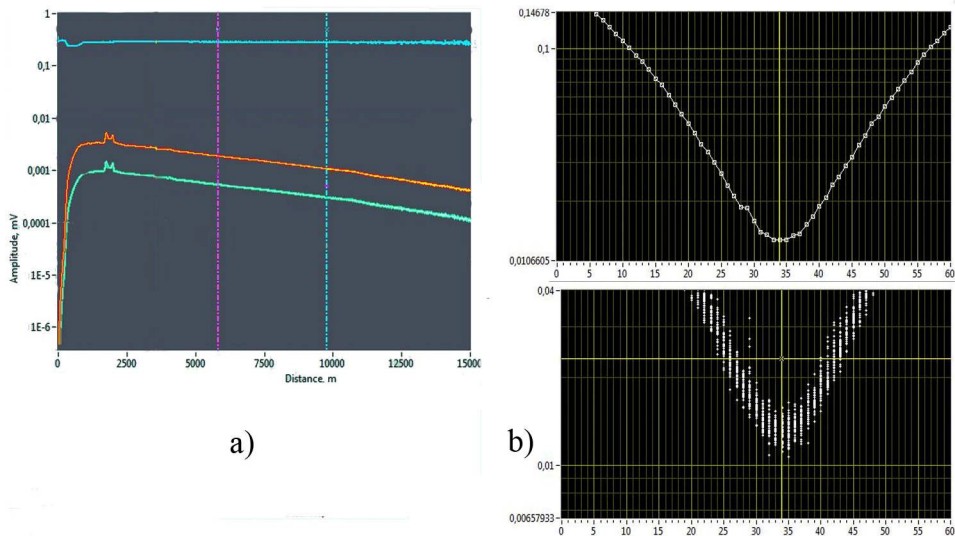

**Figure 5. (a) lidar signal used to mount the plates; (b) depolarization ratio for each pulse (bottom) and averaged over a 30 minute record (top)**

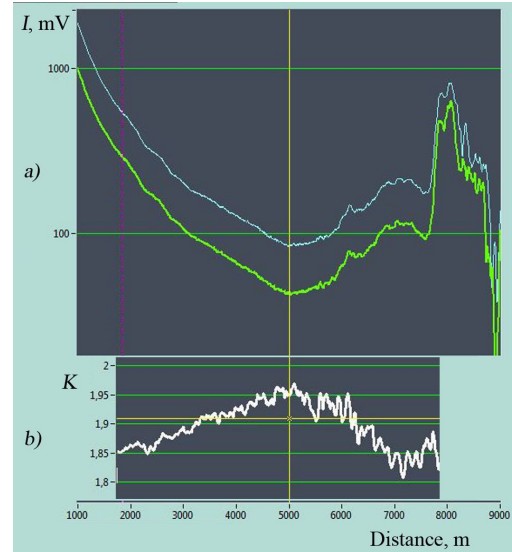

**Figure 6. Record of the calibration procedure of the polarization channels**



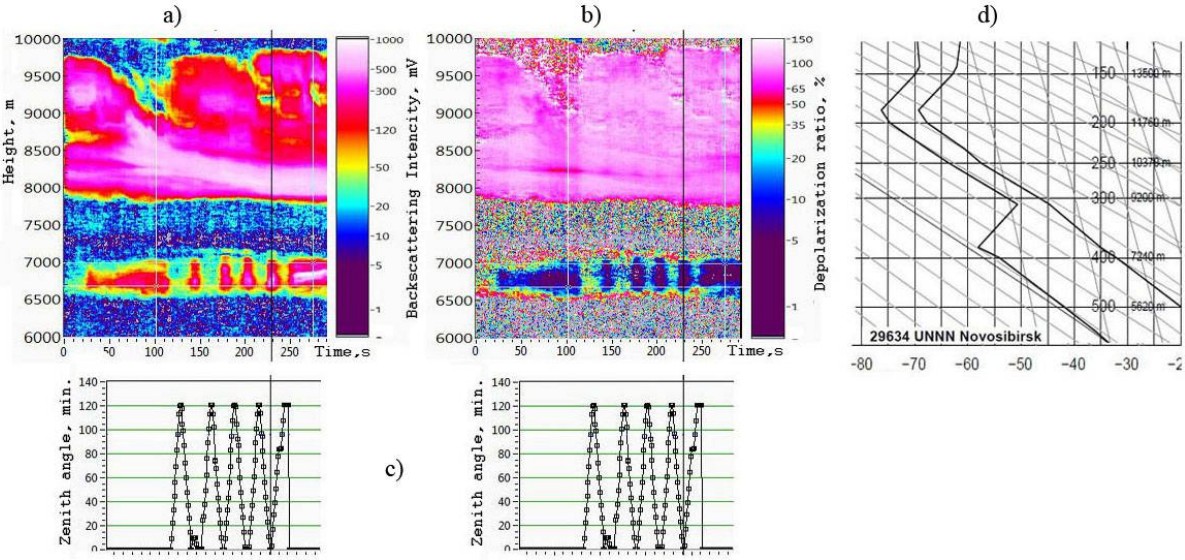


**Figure 7. Sounding data on 6 April 2018. a) intensity of $I_\parallel$ component; b) depolarization ratio $\delta^{Circ} = I_\perp / I_\parallel$ for circle laser polarization; c) zenith tilt angle α; d) weather sounding data.**

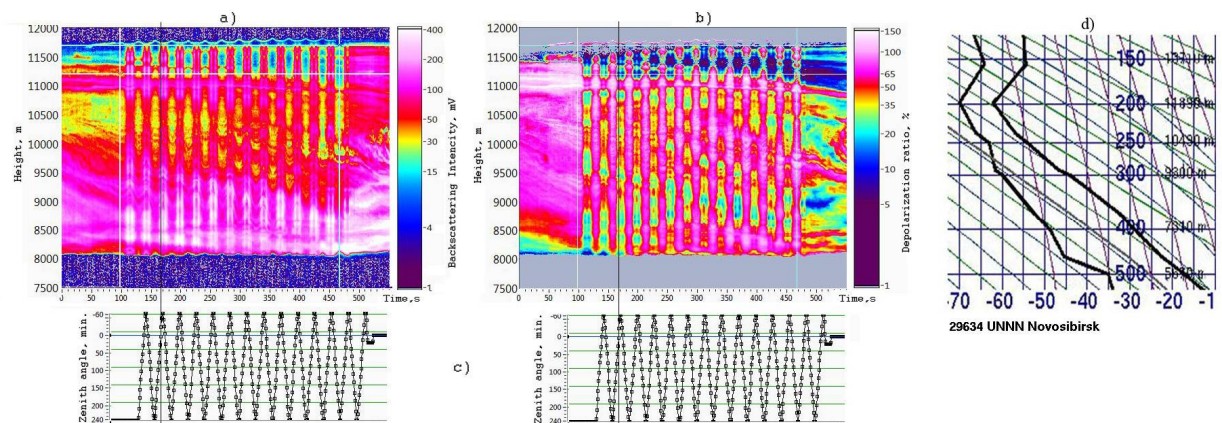

**Figure 8. Sounding data on 2 June 2018. Designations as in Fig. 7**





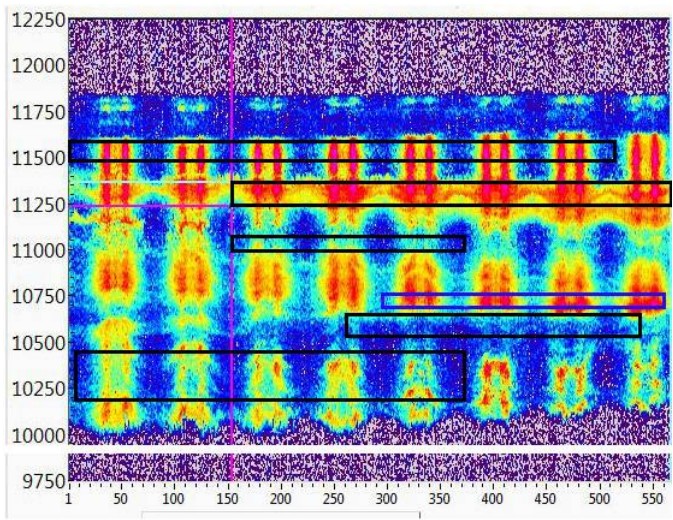

**Figure 9. Selection of cloud sections for further processing. Recorded on 02.06.2018, 12:00-12:05**

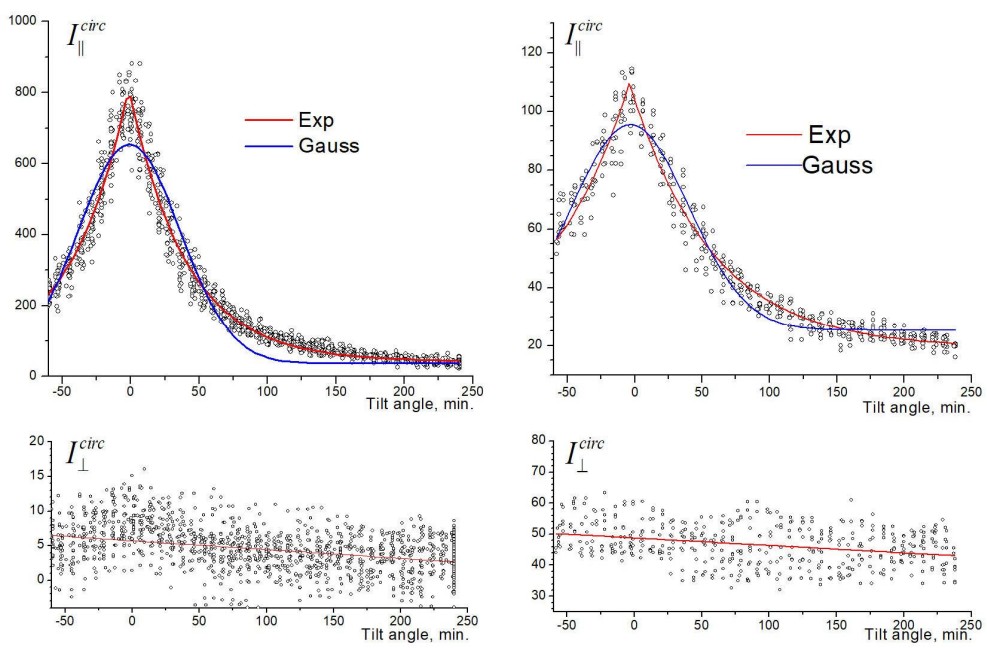

**Figure 10. Angular dependencies of the intensity of polarization components.**



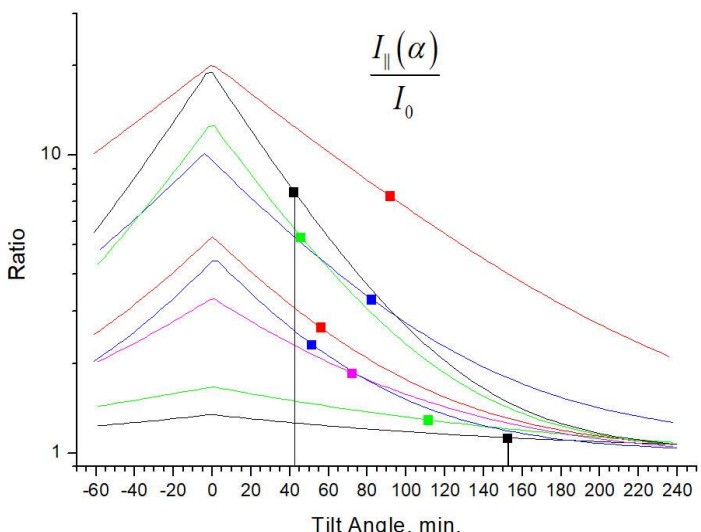

**Figure 11. Selected cases of observed distributions $I_\parallel(\alpha)$. The value of intensity is normalized to the value $I_0$.**

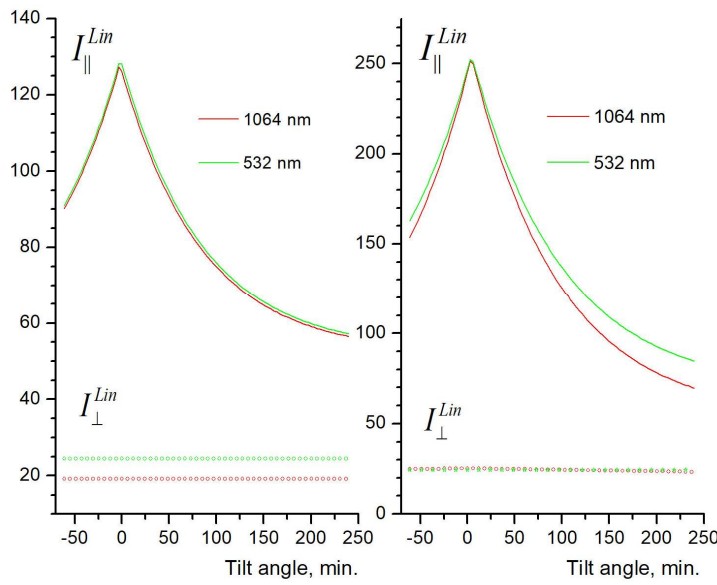

**Figure 12. Angular dependencies of polarization components for 532 nm (green) and 1064 nm (red).**