# Peer review of "Scanning polarization lidar LOSA-M3: opportunity for research of crystalline particle orientation in the ice clouds"

_Atmospheric Measurement Techniques, 2019_

## Referee Comment (RC1) · Anonymous Referee #1 · 20 Sep 2019

The authors describe a two wavelength (532 and 1064 nm) depolarization (circular and linear) scanning lidar designed for detecting and characterizing oriented ice crystals. The instrument design is provided along with some information on the calibration processes. They then present two example cases where oriented ice crystals are observed. Finally they show some analysis on the width of specular reflections of (presumably) plates which is slightly different between the two wavelength channels.

The work presented here is sound and novel. The manuscript represents a complete and well rounded effort. The results will certainly make unique and interesting contributions to the field of atmospheric science. I believe this work should be published after

revisions. A few of these revisions are minor but absolutely mandatory. Others I believe will improve the quality of the presentation but are at the discretion of the authors and editor. Finally I have included several points that I think should be clarified.

**Mandatory Edits:**

Pg 2, Line 51 "Another effect caused by the horizontally oriented columns is the corner reflection when the lidar is tilted at 30 deg..." The 30 deg corner reflection comes from plates, not columns. Columns also have a corner reflection but it is closer to 60 degrees. This is noted in A. Borovoi, I. Grishin, E. Naats, and U. Oppel, "Backscattering peak of hexagonal ice columns and plates," Opt. Lett. 25(18), 1388–1390 (2000).

Pg 2, Line 54. In this paragraph the authors cite several works stating that these works observed both horizontal orientation and azimuthal orientation. This is not true. Most of the references make no mention of observing azimuthally oriented ice crystals which implicitly seems to suggest they didn't observe any. There are a few works (such as Kaul 2004 and Balin 2011) that do mention observing this effect. Beyond that I happen to know number of the researchers cited are very skeptical about the existence of of azimuthally oriented ice crystals outside of thunderstorms. I doubt they would appreciate being cited in support of this claim. The authors need to accurately represent the results of prior work and most of the citations used here do not support the statement or even contradict the statement.

**Discretionary Edits:**

At a number of points the authors draw conclusions about their observations (for example the paragraph on Pg. 8 starting on line 235). The authors should be careful about overasserting what their observations definitively prove about the scattering volume. Hayman 2014 notes in three bullets the difference between what is physically present in the cloud and what is actually observable. Similar conditions would apply to this system. For example, the lack of variation in backscatter with angle does not rule out HOIC. There are cases where HOIC might not generate this signal. For example, HOIC columns probably would not generate the same variation in backscatter, especially over such a narrow range of scan angles. Instead, it seems to rule out the presence of pristine plates. In that context, the authors later assert that the depolarization values are indicators for the relative mass of oriented and randomly oriented ice (Pg. 10 line 284). This is certainly one possibility, but it is also possible that there are crystals that are oriented but have different habits or surface roughness. Also, the assertion that this ratio is connected to mass seems very tenuous.

It is very challenging to determine anything definitive from most oriented ice crystal observations. Presumably one would need a sophisticated microphysical model (to capture temporal evolution of ice habit, roughness, size, etc) coupled with an advanced scattering model. By themselves the observations are underconstrained.

My suggestion is that the authors use a few words to reduce the certainty of their statements. For example: "It is possible that this is because..." They might also include listing some other possible explanations for what is happening. In my opinion this would enhance the value of the paper because it would keep readers engaged with the challenges that oriented ice crystals pose.

Pg 11, Line 316 "... including exploring the azimuthal orientation of particles." I suggest being clear that this was not explored in the current work and that looking for azimuthal orientation of particles would be in future work.

Points for clarification:

Pg 3, Line 62-64 It is not clear how the authors come to the conclusion that m12=-0.22 + / -0.2 means that in 30% of the observational cases, the depolarization depends on the lidar reference plane. The value of m12 certainly does not dictate this. Is the assumption that PDF of m12 is Gaussian?

Pg 3, Line 64 "In other words..." This statement isn't totally clear. I think to clarify you want to say "...when the lidar's linear polarization rotates around..."

With regard to the near range channel, I'm a little confused about what its purpose is. Doesn't the fiber scramble the polarization modes and therefore prevent measurement of the depolarization ratio with this channel? What is this channel being used for? What ranges are the near and far range channels used for?

Pg. 5 Line 145 Just to be clear, this analysis of error in polarization angle does not translate into a baseline system error. The authors don't mention this explicitly, but there are almost certainly some effects unaccounted for (e.g. waveplate behavior, mechanical and optic axis misalignments, etc) in any system. It doesn't necessarily have to be published, but the authors should have some idea about the accuracy limits of their system and based on the data presented here, it is probably larger than 0.016%. One useful test is to look at how close the depolarization comes to zero in liquid clouds (before multiple scattering contaminates the signal).

Pg. 10 Line 278 The authors state that the signal variations with angle are smaller than the measurement errors. This really depends on what the authors mean by "measurement errors" because one can clearly see a trend in the data, so the limiting factor does not appear to be random error. If they mean this is less than the systematic error of the instrument, this is certainly a valid point. It would be good to clarify which type of errors they are referring to.

Also with regard to Figure 10 and the perpendicular measurements, I wonder if this angle dependence is the result of cross talk between the channels. Perhaps this is what the authors are referring to as "measurement error". If so it would be helpful to simply state that explicitly.

Pg. 10, Line 300 The authors describe that the depolarization measurements at 532 nm are only made for linear polarizations. This needs to be better explained in section 2 Lidar Description. I had assumed (incorrectly) that the authors were using a dual wavelength wave plate. Make it clear what wavelengths the polarization optics are designed for and please be clear throughout the manuscript that this instrument performs

the two polarization measurements only at 1064.

---

## Author Comment (AC2) · 30 Sep 2019

[revised manuscript text omitted]

50  Gaussian distribution.

Another effect caused by the horizontally oriented plates and columns is the corner reflection. It appears when the lidar is tilted at a significant angle. Depolarization of backscattered radiation has a maximum at a lidar tilt of 30 degrees for oriented plates and about 60 degrees for columns (Borovoi et al., 2000). A more detailed simulations show that when angular reflection is taken into account, the element $a_{44}$ of the light backscattering phase matrix (BSPM) is most informative in

55  determining of the flutter (Konoshonkin, 2016). For the plates, an abrupt change in the $a_{44}$ occurs at a tilt angle of 30 degrees, while for the columns this element begins to grow smoothly at 30 degrees and reaches a maximum at 60° (Konoshonkin et al., 2016). Some experiments with tilted lidars were carried out (Del Guasta et al., 2006; Hayman et al., 2012, 2014; Neely et al., 2013, Veselovskii et al., 2017) and showed a high probability of the presence of oriented particles.

Particle orientations are promoted not only by aerodynamic forces, but also by forces of a different nature, such as wind

60  shifts and electric fields. Kaul 2000 supposed that in such conditions, crystalline particles can have a preferential orientation in the horizontal plane (azimuthal orientation). It is obvious that the direction of preferential orientation is connected with the direction of action of these forces. The basis for such conclusion is the observed non-invariance of the BSPM with respect to

rotation of the coordinate system (Kaul 2000, Kaul et al. 2004, Samokhvalov et al. 2014, Hayman et al., 2014). If we use linear polarized radiation and rotate the lidar around a vertical axis, signal energy depends on the angle of lidar rotation relative to the plane of preferential orientation of the particles if $m_{12} \neq 0$ (Kokhanenko et al., 2018). According to a large array of experimentally measured BSPM, the distribution of relative frequencies for $m_{12}$ shows that the value of $m_{12}$ lies in the interval [-0.6, -0.3] with the probability about 30% (Kaul et al., 2004, Balin et al. 2009). Therefore we would have to observe the modulation of the signal very often. However, authors are unaware of such direct measurements.

[revised manuscript text omitted]

We have two sets of quarter-wave plates. One set is designed for 532 nm, another set for 1064 nm. For the wavelength of installed plates (below are the results only for installed 1064 nm plates) we can investigate both linear and circular polarizations. For the second wavelength (532 nm) polarization state when turning 45 degrees is not determined. However, in the position where the axis of the rotating phase plate coincides with the plane of polarization of the transmitter, the radiation remains linearly polarized for any wavelength. So the measurements for the wavelength of λ = 532 nm were carried out only for linear polarization of radiation. Of course, in our lidar we can use a quarter-wave plates for 532 nm if such experiments were planned.

The beam is collimated by the 7-fold achromatic expander *BE*, designed at the IAO SB RAS (Kochanenko et al., 2012). Two receivers are used – an achromatic lens *AL* with the 40 mm diameter and 200 mm focus for the near zone, and Cassegrain mirror lens *CL* with the  200 mm diameter and 1000 mm focus for the far zone. The iris diaphragms *FS1*, *FS2* in the focal plane of each lens determine the telescope field of view. A special feature of the Cassegrain lens design is installation of a video camera *VC* behind the secondary mirror, which is getting radiation through an annular diaphragm in the outer area of the secondary mirror (Simonova et al., 2015). The camera has a global shutter and is synchronized along with laser pulses. This camera setup allows observing the image of the laser spot on the objects around without parallax. It is especially important for setting the lidar field of view and for excluding the possibility of lidar orientation towards residential buildings.

The signal from the receiver of the near zone through the optic fiber *Fb* is fed to the mirror shutter *MS*, by means of which the signals of the near and far zones are alternately switched. The small receiver is closer to the transmitter, so the transition to range-square mode starts earlier (80-100 m) than for the large receiver (800-900 m). During data processing

signals from near zone (50-1200 m) and far zone (400 m-15 km) fit together at a distance of 800-900 m when range-square mode for large receiver starts. Of course, the optic fiber destroys the polarization state of the signal, so near-zone data cannot be used for polarization analysis.

[revised manuscript text omitted]

Kokhanenko, G.P., Balin Yu.S., Borovoi A.G., Klemasheva M.G., Nasonov S.V., Novoselov M.M., Penner I.E., Samoilova S.V. "Investigations of the crystalline particle orientation in high-level clouds with a scanning lidar," ", Proc. SPIE 10833, 24th International Symposium on Atmospheric and Ocean Optics: Atmospheric Physics, 1083347 (13 December 2018); https://doi.org/10.1117/12.2504129

Konoshonkin, A.V., **"Simulation of the scanning lidar signals for a cloud of monodisperse quasi-horizontal oriented particle,"** Optika Atmosfery i Okeana, 29, No. 12, 1053–1060 (2016) [in Russian].)

Konoshonkin, A., Wang, Zh., Borovoi, A., Kustova, N., Liu D., and Xie, Ch.: Backscatter by azimuthally oriented ice crystals of cirrus clouds, Opt. Express, 24(18), A1257-1268, 2016.

Lavigne, C., Roblin, A., and Chervet, P.: Solar glint from oriented crystals in cirrus clouds, Appl. Opt., 47(3), 6266–6276, 2008.

Liou, K. N.: Influence of cirrus clouds on weather and climate processes: a global perspective, J. Geophys. Res., 103, 1799–1805, 1986.

Madonna, F., Rosoldi, M., Lolli, S., Amato, F., Vande Hey, J., Dhillon, R., Zheng, Y., Brettle, M., and Pappalardo, G.: Intercomparison of aerosol measurements performed with multi-wavelength Raman lidars, automatic lidars and ceilometers in the framework of INTERACT-II campaign, Atmos. Meas. Tech., 11, 2459–2475, 2018.

Masuda, K. and Ishimoto, H.: Influence of particle orientation on retrieving cirrus cloud properties by use of total and polarized reflectances from satellite measurements, J. Quant. Spectrosc. Radiat. Transfer, 85, 183–193, 2004.

McCullough, E. M., Sica, R. J., Drummond, J. R., Nott, G., Perro, C., Thackray, C. P., Hopper, J., Doyle, J., Duck, T. J., andWalker, K. A., Depolarization calibration and measurements using the CANDAC Rayleigh–Mie–Raman lidar at Eureka, Canada, Atmos. Meas. Tech., 10, 4253–4277, 2017.

McCullough, E. M., Sica, R. J., Drummond, J. R., Nott, G. J., Perro, Ch., and Duck, T. J.: Three-channel single-wavelength lidar depolarization calibration, Atmos. Meas. Tech., 11, 861–879, 2018.

McGill, M. J., Hlavka, D. L., Hart, W. D., Spinhirne, J. D., Scott, V. S., and Schmid, B.: The Cloud Physics Lidar: Instrument description and initial measurement results, Appl. Opt., 41, 3725–3734, 2002.

Mishchenko, M. I. and Hovenier, J. W.: Depolarization of light backscattered by randomly oriented nonspherical particles, Opt. Lett., 20, 1356–1358, 1995.

Neely, R. R., Hayman, M., Stillwell, R. A., Thayer, J. P., Hardesty, R. M., O'Neill, M., Shupe, M. D., and Alvarez, C.: Polarization Lidar at Summit, Greenland for the Detection of Cloud Phase and Particle Orientation, J. Atmos. Ocean. Tech., 30, 1635–1655, 2013.

Noel, V. and Chepfer, H.: A global view of horizontally oriented crystals in ice clouds from Cloud-Aerosol Lidar and Infrared Pathfinder Satellite Observation (CALIPSO), J. Geophys. Res., 115, D00H23, 2010.

Noel, V. and Sassen, K.: Study of ice crystal orientation in ice clouds from scanning polarization lidar observations, J. Appl. Meteorol., 44(5), 653–664, 2005.

Noel, V., Chepfer, H., Ledanois, G., Delaval, A., and Flamant, P. H., Classification of particle effective shape ratios in cirrus clouds based on the lidar depolarization ratio, Appl. Opt., 41(21), 4245–4257, 2002.

Nott, G., Duck, T., Doyle, J., Coffin, M., Perro, C., Thackray, C., Drummond, J., Fogal, P., McCullough, E., and Sica, R.: A remotely operated lidar for aerosol, temperature, and water vapor profiling in the High Arctic, J. Atmos. Ocean. Tech., 29, 221–234, 2012.

Pal, S. R. and Carswell, A.I.: Polarization properties of lidar scattering from clouds at 347 nm and 694 nm, Appl. Opt., 17(15), 2321-2328, 1978.

Platt, C. M. R.: Lidar observations of a mixed-phase altostratus cloud, J. Appl. Meteorol., 16, 339–345, 1977.

Platt, C. M. R.: Lidar backscatter from horizontal ice crystal plates, J. Appl. Meteor., 17, 482–488, 1978.

Platt, C. M. R., Abshire, N. L., and McNice, G. T.: Some Microphysical Properties of an Ice Cloud from Lidar Observation of Horizontally Oriented Crystals, J. Appl. Meteorol., 17(8), 1220–1224, 1978.

Reichardt, J., Wandinger, U., Klein, V., Mattis, I., Hilber, B., and Begbie, R.: RAMSES: German Meteorological Service autonomous Raman lidar for water vapor, temperature, aerosol, and cloud measurements, Appl. Opt., 51(34), 8111-8131, 2012.

Samokhvalov, I.V., Nasonov, S.V., Stykon, A.P., Bryukhanov, I.D., Borovoi, A.G., Volkov, S.N., Kustova, N.V., and Konoshonkin, A. V., "Investigation of phase matrices of cirrus containing ensembles of oriented ice particles," Proc. SPIE 9292, 20th International Symposium on Atmospheric and Ocean Optics: Atmospheric Physics, 92922M (25 November 2014); doi: 10.1117/12.2075562

Sassen, K.: Ice crystal habit discrimination with the optical backscatter depolarization technique, J. Appl. Meteor., 16, 425–431, 1977.

Sassen, K.: Corona-produsing cirrus cloud properties derived from polarization lidar and photographic analyses, Appl. Opt., 30(24), 3421-3428, 1991.

[revised manuscript text omitted]

---

## Referee Comment (RC2) · Anonymous Referee #2 · 8 Oct 2019

The authors describe a scanning dual-wavelength lidar capable of measuring the linear (at 1064 and 532 nm) and circular (at 1064 nm) depolarization ratio. They carefully characterize the polarization properties of their well-designed lidar system. The near-zenith scan is applied to several cirrus measurements to demonstrate the advantages of their system. For the intensity of the horizontal plane of polarization an exponential dependence with the tilting angle was found with maximum values at $0°$ (zenith). The new lidar system (LOSA-M3) will enable advanced studies of ice clouds and (in future) aerosol layers. Therefore, I recommend it for publication in AMT. However, Section 4 and the corresponding figures need major revisions.

[Figure]

General (major) comments:

1. "Clouds of Upper Layers" in the title sounds strange. It would be better to replace it with "ice clouds" or "clouds in the upper troposphere". Why you mention "upper layers", you can as well observe clouds and ice clouds in lower parts of the troposphere. In general, I would omit the term "upper layers" throughout the whole manuscript. It can be replaced by "upper troposphere".

2. I miss a bit the discussion about the atmospheric relevance of your findings. What does the additional information we get by scanning through the cirrus help us in characterizing cirrus clouds? It is interesting to know, if a cirrus cloud consists of orientated or randomly orientated ice crystals. In the introduction you mention the sun glare. You could add discussion about the atmospheric implications of your findings. This will further underline the importance of your newly developed lidar system.

3. You give an exponential parameterization (equation 1). But the reader finds nowhere in the manuscript any parameters for this fit. You should definitely provide some fitting parameters for your curves (A, alpha_0, w).

4. To discuss the differences in the cirrus observations, it would be extremely helpful to provide some more information about the cirrus cloud.

Firstly, the temperature profile within the cirrus. You show some radiosonde data in Fig. 7+8, but you don't use this information in the text. At which temperature do you observe the two cirrus clouds on 6 April 2018? At colder temperatures, the ice crystals may have different properties. To improve the Figures, I would show a temperature profile exactly for the same height range (6 – 10 and 7.5 – 12 km) as in Fig. 7a+b and 8a+b instead of the shown diagram. And please add the time of radiosonde launch.

Secondly, the different exponential behaviors in Fig. 11 are related to different cirrus clouds. What additional information do you have about these cirrus clouds? Cloud height? Cloud thickness? Cloud top temperature? Temperature profile within the

cloud? Age of the cirrus cloud? Formation process? May this information help to explain the different behavior?

5. Where did you perform the measurements? In Tomsk. Can you add some coordinates? How far was the radiosonde station?

6. How did you select the measurements in Fig. 11 (ln 280)? Which criteria did you use?

7. The symbols âĹě and âŁě correspond to parallel and orthogonal normally linked to linear polarization. Circular polarization is right handed or left handed or more general it can be described as co-polar and cross-polar. Or at least mark the intensity as a circular polarized component whenever it is used to not confuse the reader with the linear polarization.

In general, you should be more careful in distinguishing the linear and circular depolarization ratio throughout the text (often it is just stated "depolarization ratio").

8. The paragraph line 286-292 describing the relation of circular and linear polarization should be placed earlier. The same holds for the information in line 300-303. Till these lines, it remained unclear how you deal with two wavelengths and a quarter wave plate. This has to be mentioned when describing Fig. 4.

9. Overall the figure captions are very short. The explanation is given in the text, but it would be very useful to include this information in the figure caption. The reader must be able to understand a figure without reading the whole text. You never know, how the final type setting of your paper will look like. Sometimes the text is quite far from the corresponding figures.

10. The figures need some improvements to be publishable:

Bring Fig. 5 + 6 in a better shape, e.g., with a white background.

Add units and/or title to the y-axis in Fig. 3, 5b, 9, 10, 11, 12

Add units and/or title to x-axis in Fig. 5b, 7c, 8c, 9

Add color bar to Fig. 9

Indicate the used wavelength in Fig. 3, 5a, 7, 8, 9, 10, 11

Indicate the date and time (and height) of the observations in Fig. 10, 11, 12

Use a better plot to illustrate the height profile of the temperature (and other necessary meteorological parameters) – Fig. 7d + 8d

Specific (minor) Comments:

11. Some formulations are not well-suited throughout the whole manuscript:

"upper layers" – use "upper troposphere" or "high altitude (clouds)"

"near zone", "far zone" – use "near range", "far range"

"chaotically oriented" – use "randomly oriented"

"sounding" – better use "measurement" or "lidar measurement" (only for the radiosonde "sounding" would be the appropriate term)

12. The date should be always in the same format: 2 June 2018 – change ln 264 and Fig. 9

13. line (ln) 34 "However" seems not necessary at this point.

14. ln 45 "mid latitudes"

15. ln 67 "the polarization characteristics are measured at . . ."

16. ln 89 "to evaluate some elements of BSPM" Which elements? Be more precise.

17. ln 89 "The lidar appearance is shown in Fig. 1" – "A photograph of the lidar system is shown in Fig. 1" – In the photograph it is pointing horizontally.

18. ln 94 "PP1 with the phase shift of 20 wavelengths is used for $\lambda$ = 532 nm, and 9.5

wavelengths for $\lambda$ = 1064 nm." What do you mean by this?

19. ln 101 "Usually, it takes from ten seconds to ??? minute." – one minute?

20. ln 144 "($\alpha$ is a small setting angle error)."

21. ln 146 Where do you get this value from?

22. ln 148-150: 45° * 0.68 ms /0.3° = 102 ms Using the information you provided, the quarter wave plate would need 102 ms to turn by 45°. That would be too slow for a laser repetition rate of 10 Hz. Maybe you just have to report one more significant digit for time?

23. ln 157-158 Here it would be helpful to already mention Fig. 5. Otherwise, the number of steps seems somehow arbitrary.

24. Fig. 5a Why do you show this plot? To my opinion Fig. 5b would be enough to understand the procedure. ln 160 "the lidar signal from two photodetectors" – From which photodetectors?

25. ln 159-164: The same procedure is done for plate B without plate A, isn't it?

26. ln 172: Haarig et al., 2017 and Althausen et al., 2000 do not describe HSR lidars such as Burton et al., 2015 and Eloranta, 2005, but Raman lidars with several wavelengths.

27. ln 180 "and 355+532+1064 nm" I would consider to add a reference to Hu et al., ACP 2019 and change the Haarig et al., 2016 reference to Haarig et al., 2017.

28. ln 211 "before the polarization prism in the receiver"

29. ln 220 The calibration was made 7 May 2017, the measurements are performed one year later. Did you perform calibration measurements in 2018 as well? What can you say about the stability of such calibration measurements?

30. ln 227 "with an error of $\pm$ 5 minutes" These are arc minutes. At this point the reader

may easily be confused with minutes in time.

31. ln 232 "initial circular polarization"

32. Why do you study the range -1 to 4° only? Many lidar systems are operated at an off-zenith angle at 5°. It would be interesting to extend your tilt angle up to 5°, even if the change from 4° to 5° will not be significant.

33. ln 250 "Values outside the vertical are close throughout the entire cloud thickness." – Close to what?

34. ln 255 "However, the signal intensity and the depolarization ratio do not remain constant"

35. ln 265-266 " The following . . ." already said before

36. ln 273 How to determine I_0 (for alpha » 4°) if the scanning cycles are only done up to 4°?

37. ln 278-279 Can you provide a mean and a standard deviation? Or maybe add it as a dashed line in Fig. 10.

38. ln 287+289 "depolarization ratio"

39. ln 305 "thus, the amplitudes of signals are reduced to one value" – How?

40. Fig. 1 Change order: First describe the left and then the right picture. It makes more senses.

41. Fig. 2 Add information on photocounting and analog detection in the picture. Otherwise, it is not clear why do you have two 532 nm detectors (PM2 and PM4) in the co-polarized branch.

42. Fig. 4a What does the green line stands for? Why green? The circular polarization is only measured in the NIR.

Fig. 4b needs more explanation.

43. Fig. 5 What signals do I see? The same question for Fig. 6.

44. Fig. 7a + 8a in the title of the color bar "Backscattering Intensity, mV"

45. Fig. 9 Do you use all data marked by a box? Where do you use it?

46. Fig. 10 Give somewhere the fit values for the exponential and Gaussian fit.

47. Fig. 11 You just show some fitting results without showing the original data points. Can you underlay your fitting curves (in bold) with the data points in the corresponding color (in a light hue). Then, the reader will see the data used for these fits.

Additionally you should definitely state in the caption or legend which data are used (date, time and probably height). Otherwise, no one will know which curve corresponds to which event even if you don't show all events (not necessary).

In general, this is a very interesting figure, showing the different behavior of cirrus clouds. The exponential decay with tilt angle is an important finding. Maybe the different width of the distribution can be used to extract more information about the state of the cirrus cloud (and its forming process or aging). And it is good to see that it holds for the circular and linear depolarization ratio (Fig. 12). As mentioned before, it would be nice to have some fit parameters for the exponential dependence. I am already curious to see some results in highly depolarizing aerosol layers. But in general the aerosol particles should be randomly orientated. It is a long list of comments made to improve the manuscript.

---

## Referee Comment (RC3) · Anonymous Referee #3 · 15 Oct 2019

The manuscript reports on a new scanning lidar instrument specifically designed for simultaneous measurements of linear and circular depolarization ratios of cloud particles. Because of this capability its main field of application is probably studies of elastic light-scattering effects associated with particle orientation in atmospheric ice clouds. After a thorough literature review on cloud-particle alignment and relevant observation and analysis techniques, a technical description of the instrument is presented, focusing on the depolarization-ratio measurements with the far-range telescope. The lidar receiver's apparently depolarization-insensitive fiber-coupled near-range subsystem and its Raman detection capability are barely mentioned. Calibration methods are discussed in detail, of which the technique for determining the relative sensitivity of

the polarization channels is particularly noteworthy. Measurement examples highlight zenith scans of circular depolarization ratio in cirrus clouds. Main results are that, (1) the cross-polarized component is almost independent of lidar tilt angle while, (2) the co-polarized component is found to have an exponential dependence with a distribution width of about $0.7°$-$2.5°$. The manuscript is well written, the results are interesting and worth publishing. Clarifications are required.

Experiment:

1. Lines 114 f.: Please, provide more information about the fiber (polarization-preserving?) and the shutter (coating?).

2. How is background scattering suppressed? There seem to be no filters in the setup, is this correct?

3. Lines 149-150: 150 Steps are required for a $45°$-turn, which would take 102 ms (according to the information provided) and thus slightly longer than the time period between the 10-Hz laser pulses. Please, comment.

4. Lines 159-160: 'Only one... channel'. Please, provide more details.

Measurement examples:

1. Lines 242 ff.: It is not obvious what is meant with 'double lines'.

2. Line 264: Figs. 7-9 present data from April and June, 2018. Then, suddenly, 1 October is mentioned. Please, provide earlier on in the section an overview of the measurements to be discussed.

3. Paragraph, lines 286 ff.: This information must be provided before the measurements are presented, because otherwise the interested reader is waiting for the linear depolarization ratios to be shown.

4. Paragraph, lines 300 ff.: This information definitely belongs to section 2 or 3!

References:

1. Differences in citation style!

2. Line 465: Summa et al. have to be moved further down in the reference list.

3. Line 467: Thomas et al. have to be moved further down in the reference list.

Figures (Styles vary considerably. Please, try to make appearance as uniform as possible):

1. Fig. 3: Y-axis title.

2. Fig. 5: Axis titles.

3. Fig. 7: Panels a-c need to be as large as possible. Panel d is probably not necessary.

4. Fig. 8: Panels a-c need to be as large as possible (and same size as in Fig. 7). Panel d is probably not necessary.

5. Fig. 9: Axis titles.

6. Fig. 9, caption (and running text): Please, use always the same date style, for instance, 6 April 2018.

7. Figs. 10-12: It is irritating that the y-axis titles are not attached to the axis.

Phrasing (Some sentences are difficult to understand. Please, consider rewording):

1. Line 100: 'Immediately'

2. Line 104: 'This... time.'

3. Line 182: 'distinguished' -> 'separated'

4. Lines 186-188: 'If... sounding.'

5. Line 269: 'record in Fig. 10b'

6. Line 305: 'thus... value.'

Typos:

1. Line 53: 'Neely et al.'

2. Line 55: 'Kaul et'

3. Line 66: 'The authors', full stop missing

4. Line 177: 'Alvarez'

5. Line 198: 'Spinhirne et al.')

6. Line 231: '6 April'

7. Line 233: 'University'

8. Line 244: 'Fig. 8a'

9. Line 305: 'ony' ?

10. Line 329: 'GP' is not on the author list.

11. Line 341: 'Alvarez'

12. Line 367: 'Burton' ?

13. Line 436: 'and Walker'

---

## Referee Comment (RC4) · Anonymous Referee #4 · 27 Oct 2019

General comments:

The paper describes a new lidar system for monitoring oriented ice crystals in clouds. My comments will focus mainly on the lidar system, since I'm not an expert in cloud remote sensing. Overall I think this work is worth publishing in AMT.

Specific comments:

The lidar description needs to be revised in order to be provided in a more clear way to the reader. Some examples:

1. Provide a full description of the various symbols in Figure 2, in the caption of the

figure.

2. Provide a full description of the measurement sequence, in terms of the measurements at near and far zones, measurements at different wavelengths, measurements with linearly- and circularly-polarized emission (and corresponding detection), so the sequence of the measurements and their time resolution is clear. The use of a new figure to provide this sequence visually would help.

3. The system relies heavily on its rotating parts, but in the text there is not much information about their synchronization. Please provide your comments on this and/or the tests you performed to check for this.

Some more comments:

1. Make Fig. 4a and 4b two different figures. It is confusing to be in the same figure, because the first refers to the rotation of the phase plates and the second refers to the definition of their initial position.

2. Change caption of Fig. 5a to "lidar signal used to mount the plates at their initial position"

---

## Author Comment (AC3) · 28 Oct 2019

Response to Referee #2

The authors are sincerely grateful for the careful reading of the work. Your comments are very useful and will be taken into account when finalizing the text. Some of them, especially concerning the refinement of the drawings, required a long time. I will now respond to comments and add a revised text.

1. "Clouds of Upper Layers" in the title sounds strange. It would be better to replace it with "ice clouds" or "clouds in the upper troposphere". Why you mention "upper layers", you can as well observe clouds and ice clouds in lower parts of the troposphere. In general, I would omit the term "upper layers" throughout the whole manuscript. It can be replaced by "upper troposphere".

I agree that the term "layer" is incorrect and very unfortunate. The terms "High-level" or "Mid-level clouds" are recommended by the International Cloud Atlas. I will try to use these terms in the manuscript and "ice clouds" in the title.

2. I miss a bit the discussion about the atmospheric relevance of your findings. What does the additional information we get by scanning through the cirrus help us in characterizing cirrus clouds? It is interesting to know, if a cirrus cloud consists of orientated or randomly orientated ice crystals. In the introduction you mention the sun glare. You could add discussion about the atmospheric implications of your findings. This will further underline the importance of your newly developed lidar system.

I can only say that both the shape of the crystalline particles and the presence of their orientation are determined by complex meteorological processes in the cloud. Therefore, the study of the orientation of crystalline particles provides significant information about these processes. However, the authors are not experts in the field of cloud physics. So I would not like to open a big discussion on this issue

*Revised text, Line 75.* The optical properties of ice clouds, including their effect on the radiation balance, are determined by both the microphysical properties of crystalline particles and the presence of their orientation. These properties are in turn determined by complex meteorological processes in the clouds. Therefore, the study of the orientation of crystalline particles provides significant information about these processes.

3. You give an exponential parameterization (equation 1). But the reader finds nowhere in the manuscript any parameters for this fit. You should definitely provide some fitting parameters for your curves (A, alpha_0, w).

It's not quite so.
*Line 317. (in the revised text)* For Fig. 10a $w$=42 arc minutes, for Fig. 10b $w$=82′.
*Line 330.* For all measurements the value $w$ is within 40-160 arc minutes.
Values $w$ are indicated in *Fig. 11* with squares. (Fig 11 and other figures is corrected)

*Line 330.* The shift $\alpha_0$ of the curve maximum from 0° is less than 2 minutes, (a symbol "$\alpha_0$" is added into the text).

The absolute values $A$ and $I_0$ are not interesting, because they are determined by the sensitivity of photodetectors. The ratio $I(0°)/I_0$ is indicated in Fig. 11.

4. To discuss the differences in the cirrus observations, it would be extremely helpful to provide some more information about the cirrus cloud. Firstly, the temperature profile within the cirrus. You show some radiosonde data in Fig. 7+8, but you don't use this information in the text. At which temperature do you observe the two cirrus clouds on 6 April 2018? At colder temperatures, the ice crystals may have different properties. To improve the Figures, I would show a temperature profile exactly for the same height range (6 – 10 and 7.5 – 12 km) as in Fig. 7a+b and 8a+b instead of the shown diagram. And please add the time of radiosonde launch.
Secondly, the different exponential behaviors in Fig. 11 are related to different cirrus clouds. What additional information do you have about these cirrus clouds? Cloud height? Cloud thickness? Cloud top temperature? Temperature profile within the cloud? Age of the cirrus cloud? Formation process? May this information help to explain the different behavior?
5. Where did you perform the measurements? In Tomsk. Can you add some coordinates?

How far was the radiosonde station?
6. How did you select the measurements in Fig. 11 (ln 280)? Which criteria did you use?

I agree that radiosonde data should be presented more clearly. But we do not know the meteorological parameters of each layer inside the cloud. It is because of the distant location of the station and the rare launches of probes (once every twelve hours). Clouds change their structure in 5-10 minutes. No radiosondes can provide such volatile information. Then, our previous observations (Balin 2011) showed that mirror layers may exist in both the lower and upper parts of the cloud. So the layer height inside the cloud also does not correlate with its polarization. The selection of cloud portions presented in the figure 11 is rather random and subjective. The only requirement is: these portions must have a pronounced dependence on the tilt angle, and there is no signal overflow when the lidar is oriented to zenith (Sect. 4.2). Today I don't know how to proceed and present the data (polarization) across all cloud height.

*Revised lines in the text*

*Line 252.* Measurements were made in Tomsk (56°28'N, 85°E)

*Line 270.* Radiozonde sounding was carried out at Novosibirsk station, about 250 km from Tomsk to SW, two records were made at 07:00 and 19:00 LST. Of course, due to the distant location of the station and the rare launches of probes (once every twelve hours) these data do not describe the fine structure of the ice cloud.

*Line 275.* A high-level cloud consists of two layers. The thin bottom layer (6600-7000 m) has a temperature minus 26-31°C, the top layer (7800-9800 m) temperature is 37-52°C below zero

*Line 287.* A cirrus cloud with a complex structure extends in a layer of 8 - 11.7 km. The temperature in the cloud varies from -34°C to -60°C.

*Line 325.* Figure 11 gives some selected dependences of intensity angle distributions for component $I_\parallel$. We did not find any correlation of the distribution parameters ($A$, $I_0$, $w$) with the height of the selected area inside the cloud. So the selection of cloud portions presented in Fig. 11 is rather subjective. The only requirement is: these portions must have a pronounced dependence on the tilt angle, and there is no signal overflow when the lidar is oriented to zenith.

7. The symbols â´Lˇe and âŁˇe correspond to parallel and orthogonal normally linked to linear polarization. Circular polarization is right handed or left handed or more general it can be described as co-polar and cross-polar. Or at least mark the intensity as a circular polarized component whenever it is used to not confuse the reader with the linear polarization.
In general, you should be more careful in distinguishing the linear and circular depolarization ratio throughout the text (often it is just stated "depolarization ratio").

I suppose that left-handed and right-handed circular polarizations are orthogonal, too. But I agree that the use of a symbol $I_\perp$ may be unreasonable for circular polarization. So I will use $I_{co}$ and $I_{cros}$ in the text and figures. Then the depolarization ratio is equal to $I_{cros}/I_{co}$ both for linear and circular polarization.

8. The paragraph line 286-292 describing the relation of circular and linear polarization should be placed earlier. The same holds for the information in line 300-303. Till these lines, it remained unclear how you deal with two wavelengths and a quarter wave plate. This has to be mentioned when describing Fig. 4.
Lines 300-303 are now in Sect.2. Lines 286-292 are shifted to the beginning of Sect. 4.1.
Revised text

*Line 111.* We have two sets of quarter-wave plates. One set is designed for 532 nm, another set for 1064 nm. For the wavelength of installed plates (below are the results only for installed 1064 nm plates) we can investigate both linear and circular polarizations. For the second wavelength (532 nm) polarization state when turning 45 degrees is not determined. However, in the position where the axis of the rotating phase plate coincides with the plane of polarization of the transmitter, the radiation remains linearly polarized for any wavelength. So the measurements for the wavelength of $\lambda$ = 532 nm were carried out only for linear polarization of radiation. Of

course, in our lidar we can use a quarter-wave plates for 532 nm if such experiments were planned.

9, 10 Figures and captions.
All figures and captions are updated

16. ln 89 "to evaluate some elements of BSPM" Which elements? Be more precise
*Line 94.* ...that makes it possible to detect the deviations of BSPM from diagonal shape (Balin et al., 2011).

18. ln 94 "PP1 with the phase shift of 20 wavelengths is used for _ = 532 nm, and 9.5 wavelengths for _ = 1064 nm." What do you mean by this?
*Line 98.* For coincidence of the polarization planes, the phase plate *PP1* with the phase shift of 20 wavelengths for λ = 532 nm and 9.5 wavelengths for λ = 1064 nm is installed. The rotation of this plate causes the rotation of the polarization plane for 1064 nm but does not affect the polarization of 532 nm.

21. ln 146 Where do you get this value from?
*Line 160.* Accuracy of installation angle is θ=34′=0.0125 rad. ... $\theta$ is a small setting angle error. Δδ=θ$^2$=0.000156≈0.016%

22. ln 148-150: 45_ * 0.68 ms /0.3_ = 102 ms Using the information you provided, the quarter wave plate would need 102 ms to turn by 45_. That would be too slow for a laser repetition rate of 10 Hz. Maybe you just have to report one more significant digit for time?

I think, 102 ms is not much different from 100 ms need for 10Hz repetition rate. I have already inserted next lines into the text.
*Line 133.* The rotation of the mirror obturator and platforms with phase plates is synchronized with the external trigger of the laser. So laser pulse frequency is about 10 pulse per second, but its exact value is determined by the obturator controller.

23. ln 157-158 Here it would be helpful to already mention Fig. 5. Otherwise, the number of steps seems somehow arbitrary
This is a very interesting offer.
24. Fig. 5a Why do you show this plot?
25. ln 159-164: The same procedure is done for plate B without plate A, isn't it?
*revised text,Line 175.* The rotation angle setting is monitored by the zero position sensors of platforms. When installing the plates in the platform frame, the plate axis can be shifted relative to the sensor at a certain angle, initially unknown. However, a laser pulse must be produced at the moment when the axis of the plates coincides with the reference plane of the lidar. The exact positions of the plates in the frame is set separately for plate A and B. We set one plate (e.g. plate B) in its channel (receiver for plate B) and turn on the rotation. A section of a homogeneous atmosphere with small aerosol content is selected. Figure 5a shows the lidar signals from two photodetectors ( $P_{co}$ and $P_{cros}$ ), summarized over all positions of plate B (red and green lines). A height range from 6 to 9 km was chosen, on which the depolarization ratio (blue line) is constant.

For each pulse, the rotation angle of the plate was recorded and the average value of the depolarization ratio over the range of 6–9 km was calculated. These values are shown in Fig. 5b (bottom frame). The dependence averaged over 30 minutes is shown in the upper frame in Fig. 5b. Minimal depolarization is observed at the 34th step of the platform. The accuracy of platform setting is ± 1 step, which corresponds to 0.03% error with respect to depolarization ratio. Similar adjustment of the plate A gives an exact position at the 45$^{th}$ platform step.

A timing diagram in Fig. 4b shows the position of the laser pulses relative to the zero position of the plates. 31 ms interval (45 steps) for plate A and 23 ms (34 steps) for plate B passes before

the first laser pulse. The situation repeats every 8 pulses. As already mentioned, the frequency of laser pulses is strictly synchronized with the rotation of the plates.

29. ln 220 The calibration was made 7 May 2017, the measurements are performed
one year later. Did you perform calibration measurements in 2018 as well? What can
you say about the stability of such calibration measurements?
Of course, Similar calibration procedures were carried out before measurements in April and June 2018. I will clarify the values of these constants.

32. Why do you study the range -1 to 4_ only? Many lidar systems are operated at an off-zenith angle at 5_. It would be interesting to extend your tilt angle up to 5_, even if the change from 4_ to 5_ will not be significant.
The most interesting is a scanning over 30° when the polarization from oriented plates changes abruptly. But due to the inhomogeneities of ice clouds scanning should be done in a short time. We chose a small scanning angle for this reason. I think in future measurements we will use a wider angle. May be the scanning from -5 to +5 will be used, because the symmetry around vertical position, I suppose, can indicate cloud uniformity. And certainly the scanning around 30° is interesting.

33. ln 250 "Values outside the vertical are close throughout the entire cloud thickness." – Close to what?
Values $\delta^{Circ}$ outside the vertical are close to 100% throughout the entire cloud thickness.

36. ln 273 How to determine I_0 (for alpha » 4_) if the scanning cycles are only done up to 4_?
Certainly, $I_0$ is obtained by fitting the function by the least squares method. But it is better to remove (a>>4) from the text.
*Revised: Line 315.* $I_0$ is the offset of dependence determined by a signal without the specular component

37. ln 278-279 Can you provide a mean and a standard deviation? Or maybe add it as a dashed line in Fig. 10.
47. Fig. 11 You just show some fitting results without showing the original data points. Can you underlay your fitting curves (in bold) with the data points in the corresponding color (in a light hue). Then, the reader will see the data used for these fits.
I think that the scatter of experimental data for single pulses well demonstrates approximation errors. I added data points to Fig. 11

39. ln 305 "thus, the amplitudes of signals are reduced to one value" – How?
*Line 345.* The relative sensitivity of photodetectors at 532 and 1064 nm was not calibrated. Therefore the intensities of polarization components for 532 nm (both $I_{co}^{Lin}$ and $I_{cros}^{Lin}$) were normalized so that the intensity maximum of $I_{co}^{Lin}$ (0°) in the vertical position coincided with $I_{co}^{Lin}$ (0°) for 1064 nm.

Thank you for your comments. We will take all your remarks into account in preparing the final text.

Sincerely,
Grigory Kokhanenko

---

## Author Comment (AC4) · 28 Oct 2019

Revised text after rev. 2 and 3

Please also note the supplement to this comment:
https://www.atmos-meas-tech-discuss.net/amt-2019-326/amt-2019-326-AC4-supplement.pdf

---

## Author Comment (AC5) · 28 Oct 2019

The authors are sincerely grateful for the careful reading of the work. We'll try to take into account all your comments when finalizing the text.

***Experiment***
1. Lines 114 f.: Please, provide more information about the fiber (polarization preserving?) and the shutter (coating?).
*revised text*
*Line 127.* The signal from the receiver of the near range through the optic fiber *Fb* is fed to the mirror shutter (aluminium coated obturator) *MS* , by means of which the signals of the near and far ranges are alternately switched. The small receiver is closer to the transmitter, so the transition to range-square mode starts earlier (80-100 m) than for the large receiver (800-900 m). During data processing signals from near range (50-1200 m) and far range (400 m-15 km) fit together at a distance of 800-900 m when range-square mode for large receiver starts. Of course, the silica optical fiber (1 mm diameter) destroys the polarization state of the signal, so near-range data cannot be used for polarization analysis.

2. How is background scattering suppressed? There seem to be no filters in the setup, is this correct?
Of course, we use interference filters before each detector. They are depicted in Fig. 2 as narrow rectangles, but were not mentioned in the text.
*Line 149.* Interference filters (*IF*) with bandwidth about 1 nm are placed in front of the detectors.

3. Lines 149-150: 150 Steps are required for a 45_-turn, which would take 102 ms (according to the information provided) and thus slightly longer than the time period between the 10-Hz laser pulses. Please, comment.
*Line 133* - The rotation of the mirror obturator and platforms with phase plates is synchronized with the external trigger of the laser. So laser pulse frequency is about 10 pulse per second, but its exact value is determined by the obturator controller.

4. Lines 159-160: 'Only one: : : channel'. Please, provide more details.
*Line 177* -  The exact positions of the plates is set separately for plate A and B. We set one plate (e.g. plate B) in its channel (receiver for plate B) and turn on the rotation.

***Measurements***
1. Lines 242 ff.: It is not obvious what is meant with 'double lines'.
*Line 283* - In Fig. 7a we saw clearly expressed single line of maximum signal, because the vertical position was the edge position when scanning. When we scan our lidar from 4° to -1° and back, vertical positions (0°) are close to each other. So in Fig. 8a two lines of maximum signals are close to each other and look like a double line.

2. Line 264: Figs. 7-9 present data from April and June, 2018. Then, suddenly, 1 October is mentioned. Please, provide earlier on in the section an overview of the measurements to be discussed.
It is a mistake. October 1 is not need here. Below we present the data obtained on 2 June.

3. Paragraph, lines 286 ff.: This information must be provided before the measurement are presented, because otherwise the interested reader is waiting for the linear depolarization ratios to be shown.
Thank you, I moved this lines to the beginning of Sect. 4.1

4. Paragraph, lines 300 ff.: This information definitely belongs to section 2 or 3!

This lines are moved to Sect. 2 and clarified

*Line 111.* We have two sets of quarter-wave plates. One set is designed for 532 nm, another set for 1064 nm. For the wavelength of installed plates (below are the results only for installed 1064 nm plates) we can investigate both linear and circular polarizations. For the second wavelength (532 nm) polarization state when turning 45 degrees is not determined. However, in the position where the axis of the rotating phase plate coincides with the plane of polarization of the transmitter, the radiation remains linearly polarized for any wavelength. So the measurements for the wavelength of $\lambda = 532$ nm were carried out only for linear polarization of radiation. Of course, in our lidar we can use a quarter-wave plates for 532 nm if such experiments were planned.

***Phrasing***

4. Lines 186-188: 'If: : : sounding.'

*Line 211.* The channel calibration problem is solved in most devices for polarization measurements. An exception is devices where signals of different polarization are sequentially directed to one photodetector (Platt, 1977; Eloranta and Piironen, 1994; McCullough et al., 2017).

6. Line 305: 'thus: : : value.'

*Line 345.* The relative sensitivity of photodetectors at 532 and 1064 nm was not calibrated. Therefore the intensities of polarization components for 532 nm (both $I_{co}^{Lin}$ and $I_{cros}^{Lin}$) were normalized so that the intensity maximum of $I_{co}^{Lin}$ (0°) in the vertical position coincided with $I_{co}^{Lin}$ (0°) for 1064 nm.

***Typos***

*3.* Line 66 'The authors', **full stop missing**

We did not quite understand what does this means, so I've replaced the sentence:

Line 67. However, we did not find references to such direct experiments in the literature.

***Figures.***

I substantially reworked all the figures and their captions in accordance with the comments of the reviewers.

References and typos.

Thank you for your comments. Your remarks were very helpful. We tried to take everything into account in the revised text.

Sincerely,
Grigory Kokhanenko

---

## Author Comment (AC6) · 28 Oct 2019

[revised manuscript text omitted]

Another effect caused by the horizontally oriented plates and columns is the corner reflection. It appears when the lidar is tilted at a significant angle. Depolarization of backscattered radiation has a maximum at a lidar tilt of 30 degrees for oriented plates and about 60 degrees for columns (Borovoi et al., 2000). A more detailed simulations show that when angular reflection is taken into account, the element $a_{44}$ of the light backscattering phase matrix (BSPM) is most informative in determining of the flutter (Konoshonkin, 2016). For the plates, an abrupt change in the $a_{44}$ occurs at a tilt angle of 30

degrees, while for the columns this element begins to grow smoothly at 30 degrees and reaches a maximum at 60° (Konoshonkin et al., 2016). Some experiments with tilted lidars were carried out (Del Guasta et al., 2006; Hayman et al., 2012, 2014; Neely et al., 2013, Veselovskii et al., 2017) and showed a high probability of the presence of oriented particles.

Particle orientations are promoted not only by aerodynamic forces, but also by forces of a different nature, such as wind shifts and electric fields. Kaul, 2000 supposed that in such conditions, crystalline particles can have a preferential orientation in the horizontal plane (azimuthal orientation). It is obvious that the direction of preferential orientation is connected with the direction of action of these forces. The basis for such conclusion is the observed non-invariance of the BSPM with respect to rotation of the coordinate system (Kaul, 2000; Kaul et al., 2004; Hayman et al., 2014; Samokhvalov et al., 2014). If we use linear polarized radiation and rotate the lidar around a vertical axis, signal energy depends on the angle of lidar rotation relative to the plane of preferential orientation of the particles if $m_{12} \neq 0$ (Kokhanenko et al., 2018). According to a large array of experimentally measured BSPM, the distribution of relative frequencies for $m_{12}$ shows that the value of $m_{12}$ lies in the interval [-0.6, -0.3] with the probability about 30% (Kaul et al., 2004; Balin et al., 2009). Therefore we would have to observe the modulation of the signal very often. However, we did not find references to such direct experiments in the literature.

It should be mentioned that most of the works used 532 nm wavelength for measurements. Due to technical difficulties, only a small number of works use the first harmonic of the Nd: YAG laser (1064 nm) that is optimal for recording aerosol layers (McGill et al., 2002; Burton et al., 2015; Haarig et al., 2016; Haarig et al., 2017; Veselovskii et al., 2017). The assumption of independence of crystalline particles scattering of the wavelength is not always justified (Tao et al., 2008; Vaughan et al., 2010; Borovoi et al., 2014). Therefore, comparison of polarization and amplitude (color ratio) of signals at two wavelengths can provide the additional information about properties of crystalline particles.

The optical properties of ice clouds, including their effect on the radiation balance, are determined by both the microphysical properties of crystalline particles and the presence of their orientation. These properties are in turn determined by complex meteorological processes in the clouds. Therefore, the study of the orientation of crystalline particles provides significant information about these processes. The article describes a scanning polarization lidar LOSA-M3, developed at the IAO SB RAS. The first results of studying the crystalline particles orientation by means of this lidar are given herein. The main purpose of this lidar is to study the optical characteristics of the mid- and high-level ice clouds at two wavelengths - 532 and 1064 nm. The design, optical scheme and principle of operation of the scanning polarization lidar are described in Sect. 2. Methods of instrument setup and calibration of the polarization channels are described in Sect. 3. Examples of observations of horizontally oriented particles in the cirrus clouds are given in Sect. 4.

**2 Lidar description**

Scanning polarizing lidar LOSA-M3 is a continuation of a series of small-size lidars LOSA-MS and LOSA-M2, developed and operated in the IAO SB RAS (Bairashin et al., 2005, 2011) in the Laboratory of Optical Sensing of the Atmosphere (LOSA). All of these lidars are intended primarily for use in field conditions, which impose certain restrictions on the weight-dimensional characteristics and the energy potential of the device. The main features of LOSA-M3 lidar are the following: 1) automatic scanning device, which allows to change the direction of sensing within the upper hemisphere at the speed of up to 1.5 degrees per second with the accuracy of angle measurement setting at least 1 arc minute; 2) separation of polarization components of the received radiation - $I_{co}$, which coincides with the original radiation, and the component $I_{cros}$ orthogonal to it, is carried out directly behind the receiving telescope, without installing the elements distorting polarization, such as dichroic mirrors and beamsplitters; and 3) continuous alternation of the initial polarization state (linear - circular) from pulse to pulse that makes it possible to detect the deviations of BSPM from diagonal shape (Balin et al., 2011).

A photograph of the lidar system is shown in Fig. 1.

The optical scheme of the lidar is shown in Fig. 2. The Q-Smart 850 (Quantel) laser with a fundamental harmonic energy of 850 mJ is used with the repetition frequency of 10 Hz (externally triggered). Radiation is linearly polarized, but the polarization plane of the second harmonic (532 nm) is rotated by 45° relative to the first. For coincidence of the polarization planes, the phase plate *PP1* with the phase shift of 20 wavelengths for $\lambda = 532$ nm and 9.5 wavelengths for $\lambda = 1064$ nm is installed. The rotation of this plate causes the rotation of the polarization plane for 1064 nm but does not affect the polarization of 532 nm. Turning the plate helps achieve the coincidence of the polarization planes for two harmonics. The Glan prism *GP* improves the polarization contrast of radiation. The quarter wave $\lambda/4$ phase plate *PP2* serves to transform the polarization state (linear-circular).

Rotation of the phase plates (the analogous plate *PP3* is placed in front of the analyzer) is performed by means of the rotating platform 8RU-M (Standa) in synchronism with the laser pulses frequency. Thus, the phase plate can rotate by 45° between pulses; at the same time, the state of polarization will consistently change from linear to circular and vice versa. Lidar signals are separately recorded for each position of the plates *PP2* and *PP3* and can be summed up (accumulated) in further processing for a certain period of time. Usually it takes from ten seconds to one minute. Thus, a synchronous rotation of two plates – *PP2* in the transmission channel and *PP3* in the receiver - allows measuring the polarization characteristics (e.g. depolarization ratio $\delta = I_{\perp}/I_{\parallel}$ ) simultaneously for both linear and circular polarizations.

We have two sets of quarter-wave plates. One set is designed for 532 nm, another set for 1064 nm. For the wavelength of installed plates (below are the results only for installed 1064 nm plates) we can investigate both linear and circular polarizations. For the second wavelength (532 nm) polarization state when turning 45 degrees is not determined. However, in the position where the axis of the rotating phase plate coincides with the plane of polarization of the transmitter, the radiation remains linearly polarized for any wavelength. So the measurements for the wavelength of $\lambda = 532$ nm were carried out only for linear polarization of radiation. Of course, in our lidar we can use a quarter-wave plates for 532 nm if such experiments were planned.

The beam is collimated by the 7-fold achromatic expander *BE*, designed at the IAO SB RAS (Kochanenko et al., 2012). Two receivers are used – an achromatic lens *AL* with the 40 mm diameter and 200 mm focus for the near range, and

Cassegrain mirror lens *CL* with the 200 mm diameter and 1000 mm focus for the far range. The iris diaphragms *FS1*, *FS2* in the focal plane of each lens determine the telescope field of view. A special feature of the Cassegrain lens design is installation of a video camera *VC* behind the secondary mirror, which is getting radiation through an annular diaphragm in the outer area of the secondary mirror (Simonova et al., 2015). The camera has a global shutter and is synchronized along with laser pulses. This camera setup allows observing the image of the laser spot on the objects around without parallax. It is especially important for setting the lidar field of view and for excluding the possibility of lidar orientation towards residential buildings.

The signal from the receiver of the near range through the optic fiber *Fb* is fed to the mirror shutter (aluminium coated obturator) *MS* , by means of which the signals of the near and far ranges are alternately switched. The small receiver is closer to the transmitter, so the transition to range-square mode starts earlier (80-100 m) than for the large receiver (800-900 m).

During data processing signals from near range (50-1200 m) and far range (400 m-15 km) fit together at a distance of 800-900 m when range-square mode for large receiver starts. Of course, the silica optical fiber (1 mm diameter) destroys the polarization state of the signal, so near-range data cannot be used for polarization analysis.

The rotation of the mirror obturator and platforms with phase plates is synchronized with the external trigger of the laser. So laser pulse frequency is about 10 pulse per second, but its exact value is determined by the obturator controller.

The lens *L* forms a quasi-parallel beam that enters through the phase plate *PP3* (similar to the plate *PP2*) to the Wollaston prism *WP*. The prism divides the beam into two components with orthogonal polarization, which are further divided along the wavelengths by dichroic beamsplitters. Unlike the schemes in which the wavelength division is carried out before separation of polarization components, there is no distortion of the polarization state when reflected from dichroic elements and there is no need to apply laborious calculations of the instrument vector and correction of the measured polarization (Di et al., 2016).

A beamsplitter *BS1* (Di-757, Semrock) is placed in the cross-polarization channel, transmitting radiation of 1064 nm and reflecting 532 nm. In the channel, corresponding to initial polarization, beamsplitters *BS2* (transmitting 1064 nm) and *BS3* (reflecting 532 nm and transmitting 607 nm) are installed. The radiation is detected by photodetectors in the analog mode: the avalanche photodiodes *APD1*, *APD2* for 1064 nm (C30956EH-TC Perkin Elmer, 3 mm diameter of the receiving area), photoelectric multipliers *PM1*, *PM2* (H11506 Hamamatsu) for 532 nm. Weak signals of Raman scattering at 607 nm are recorded with the photomultiplier *PM3* (H11706P Hamamatsu) in the photon counting mode. Part of 532-nm radiation is removed by the glass plate *BS4* on the photomultiplier *PM4* (H11706P), operating in the photon counting mode, which allows comparing the signals at 607 and 532 nm within one dynamic range. Interference filters (*IF*) with bandwidth about 1 nm are placed in front of the detectors.

The signals from photodetectors are processed by 12-bit ADCs LA-n12USB (Rudnev-Shilyaev) in case of analog signals or by a 200 MHz photon counter (IOA SB RAS) for Raman signals and input to the computer. Peculiarity of lidar is alternation of the transceiver parameters from pulse to pulse: changing of signals from the near and far ranges, the angle of the phase plates' rotation, and the rotation angle of the lidar during scanning. This eliminates a possibility of accumulating signals directly after digitizing. In our case, each signal is assigned a digital code, corresponding to the position of lidar elements, and the signals are sorted during computer processing.

**3 Tuning of lidar optical elements**

**3.1 Coinciding the polarization planes**

One of the main sources of errors in polarization measurements is a discrepancy between the polarization planes of the emitting and receiving channels. The Glan prism (*GP* in Fig. 2) is installed in a rotating frame and allows aligning the planes of emitter and receiver. Accuracy of installation angle is θ=34′=0.0125 rad. Figure 3 shows the measured value of the cross-polarized component (normalized to the minimum = 1), depending on the angle of the prism rotation. A signal from a uniform aerosol layer with a constant value of δ~1% was recorded with averaging in over 4.2-5 km range and 3 minutes interval.

Let us have the signal components $P_{\parallel}$ and $P_{\perp}$, and a true depolarization ratio $\delta = P_{\perp}/P_{\parallel}$. Signal at the cross-polarization receiver, when the prism position is inaccurate, will be $P'_{\perp} = P_{\perp} \cos^2 \theta + P_{\parallel} \sin^2 \theta \approx P_{\perp} + P_{\parallel}\theta^2$ ($\theta$ is a small setting angle error). Hence, the measured depolarization ratio will be $\delta' = P'_{\perp}/P_{\parallel} = \delta + \theta^2$, and the error of measured depolarization ratio (due to inaccurate prism position) is Δδ=θ²=0.000156≈0.016%.

**3.2 Phase plates setup**

Phase quarter-wave plates are mounted on a rotating platform driven by a stepper motor, working 1200 steps per revolution. One step of the platform is 0.3° and takes 0.68 ms of time, which allows the plate to rotate 45 degrees in the time period between two laser pulses (about 100 ms). During one revolution of the platform 8 laser shots are made. Positions of the plates A (transmitter) and B (receiver) for four consecutive pulses are shown in Fig. 4a. The bold line shows the direction of a fast axis of the plates. With such an arrangement of axes, the signals of the cross-polarized component with both linear and circular polarization are recorded by the same photodetector.

The rotation angle setting is monitored by the zero position sensors of platforms. When installing the plates in the platform frame, the plate axis can be shifted relative to the sensor at a certain angle, initially unknown. However, a laser pulse must be produced at the moment when the axis of the plates coincides with the reference plane of the lidar. The exact positions of the plates in the frame is set separately for plate A and B. We set one plate (e.g. plate B) in its channel (receiver for plate B) and turn on the rotation. A section of a homogeneous atmosphere with small aerosol content is selected. Figure 5a shows the lidar signals from two photodetectors ($P_{co}$ and $P_{cros}$), summarized over all positions of plate B (red and green lines). A height range from 6 to 9 km was chosen, on which the depolarization ratio (blue line) is constant.

For each pulse, the rotation angle of the plate was recorded and the average value of the depolarization ratio over the range of 6–9 km was calculated. These values are shown in Fig. 5b (bottom frame). The dependence averaged over 30 minutes is shown in the upper frame in Fig. 5b. Minimal depolarization is observed at the 34th step of the platform. The accuracy of platform setting is ± 1 step, which corresponds to 0.03% error with respect to depolarization ratio. Similar adjustment of the plate A gives an exact position at the 45[th] platform step.

    A timing diagram in Fig. 4b shows the position of the laser pulses relative to the zero position of the plates. 31 ms interval (45 steps) for plate A and 23 ms (34 steps) for plate B passes before the first laser pulse. The situation repeats every 8 pulses. As already mentioned, the frequency of laser pulses is strictly synchronized with the rotation of the plates.

**3.3 Calibration of the polarization channels**

    Measurements of the polarization of backscattered radiation require careful consideration of polarization distortions in the receiving paths and sensitivity of photodetectors in the channels of original and cross-polarization (Freudenthaler, 2016; Belegante et al., 2018; McCullough et al., 2018;). The task of observations in lidar networks is to monitor the optical and microphysical properties of aerosol, which requires restoring not only the backscattering coefficient, but also the lidar ratio and attenuation. Therefore, a large number of lidars are designed as aerosol-Raman (Althausen et al., 2000; Whiteman et al., 2007; Reichardt et al., 2012; Groß et al., 2015; Haarig et al., 2017; Madonna et al., 2018) or multiwave high spectral resolution lidars (HSRL) (Eloranta, 2005; Burton et al., 2015). Most of these lidars use dichroic beamsplitters as wavelength dividers, and polarizing elements (film polarizers) are installed after the beamsplitters and deflecting mirrors (Nott et al., 2012; Engelmann et al., 2016; McCullough et al., 2018). This leads to distortions in recording of polarization components, which require complex procedures for determining the eigenvectors of polarization of the lidar and taking them into account when restoring the polarization state of scattered radiation (Alvarez et al., 2006; Haymann et al., 2012; Freudenthaler et al., 2009; Bravo-Aranda et al., 2016; Di et al., 2016; Freudenthaler, 2016).

[revised manuscript text omitted]

**4 Observations of ice clouds during lidar zenith scanning**

In this paper we present the results of observations of the ice cloud cover under the lidar zenith scanning. Measurements were made in Tomsk (56°28'N, 85°E) in April-June 2018. Scanning was carried out at the rate of 0.5 degrees per second, which corresponds to three arc minutes shift of sensing direction between laser pulses. The angle is measured from the vertical position of radiation beam, which was set with an error of ± 5 arc minutes. Moreover, in some cases the scanning was carried out with the lidar axis passing through the zenith (ranging from -1 ° to +4°), which made it possible to control the accuracy of the lidar vertical setting. The measurement data array is not large and is mainly used here for testing the lidar performance.

**4.1 Zenith scanning at 1064 nm wavelength**

Below we present the data for polarized components at 1064 nm related to radiation with initial circular polarization. The depolarization of linearly polarized radiation is generally less than that of circularly polarized radiation. For randomly distributed particles, the linear depolarization is two times less than for circular (Mishchenko and Hovenier, 1995; Gimmestad, 2008). For oriented particles the difference is smaller, since the BSPM element $m_{12}$ is not equal to zero (Balin et al., 2011). Low value of depolarization results in that the cross-polarized component $I_{cros}^{Lin}$ very often does not stand out against the background of the photodetector noise in the specular reflection. Therefore, the dependences of polarized components for 1064 nm are presented for initial circular polarization of laser beam. Dependences $I_{co}(\alpha)$ for linear and circular polarizations for scan angles 0-4° do not differ within the error limits.

Figure 7 shows the lidar data observed on 6 April 2018. The recording starts at 10:25 local time (UTC+7), time from the start is shown on X-axis. Fig. 7a gives the component $I_{co}^{circ}$ (signal intensity, mV); 7b – the depolarization ratio $\delta^{Circ} = I_{cros}^{Circ}/I_{co}^{Circ}$, %; 7d – the zenith scanning angle (arc minutes), time axis is aligned with Fig. 7a and 7b. Data of weather sounding (7c) are taken from the site of the University of Wyoming (http://weather.uwyo.edu). Radiozonde sounding was carried out at Novosibirsk station, about 250 km from Tomsk to SW, two records were made at 07:00 and 19:00 LST. Of course, due to the distant location of the station and the rare launches of probes (once every twelve hours) these data do not describe the fine structure of the ice cloud.

Duration of recording is 300 seconds. In the interval from 120 to 250 seconds, scanning was carried out in the range from 0 to 2 degrees. A high-level cloud consists of two layers. The thin bottom layer (6600-7000 m) has a temperature minus 26-31°C, the top layer (7800-9800 m) temperature is 37-52°C below zero. The behavior of these layers is significantly different. Characteristics of the top layer do not change when scanning, which indicates the chaotic orientation of particles in the cloud. In the bottom layer a pronounced modulation of signal intensity and polarization is observed, characteristic for a predominantly horizontal orientation of particles. The maximum intensity of signal with the vertical sensing direction corresponds to the minimum of depolarization ratio. An extremely low value $\delta^{Circ}$ in the zenith direction $\delta^{Circ}_{Zen} \approx 4-5\%$ corresponds to the specular reflection.

The other situation is observed on 2 June 2018, 09:55 LST (Fig. 8). Duration of this recording is 550 seconds. The maximum inclination was 4°, the beam passed through the vertical by 1° while scanning. In Fig. 7a we saw clearly expressed single line of maximum signal, because the vertical position was the edge position when scanning. When we scan our lidar from 4° to -1° and back, vertical positions (0°) are close to each other. So in Fig. 8a two lines of maximum signals are close to each other and look like a double line.

A cirrus cloud with a complex structure extends in a layer of 8 - 11.7 km. The temperature in the cloud varies from -34°C to -60°C. Pronounced modulation of the signal intensity and depolarization ratio are observed throughout the entire height of the cloud. As in the previous case, the maximum intensity of the signal corresponds to the minimum of depolarization ratio. However, the minimum values $\delta^{Circ}_{Zen}$ differ significantly in various parts of the cloud. In the lower part of the cloud, the value $\delta^{Circ}_{Zen} \approx 0.6$ indicates the predominance of chaotically oriented particles and a small proportion of horizontally oriented particles. In the upper part (11.2-11.7 km) $\delta^{Circ}_{Zen} < 5\%$ is characteristic of mirror reflection and indicates a pronounced horizontal orientation of particles in this part of the cloud. Values $\delta^{Circ}$ outside the vertical are close to 100% throughout the entire cloud thickness. This suggests that particles in the cloud differ only in the degree of their horizontal orientation.

**4.2 Dependence of the signal intensity on the lidar tilt angle**

The ice clouds never constitute a formation uniform in height and constant in time. Pronounced layers with the thickness of hundreds of meters are regularly observed in the structure of ice clouds. They differ in the state of depolarization and signal intensity from the higher and lower regions and are supposedly homogeneous in composition and degree of particle orientation. However, the signal intensity and the depolarization ratio do not remain constant, but vary with time rather quickly. The height of layers also varies. Weak values of the backscatter signals lead to the need of averaging the signals over the height of the selected layer and over time about 3-5 minutes. As a result, it is necessary to pre-select sections of the cloud, characterized by an approximately constant value of intensity and depolarization ratio, and lasting for several scan cycles to measure the dependence of echo signal characteristics on the tilt angle. An example of such procedure for selecting the cloud sections to be studied is shown in Fig. 9.

In the given 5-minute recording, 6 sections were selected (marked with rectangles) with the duration from three to seven scan cycles. The rest of cloud portions were not analyzed on this record. In total, about 20 records were selected during observations on 2 June 2018, which have a pronounced dependence on the tilt angle, and there is no signal overflow when the lidar is oriented to zenith.

Figure 10 indicates the typical dependences of signal intensities of parallel $I_{co}^{circ}$ (circular initial polarization) and cross- polarized $I_{cros}^{circ}$ components on the tilt angle α. The record shown in Fig. 10a was registered at 12:00 LST with averaging of characteristics over the layer from 11470m to 11600m. The record shown in Fig. 10b was made at 10:00 LST with averaging of characteristics over the layer from 10980 to 11270m. The signal was accumulated during 10 minutes. These measurements and all data, obtained in June 2018, are described satisfactorily by the following exponential dependence

$$I(\alpha) = I_0 + A\exp(-|\alpha - \alpha_0|/w) \tag{1}$$

(red line in Fig. 10), where I is the signal intensity, $I_0$ is the offset of dependence determined by a signal without the specular component, $A$ is the constant, depending on the contribution of specular reflection in total intensity, α is the tilt angle, $w$ determines the width of distribution, and $\alpha_0$ indicates the error of lidar targeting to the vertical. For Fig. 10a $w$=42 arc minutes, for Fig. 10b $w$=82′. The Gaussian function used in Noel and Sassen, 2005 (blue line in Fig. 10) is noticeably wider and poorly describes the observed distribution, perhaps because it does not take into consideration the intensity peak at α=0°.

Cross-polarized signals $I_{cros}^{circ}$ have random variations from pulse to pulse comparable with the average value. In most cases the values of $I_{cros}^{circ}$ show a weak decline of intensity with the angle, but these variations do not exceed instrumental errors (about 1% for depolarization ratio). Figure 10a shows the variations even less the 0.5%. Moreover, a slightly noticeable maximum at α=0° looks like signal penetration from parallel to perpendicular channel. Therefore we think, that linear trend is not statistically reliable. We can conclude that $I_{cros}^{circ}$ is practically independent on the tilt angle.

Figure 11 gives some selected dependences of intensity angle distributions for component $I_{co}^{circ}$. We did not find any correlation of the distribution parameters ($A$, $I_0$, $w$) with the height of the selected area inside the cloud. So the selection of cloud portions presented in Fig. 11 is rather subjective. The only requirement is: these portions must have a pronounced dependence on the tilt angle, and there is no signal overflow when the lidar is oriented to zenith. The dependences are normalized to the value $I_0$ obtained by fitting according to the Eq. (1). Squares indicate the tilt angles, corresponding to the distribution width $w$. For all measurements the value $w$ is within 40-160 arc minutes. The shift $\alpha_0$ of the curve maximum from 0° is less than 2 minutes, which indicates a good lidar orientation to the zenith. Since the signal intensity offset $I_0$ is determined by particles with any orientation (both random and horizontal), the ratio $I_{co}^{Circ}(0°)/I_0$ may reflect the contribution of mirror particles to the lidar signal. The left frame shows cases of a fairly narrow distribution, $w$ is within 40- arc minutes. The ratio $I_{co}^{Circ}(4°)/I_0$ is less then 1.1. The right frame shows cases with $I_{co}^{Circ}(4°)/I_0 > 1.1$, among these cases there are distributions with a large (up to 159 arc min) width.

**4.3 Angular distributions for green and infrared wavelengths**

The obtained dependence $I_{co}(\alpha)$ in general terms should correspond to distribution of specularly reflecting particles along the angles of deviation from the horizontal plane (flutter). However, due to the phenomenon of backscattered radiation diffraction on the particle's contour (Borovoi et al., 2008), the distribution is wider than the distribution of particles along the angles of inclination. In addition, the ratio of backscattering coefficients (color ratio) for α >> 4° (and for randomly oriented particles in any direction) is not equal to unity (Tao et al., 2008; Vaughan et al., 2010; Borovoi et al., 2014). Therefore, the distribution $I_{co}(\alpha)$ may depend on the radiation wavelength.

As mentioned in the Sect. 2 the measurements for the wavelength of λ = 532 nm were carried out only for linear polarization of radiation, as we used the quarter-wave plates in the transceiver for the wavelength λ = 1064 nm. Two cases with the maximum angular differences are shown in Fig. 12. The relative sensitivity of photodetectors at 532 and 1064 nm was not calibrated. Therefore the intensities of polarization components for 532 nm (both $I_{co}^{Lin}$ and $I_{cros}^{Lin}$) were normalized so that the intensity maximum of $I_{co}^{Lin}(0°)$ in the vertical position coincided with $I_{co}^{Lin}(0°)$ for 1064 nm. In the left figure the angle distributions are the same, in the right figure the distribution for 532 nm is much wider ($I_{532}(4°)/I_{1064}(4°) = 1.5$). In all other cases we have intermediate states: the distribution width $w$ for 532 nm is equal or bigger than for 1064 nm. The reason for such behavior of dependences is not yet clear for the authors.

**5 Conclusions**

[revised manuscript text omitted]

Kokhanenko, G.P., Balin Yu.S., Borovoi A.G., Klemasheva M.G., Nasonov S.V., Novoselov M.M., Penner I.E., Samoilova S.V. "Investigations of the crystalline particle orientation in high-level clouds with a scanning lidar," ", Proc. SPIE 10833, 24th International Symposium on Atmospheric and Ocean Optics: Atmospheric Physics, 1083347 (13 December 2018); https://doi.org/10.1117/12.2504129

Konoshonkin, A.V., "Simulation of the scanning lidar signals for a cloud of monodisperse quasi-horizontal oriented particle," Optika Atmosfery i Okeana, 29, No. 12, 1053–1060 (2016) [in Russian].)

Konoshonkin, A., Wang, Zh., Borovoi, A., Kustova, N., Liu D., and Xie, Ch.: Backscatter by azimuthally oriented ice crystals of cirrus clouds, Opt. Express, 24(18), A1257-1268, 2016.

Lavigne, C., Roblin, A., and Chervet, P.: Solar glint from oriented crystals in cirrus clouds, Appl. Opt., 47(3), 6266–6276, 2008.

Liou, K. N.: Influence of cirrus clouds on weather and climate processes: a global perspective, J. Geophys. Res., 103, 1799–1805, 1986.

Madonna, F., Rosoldi, M., Lolli, S., Amato, F., Vande Hey, J., Dhillon, R., Zheng, Y., Brettle, M., and Pappalardo, G.:
Intercomparison of aerosol measurements performed with multi-wavelength Raman lidars, automatic lidars and ceilometers in the framework of INTERACT-II campaign, Atmos. Meas. Tech., 11, 2459–2475, 2018.

Masuda, K. and Ishimoto, H.: Influence of particle orientation on retrieving cirrus cloud properties by use of total and polarized reflectances from satellite measurements, J. Quant. Spectrosc. Radiat. Transfer, 85, 183–193, 2004.

McCullough, E. M., Sica, R. J., Drummond, J. R., Nott, G., Perro, C., Thackray, C. P., Hopper, J., Doyle, J., Duck, T. J., and
Walker, K. A., Depolarization calibration and measurements using the CANDAC Rayleigh–Mie–Raman lidar at Eureka, Canada, Atmos. Meas. Tech., 10, 4253–4277, 2017.

McCullough, E. M., Sica, R. J., Drummond, J. R., Nott, G. J., Perro, Ch., and Duck, T. J.: Three-channel single-wavelength lidar depolarization calibration, Atmos. Meas. Tech., 11, 861–879, 2018.

McGill, M. J., Hlavka, D. L., Hart, W. D., Spinhirne, J. D., Scott, V. S., and Schmid, B.: The Cloud Physics Lidar:
Instrument description and initial measurement results, Appl. Opt., 41, 3725–3734, 2002.

Mishchenko, M. I. and Hovenier, J. W.: Depolarization of light backscattered by randomly oriented nonspherical particles, Opt. Lett., 20, 1356–1358, 1995.

Neely, R. R., Hayman, M., Stillwell, R. A., Thayer, J. P., Hardesty, R. M., O'Neill, M., Shupe, M. D., and Alvarez, C.: Polarization Lidar at Summit, Greenland for the Detection of Cloud Phase and Particle Orientation, J. Atmos. Ocean.
Tech., 30, 1635–1655, 2013.

Noel, V. and Chepfer, H.: A global view of horizontally oriented crystals in ice clouds from Cloud-Aerosol Lidar and Infrared Pathfinder Satellite Observation (CALIPSO), J. Geophys. Res., 115, D00H23, 2010.

Noel, V. and Sassen, K.: Study of ice crystal orientation in ice clouds from scanning polarization lidar observations, J. Appl. Meteorol., 44(5), 653–664, 2005.

Noel, V., Chepfer, H., Ledanois, G., Delaval, A., and Flamant, P. H., Classification of particle effective shape ratios in cirrus clouds based on the lidar depolarization ratio, Appl. Opt., 41(21), 4245–4257, 2002.

Nott, G., Duck, T., Doyle, J., Coffin, M., Perro, C., Thackray, C., Drummond, J., Fogal, P., McCullough, E., and Sica, R.: A remotely operated lidar for aerosol, temperature, and water vapor profiling in the High Arctic, J. Atmos. Ocean. Tech., 29, 221–234, 2012.

Pal, S. R. and Carswell, A.I.: Polarization properties of lidar scattering from clouds at 347 nm and 694 nm, Appl. Opt., 17(15), 2321-2328, 1978.

Platt, C. M. R.: Lidar observations of a mixed-phase altostratus cloud, J. Appl. Meteorol., 16, 339–345, 1977.

Platt, C. M. R.: Lidar backscatter from horizontal ice crystal plates, J. Appl. Meteor., 17, 482–488, 1978.

Platt, C. M. R., Abshire, N. L., and McNice, G. T.: Some Microphysical Properties of an Ice Cloud from Lidar Observation
of Horizontally Oriented Crystals, J. Appl. Meteorol., 17(8), 1220–1224, 1978.

Reichardt, J., Wandinger, U., Klein, V., Mattis, I., Hilber, B., and Begbie, R.: RAMSES: German Meteorological Service autonomous Raman lidar for water vapor, temperature, aerosol, and cloud measurements, Appl. Opt., 51(34), 8111-8131, 2012.

Samokhvalov, I.V., Nasonov, S.V., Stykon, A.P., Bryukhanov, I.D., Borovoi, A.G., Volkov, S.N., Kustova, N.V., and
Konoshonkin, A. V., "Investigation of phase matrices of cirrus containing ensembles of oriented ice particles," Proc. SPIE 9292, 20th International Symposium on Atmospheric and Ocean Optics: Atmospheric Physics, 92922M (25 November 2014); doi: 10.1117/12.2075562

Sassen, K.: Ice crystal habit discrimination with the optical backscatter depolarization technique, J. Appl. Meteor., 16, 425–431, 1977.

Sassen, K.: Corona-produsing cirrus cloud properties derived from polarization lidar and photographic analyses, Appl. Opt., 30(24), 3421-3428, 1991.

Sassen, K. and Benson, S.: A midlatitude cirrus cloud climatology from the Facility for Atmospheric Remote Sensing: II. Microphysical properties derived from lidar depolarization, J. Atmos. Sci., 58(15), 2103–2112, 2001.

Sassen, K., Griffin, M. K., and Dodd, G. C.: Optical scattering and microphysical properties of subvisual cirrus clouds, and
climatic implications, J. Appl. Meteorol., 28(2), 91-98, 1989.

She, C.-Y.: Spectral structure of laser light scattering revisited: Bandwidths of nonresonant scattering lidars, Appl. Opt., 40, 4875–4884, 2001.

Simonova, G. V., Balin, Yu. S., Kokhanenko, G. P., Ponomarev, Yu. N., and Rynkov, O. A., Scanning multi-wave lidar for atmosphere objects sensing, Invention patent of Russian Federation, № 2593524, 2015.

Spinhirne, J. D., Hansen, M. Z., and Caudill, L. O. Cloud top remote sensing by airborne lidar, Appl. Opt., 21, 1564–1571, 1982.

Stillwell, R. A., Neely III, R. R., Thayer, J. P., Shupe, M. D., and Turner, D. D.: Improved cloud-phase determination of low-level liquid and mixed-phase clouds by enhanced polarimetric lidar, Atmos. Meas. Tech., 11, 835–859, 2018.

Strawbridge, K. B.: Developing a portable, autonomous aerosol backscatter lidar for network or remote operations, Atmos.
Meas. Tech., 6, 801–816, 2013.

Summa, D., Di Girolamo, P., Stelitano, D., and Cacciani, M.: Characterization of the planetary boundary layer height and structure by Raman lidar: comparison of different approaches, Atmos. Meas. Tech., 6, 3515–3525, 2013.

Thomas, L., Cartwright, J. C., and Wareing, D. P.: Lidar observation of the horizontal orientation of ice crystals in cirrus clouds, Tellus 42B, 211–216, 1990.

Tao, Z., McCormick, M. P., Wu, D., Liu, Z., and Vaughan, M. A.: Measurements of cirrus cloud backscatter color ratio with a two-wavelength lidar, Appl. Opt., 47(10), 1478-1485, 2008.

Vaughan, M. A., Liu, Z., McGill, M. J., Hu, Y., and Obland, M. D., On the spectral dependence of backscatter from cirrus clouds: Assessing CALIOP's 1064 nm calibration assumptions using cloud physics lidar measurements, J. Geophys. Res., 115, D14206, 2010.

Veselovskii, I., Goloub, P., Podvin, T., Tanre, D., Ansmann, A., Korenskiy, M., Borovoi, A., Hu, Q., and Whiteman, D.N. Spectral dependence of backscattering coefficient of mixed phase clouds over West Africa measured with two-wavelength Raman polarization lidar: Features attributed to ice-crystals corner reflection, J. Quant. Spectrosc. Radiat. Transf., 202, 74–80, 2017.

Whiteman, D. N., Veselovskii, I., Cadirola, M., Rush, K., Comer, J., Potter, J., and Tola, R.: Demonstration Measurements
of Water Vapor, Cirrus Clouds, and Carbon Dioxide Using a High-Performance Raman Lidar, J. Atmos. Ocean. Tech., 24, 1377–1388, 2007.

Volkov, S. N., Kaul, B. V., and Samokhvalov, I. V.: A technique for processing of lidar measurements of backscattering matrices, Atmos. Oceanic Opt., 15, 891–895, 2002.

Yoshida, R., Okamoto, H., Hagihara, Y., and Ishimoto, H.: Global analysis of cloud phase and ice crystal orientation from
Cloud-Aerosol Lidar and Infrared Pathfinder Satellite Observation (CALIPSO) data using attenuated backscattering and depolarization ratio, J. Geophys. Res., 115, D00H3, 2010.

You, Y., Kattawar, G. W., Yang, P., Hu, Y. X., and Baum, B. A.: Sensitivity of depolarized lidar signals to cloud and aerosol particle properties, J. Quant. Spectrosc. Radiat. Transf., 100(1-3), 470–482, 2006.

Young, A. T.: Revised depolarization corrections for atmospheric extinction, Appl. Opt., 19, 3427–3428, 1980.

[Figure]

**Figure 1. Photographs of the lidar on a rotary column in the laboratory room (left) and on the institute building roof (right).**

[Figure]

**Figure 2. Optical circuit of the LOSA-M3 lidar**

[Figure]

**Figure 3. Adjusting the position of the Glan prism. The dependence of the cross-polarized component $P_{cros}$ (normalized to the minimum = 1) on the angle θ of the prism rotation is shown. Measured values $P_{cros}$ (circles) and fitting curve (red line).**

[Figure]

Figure 4. (a) Diagram of the rotation of the phase plates. Plate mounting angles for the four consecutive laser pulses are shown. The bold line shows the direction of a fast axis of the plates. (b) The time position of the laser pulses relative to the zero positions of the plates. The situation repeats every 8 pulses.

[Figure]

**Figure 5. (a) Lidar signals from two photodetectors ( $P_{co}$ and $P_{cros}$ ), summarized over all positions of plate B (red and green lines).**

**A height range 6-9 km with constant depolarization ratio (blue line) is chosen to adjust the plates. (b) Depolarization ratio for each**

**pulse (bottom) and averaged over a 30 minute record (top).**

[Figure]

**Figure 6. Channel calibration procedure. (a) The record of a signal (1064 nm) integrated by the plate rotation angle, components** $I_{co}$ **(green) and** $I_{cros}$ **(red). Integration was carried out during four revolutions of the plate. (b) the value of calibration constant** *K,* **calculated in the height range of 1800-8000 m. The mean value** *K=1.91.*

[Figure]

**Figure 7. Lidar data on 6 April 2018, 10:25 LST (UTC+7) for the initial circular polarization. (a) Intensity of $I_{co}^{Circ}$ component; (b) depolarization ratio $\delta^{Circ} = I_{cros}^{Circ} / I_{co}^{Circ}$ ; (c) weather sounding data (Novosibirsk station, 07:00 and 19:00 LST); (d) zenith tilt angle (arc min), time axis is aligned with 7a and 7b.**

[Figure]

**Figure 8. Lidar data on 2 June 2018, 09:55 LST. Designations as in Fig. 7.**

[Figure]

**Figure 9. Selection of cloud sections for further processing. Recorded on 2 June 2018, 12:00-12:05 LST. Rectangles highlight areas which have a pronounced dependence on the tilt angle, and there is no signal overflow when the lidar is oriented to zenith.**

[Figure]

**Figure 10. Typical angular dependences of the intensity of polarization components, 2 June 2018. Top frames – co-polarized components $I_{co}^{Circ}$, bottom frames - $I_{cros}^{Circ}$. (a) 12:00 LST, averaging over the layer from 11470m to 11600m. (b) 10:00 LST, the layer 10980-11270 m. The record (a) shows low depolarization ratio and narrow ($w$=42 arc min.) distribution. The record (b) has a wider distribution ($w$=82 arc min.) and large depolarization**

[Figure]

**Figure 11.** Selected cases of observed distributions $I_{co}^{Circ}$. For all curves the values of intensity are normalized to the values $I_0$. obtained by fitting according to the Eq. (1). Squares indicate the tilt angles, corresponding to the distribution width $w$. The left frame shows cases of a fairly narrow distribution, $w$ is within 40-75 arc minutes. The ratio $I(4°)/I_0$ is less then 1.1. The right frame shows cases with $I(4°)/I_0 > 1.1$, among these cases there are distributions with a large (up to 159 arc min) width.

[Figure]

**Figure 12. Angular dependencies of polarization components for 532 nm (green) and 1064 nm (red). Two cases with the maximum angular differences are shown. In all other cases the distribution width *w* for 532 nm is equal or bigger than for 1064 nm.**

---

## Author Comment (AC7)

Response to Reviewer 4

Authors are grateful to the reviewer for a careful reading of the work and valuable comments

**1.** Provide a full description of the various symbols in Figure 2, in the caption of the figure.

I expanded the caption to the figure
**Figure 2. Optical circuit of the LOSA-M3 lidar. PP: phase plates; GP: Glan prism; BE: achromatic beam expander; AL: achromatic lens 40 mm diameter; CL: Cassegrain mirror lens *CL* 200 mm diameter; VC: video camera; FS1, FS2: iris diaphragms; Fb: optic fiber; MS: mirror shutter; L: lense; WP: Wollaston prism; BS: beamsplitters; APD: avalanche photodiodes; PMT: photomultiplier tubes.**

**2.** Provide a full description of the measurement sequence, in terms of the measurements at near and far zones, measurements at different wavelengths, measurements with linearly- and circularly-polarized emission (and corresponding detection), so the sequence of the measurements and their time resolution is clear. The use of a new figure to provide this sequence visually would help.
3. The system relies heavily on its rotating parts, but in the text there is not much information about their synchronization. Please provide your comments on this and/or the tests you performed to check for this.

I think that the addition of a new figure is difficult, since there are already 13 figures. Some information about the synchronization of laser pulses and plate rotation is shown in Figs. 4, 6. I have reworked some pieces of the text related to the obturator and phase plates where the alternation of near and far ranges is described.
*Line 134.* Shutter controller sets the obturator rotation frequency, the rotation speed of the phase plates and externally triggers the laser. So laser pulse frequency is about 10 pulse per second, but its exact value is synchronized with the rotation of the mirror obturator and platforms with phase plates.
There may be times when we are only interested in distant objects, such as high-level clouds. In this case, the obturator's rotation speed doubles, and the laser only starts when the shutter is open. The frequency of the laser pulses remains the same (10 Hz), but only far range signals are recorded.
*Line 195.* The diagrams indicated in Figs. 4, 6 refer to the cases of registration only the far range signals. If we register both near and far range signals, the plates rotate through the angle 22.5° between laser pulses. Intermediate positions correspond to the near range signals and have an undefined polarization.

Some more comments:
1. Make Fig. 4a and 4b two different figures. It is confusing to be in the same figure, because the first refers to the rotation of the phase plates and the second refers to the definition of their initial position.

This is a reasonable offer. Moreover, in the last version of the text (uploaded 28 Oct) I first refer to Fig. 4a, then Fig 5, and then 4b. Therefore now Fig. 4b will be Fig. 6.

2. Change caption of Fig. 5a to "lidar signal used to mount the plates at their initial position"

Now the caption of Fig. 5 is:
**Figure 5. (a) Lidar signals from two photodetectors ( $P_{co}$ and $P_{cros}$ ), summarized over all positions of plate B (red and green lines). A height range 6-9 km with constant depolarization ratio (blue line) is chosen to adjust the plates. (b) Depolarization ratio for each pulse (bottom) and averaged over a 30 minute record (top).**
I believe that the new caption takes into account the need to show the lidar signal in the figure 5a.

Sincerely
Grigory Kokhanenko

---

## Author Response (AR1)

**Comment 1**

Pg 2, Line 51. "Another effect caused by the horizontally oriented columns is the corner reflection when the lidar is tilted at 30 deg..." The 30 deg corner reflection comes from plates, not columns. Columns also have a corner reflection but it is

5 closer to 60degrees. This is noted in A. Borovoi, I. Grishin, E. Naats, and U. Oppel, "Backscattering peak of hexagonal ice columns and plates," Opt. Lett. 25(18), 1388–1390 (2000).

**Response**

Yes you are right. Depolarization of backscattered radiation has a maximum at a lidar tilt of 30 degrees for oriented plates and about 60 degrees for columns (Borovoi et al., 2000). A more detailed simulations show that when angular reflection is

10 taken into account, the element  $a_{44}$  of the BSPM is most informative in determining of the flutter (Konoshonkin, 2016). For the plates, an abrupt change in the  $a_{44}$  occurs at a tilt angle of 30 degrees, while for the columns this element begins to grow smoothly at 30 degrees and reaches a maximum at 60° (Konoshonkin et al., 2016). Reference to add:

Borovoi, A., Grishin, I., Naats, E., and Oppel, U., "Backscattering peak of hexagonal ice columns and plates," Opt. Lett., 25(18), 1388–1390 (2000).

**Revised** text**

15

Another effect caused by the horizontally oriented plates and columns is the corner reflection. It appears when the lidar is

- 20 tilted at a significant angle. Depolarization of backscattered radiation has a maximum at a lidar tilt of 30 degrees for oriented plates and about 60 degrees for columns (Borovoi et al., 2000). A more detailed simulations show that when angular reflection is taken into account, the element  $a_{44}$  of the light backscattering phase matrix (BSPM) is most informative in determining of the flutter (Konoshonkin, 2016). For the plates, an abrupt change in the  $a_{44}$  occurs at a tilt angle of 30 degrees, while for the columns this element begins to grow smoothly at 30 degrees and reaches a maximum at 60°
- 25 (Konoshonkin et al., 2016). Some experiments with tilted lidars were carried out (Del Guasta et al., 2006; Hayman et al., 2012, 2014; Neely et al., 2013, Veselovskii et al., 2017) and showed a high probability of the presence of oriented particles.

**Comment 2**

Pg 2, Line 54. In this paragraph the authors cite several works stating that these works observed both horizontal orientation

30 and azimuthal orientation. This is not true. Most of the references make no mention of observing azimuthally oriented ice crystals which implicitly seems to suggest they didn't observe any. There are a few works (such as Kaul 2004 and Balin 2011) that do mention observing this effect. Beyond that I happen to know number of the researchers cited are very skeptical about the existence of of azimuthally oriented ice crystals outside of thunderstorms. I doubt they would appreciate being

Konoshonkin, A.V., "Simulation of the scanning lidar signals for a cloud of monodisperse quasi-horizontal oriented particle," Optika Atmosfery i Okeana, 29, No. 12, 1053–1060 (2016) [in Russian].)

cited in support of this claim. The authors need to accurately represent the results of prior work and most of the citations 35 used here do not support the statement or even contradict the statement.

Response

Indeed I put this phrase very incorrectly. Sorry, I mistakenly cited some authors who did not mention the azimuthal orientation.

I reason like this. If azimuthal orientation takes place, the parameters of the backscattered radiation depend on the 40 orientation of the lidar reference plane relative to the direction of the preferential orientation of particles. *Direct measurements* of azimuthal orientation should be carried out as follows. We set lidar vertical and use linear polarized radiation. If the lidar reference plane coincides with the direction of the action of the orienting factor, the matrix of this cloud will have a block form

$$\mathbf{M} = \begin{vmatrix} m_{11} & m_{12} & 0 & 0 \\ m_{12} & m_{22} & 0 & 0 \\ 0 & 0 & m_{33} & m_{34} \\ 0 & 0 & -m_{34} & m_{44} \end{vmatrix}$$
, (Kaul, 2000). Then we rotate the lidar around a vertical axis. One can easily show

45 (Kokhanenko et al., 2018) that signal energy  $E^{lin} = m_{11} + m_{12} \cos 2\varphi$  depends on the angle of rotation of the lidar relative to the plane of preferential orientation of the particles in the cloud if  $m_{12} \neq 0$ . According to a large array of experimentally measured BSPM (Kaul et al., 2004), the average value of  $m_{12} = -0.22$ . The distributions of relative frequencies for  $m_{12}$  shows the probability that the value of  $m_{12}$  lies in the interval [-0.6, -0.3] is approximately equal to 30% (Kaul et al., 2004, Balin et al. 2009). Therefore we would have to observe the modulation of the signal very often.

50

The density of the cloud can change during the time of rotation. The signal from circular polarization serves as a reference for tracking changes in the signal that are not related to the rotation of the lidar. The function

$$F(\varphi) = \frac{E^{lin}(\varphi)}{E^{circ}(\varphi)} = 1 + \frac{m_{12}}{m_{11}} \cos 2\varphi \text{ varies with a period of 180° and one of the extremes of this dependence (max or min, this$$

depends on the sign of the element m12) coincides with the position of the plane of symmetry. However, authors are unaware of such direct measurements. Because our lidar (i) can scan around vertical axes, and (ii) measures both linear and circular
polarization, we can make observations using this technique. It is unfortunate, but all the measurements we carried out to date, have not shown the presence of azimuthal orientation. We plan to continue such observations in the future.

*Indirect evidence* of the existence of a preferential azimuthal orientation can be obtained from the form of measured BSPM. Instead of lidar rotation we can transform the matrix using the rotation matrix  $R(\phi)$  to the plane of symmetry. If the matrix may be represented in the block form, this suggests a high probability of the presence of a fraction of oriented

60 particles (Kaul 2000, Kaul et al. 2004, Samokhvalov et al. 2014). In a similar manner Hayman et al., 2014 simulated the rotation of measured matrix and made a conclusion about the orientation of the particles based on the change of the calculated depolarization ratio.

References to add:

Kokhanenko, G.P., Balin Yu.S., Borovoi A.G., Klemasheva M.G., Nasonov S.V., Novoselov M.M., Penner I.E., Samoilova

- S.V. "Investigations of the crystalline particle orientation in high-level clouds with a scanning lidar," ", Proc. SPIE 10833, 24th International Symposium on Atmospheric and Ocean Optics: Atmospheric Physics, 1083347 (13 December 2018); <a href="https://doi.org/10.1117/12.2504129">https://doi.org/10.1117/12.2504129</a>
  - Samokhvalov, I.V., Nasonov, S.V., Stykon, A.P., Bryukhanov, I.D., Borovoi, A.G., Volkov, S.N., Kustova, N.V., and Konoshonkin, A. V., "Investigation of phase matrices of cirrus containing ensembles of oriented ice particles," Proc. SPIE
- 70 9292, 20th International Symposium on Atmospheric and Ocean Optics: Atmospheric Physics, 92922M (25 November 2014); doi: 10.1117/12.2075562

**Revised** text**

Particle orientations are promoted not only by aerodynamic forces, but also by forces of a different nature, such as wind shifts and electric fields. Kaul 2000 supposed that in such conditions, crystalline particles can have a preferential orientation in the horizontal plane (azimuthal orientation). It is obvious that the direction of preferential orientation is connected with the

- in the horizontal plane (azimuthal orientation). It is obvious that the direction of preferential orientation is connected with the direction of action of these forces. The basis for such conclusion is the observed non-invariance of the BSPM with respect to rotation of the coordinate system (Kaul 2000, Kaul et al. 2004, Samokhvalov et al. 2014, Hayman et al., 2014). If we use linear polarized radiation and rotate the lidar around a vertical axis, signal energy depends on the angle of lidar rotation relative to the plane of preferential orientation of the particles if  $m_{12} \neq 0$  (Kokhanenko et al., 2018). According to a large
- array of experimentally measured BSPM, the distribution of relative frequencies for  $m_{12}$  shows that the value of  $m_{12}$  lies in the interval [-0.6, -0.3] with the probability about 30% (Kaul et al., 2004, Balin et al. 2009). Therefore we would have to observe the modulation of the signal very often. However, authors are unaware of such direct measurements.

**Discretionary Edits**

**85 *Comment 3*.**

... The authors should be careful about overasserting what their observations definitively prove about the scattering volume. . (Pg. 8 starting on line 235)... In that context, the authors later assert that the depolarization values are indicators for the relative mass of oriented and randomly oriented ice (Pg. 10 line 284). .....

**Response**

90 Your comments are very useful, and I will try to take it into account when finalizing the text. As for line 284, I try to revise this text

**Revised** text**

pg.10, Line 302

the ratio  $(I_{\parallel}(0^{\circ}) - I_{0})/I_{0}$  may reflect the contribution of mirror particles to the lidar signal.

**Comment 4.**

*Pg 11, Line 316 "… including exploring the azimuthal orientation of particles." I suggest being clear that this was not explored in the current work and that looking for azimuthal orientation of particles would be in future work.*

**Response**

100 I slightly changed the text

**Revised** text**

pg.11, Line 331

Since the circular polarization signal does not depend on the rotation of the lidar relative to the direction of particle orientation, this polarization can be used as a reference for investigation the azimuthal orientation of the particles.

105

**Comment 5.**

Pg 3, Line 62-64 It is not clear how the authors come to the conclusion that m12=-0.22 + / -0.2 means that in 30% of the observational cases, the depolarization depends on the lidar reference plane. The value of m12 certainly does not dictate this. Is the assumption that PDF of m12 is Gaussian?

110 Pg 3, Line 64 "In other words: : :" This statement isn't totally clear. I think to clarify you want to say "...when the lidar's linear polarization rotates around: : :"

**Response**

See the response to comment 2

**115 *Comment* 6.**

With regard to the near range channel, I'm a little confused about what its purpose is. Doesn't the fiber scramble the polarization modes and therefore prevent measurement of the depolarization ratio with this channel? What is this channel being used for? What ranges are the near and far range channels used for?

**Added text**

120 pg.4, Line 124.

The small receiver is closer to the transmitter, so the transition to range-square mode starts earlier (80-100 m) than for the large receiver (800-900 m). During data processing signals from near zone (50-1200 m) and far zone (400 m-15 km) fit together at a distance of 800-900 m when range-square mode for large receiver starts. Of course, the optic fiber destroys the polarization state of the signal, so near-zone data cannot be used for polarization analysis.

125

**Comment 7.**

*Pg. 10 Line 278. The authors state that the signal variations with angle are smaller than the measurement errors. This really depends on what the authors mean by "measurement errors" because one can clearly see a trend in the data, so the limiting*

factor does not appear to be random error. If they mean this is less than the systematic error of the instrument, this is certainly a valid point. It would be good to clarify which type of errors they are referring to.

Also with regard to Figure 10 and the perpendicular measurements, I wonder if this angle dependence is the result of cross talk between the channels. Perhaps this is what the authors are referring to as "measurement error". If so it would be helpful to simply state that explicitly.

**Response**

135 Analysis of measurements errors is very difficult. As for "cross talk", I suspected that this is so, but I did not know how to express it.

**Revised text**

pg.10, Line 293.

Cross-polarized signals  $I_{\perp}$  have random variations from pulse to pulse comparable with the average value. In most cases the

140 values of  $I_{\perp}$  show a weak decline of intensity with the angle, but these variations do not exceed instrumental errors (about 1% for depolarization ratio). Figure 10a shows the variations even less the 0.5%. Moreover, a slightly noticeable maximum at  $\alpha=0^{\circ}$  looks like signal penetration from parallel to perpendicular channel. Therefore we think, than linear trend is not statistically reliable. We can conclude that  $I_{\perp}$  is practically independent of the tilt angle.

**145 *Comment 8.**

Pg. 10, Line 300 The authors describe that the depolarization measurements at 532 nm are only made for linear polarizations. This needs to be better explained in section 2 Lidar Description. I had assumed (incorrectly) that the authors were using a dual wavelength wave plate. Make it clear what wavelengths the polarization optics are designed for and please be clear throughout the manuscript that this instrument performs the two polarization measurements only at 1064.

**150 Response**

The measurements for linear polarization can be made simultaneously for two wavelength 532 and 1064 nm. As for circular polarization, it is depend on the installed quarter-wave plates. Experiments described in the article were made with the plate, designed for 1064 nm. We can use a quarter-wave plate for 532 nm if it is planned. Similar we change a half-wave plate (532 or 1064nm) for calibration.

**155 Revised text**

pg.4, Line 107.

We have two sets of quarter-wave plates. One set is designed for 532 nm, another set for 1064 nm. For the wavelength of installed plates (below are the results only for installed 1064 nm plates) we can investigate both linear and circular polarizations. For the second wavelength (532 nm) polarization state when turning 45 degrees is not determined. However,

160 in the position where the axis of the rotating phase plate coincides with the plane of polarization of the transmitter, the radiation remains linearly polarized for any wavelength. So the measurements for the wavelength of  $\lambda = 532$  nm were carried

out only for linear polarization of radiation. Of course, in our lidar we can use a quarter-wave plates for 532 nm if such experiments were planned.

170

**Comment 1**

"Clouds of Upper Layers" in the title sounds strange. It would be better to replace it with "ice clouds" or "clouds in the upper troposphere". Why you mention "upper layers", you can as well observe clouds and ice clouds in lower parts of the troposphere. In general, I would omit the term "upper layers" throughout the whole manuscript. It can be replaced by "upper troposphere".

**Response**

I agree that the term "layer" is incorrect and very unfortunate. The terms "High-level" or "Mid-level clouds" are recommended by the International Cloud Atlas. I will try to use these terms in the manuscript and "ice clouds" in the title.

**175 *Comment 2**

I miss a bit the discussion about the atmospheric relevance of your findings. What does the additional information we get by scanning through the cirrus help us in characterizing cirrus clouds? It is interesting to know, if a cirrus cloud consists of orientated or randomly orientated ice crystals. In the introduction you mention the sun glare. You could add discussion about the atmospheric implications of your findings. This will further underline the importance of your newly developed

180 *lidar system*.

**Response**

I can only say that both the shape of the crystalline particles and the presence of their orientation are determined by complex meteorological processes in the cloud. Therefore, the study of the orientation of crystalline particles provides significant information about these processes. However, the authors are not experts in the field of cloud physics. So I would not like to

185 open a big discussion on this issue

*Revised text, Line 75.* The optical properties of ice clouds, including their effect on the radiation balance, are determined by both the microphysical properties of crystalline particles and the presence of their orientation. These properties are in turn determined by complex meteorological processes in the clouds. Therefore, the study of the orientation of crystalline particles provides significant information about these processes.

**190**

**Comment 3**

You give an exponential parameterization (equation 1). But the reader finds nowhere in the manuscript any parameters for this fit. You should definitely provide some fitting parameters for your curves (A, alpha\_0, w).

**Response**

**195 It's not quite so.**

*Line 317. (in the revised text)* For Fig. 10a *w*=42 arc minutes, for Fig. 10b *w*=82'. *Line 330.* For all measurements the value *w* is within 40-160 arc minutes.

Values w are indicated in *Fig. 11* with squares. (Fig 11 and other figures is corrected)

*Line 330.* The shift  $\alpha_0$  of the curve maximum from 0° is less than 2 minutes, (a symbol " $\alpha_0$ " is added into the text).

200 The absolute values A and  $I_0$  are not interesting, because they are determined by the sensitivity of photodetectors. The ratio  $I(0^\circ)/I_0$  is indicated in Fig. 11.

**Comments 4, 5, 6**

- 4. To discuss the differences in the cirrus observations, it would be extremely helpful to provide some more information about the cirrus cloud. Firstly, the temperature profile within the cirrus. You show some radiosonde data in Fig. 7+8, but you don't use this information in the text. At which temperature do you observe the two cirrus clouds on 6 April 2018? At colder temperatures, the ice crystals may have different properties. To improve the Figures, I would show a temperature profile exactly for the same height range (6 – 10 and 7.5 – 12 km) as in Fig. 7a+b and 8a+b instead of the shown diagram. And please add the time of radiosonde launch.
- 210 Secondly, the different exponential behaviors in Fig. 11 are related to different cirrus clouds. What additional information do you have about these cirrus clouds? Cloud height? Cloud thickness? Cloud top temperature? Temperature profile within the cloud? Age of the cirrus cloud? Formation process? May this information help to explain the different behavior?
  5. Where did you perform the measurements? In Tomsk. Can you add some coordinates? How far was the radiosonde station?
- 215 6. How did you select the measurements in Fig. 11 (ln 280)? Which criteria did you use?

**Response**

I agree that radiosonde data should be presented more clearly. But we do not know the meteorological parameters of each layer inside the cloud. It is because of the distant location of the station and the rare launches of probes (once every twelve

- 220 hours). Clouds change their structure in 5-10 minutes. No radiosondes can provide such volatile information. Then, our previous observations (Balin 2011) showed that mirror layers may exist in both the lower and upper parts of the cloud. So the layer height inside the cloud also does not correlate with its polarization. The selection of cloud portions presented in the figure 11 is rather random and subjective. The only requirement is: these portions must have a pronounced dependence on the tilt angle, and there is no signal overflow when the lidar is oriented to zenith (Sect. 4.2). Today I don't know how to
- 225 proceed and present the data (polarization) across all cloud height.

**Revised lines in the text**

*Line 252.* Measurements were made in Tomsk (56°28'N, 85°E)

*Line 270.* Radiozonde sounding was carried out at Novosibirsk station, about 250 km from Tomsk to SW, two records were made at 07:00 and 19:00 LST. Of course, due to the distant location of the station and the rare launches of probes (once

230 every twelve hours) these data do not describe the fine structure of the ice cloud.

*Line 275.* A high-level cloud consists of two layers. The thin bottom layer (6600-7000 m) has a temperature minus 26-31°C, the top layer (7800-9800 m) temperature is 37-52°C below zero

*Line 287.* A cirrus cloud with a complex structure extends in a layer of 8 - 11.7 km. The temperature in the cloud varies from  $-34^{\circ}$ C to  $-60^{\circ}$ C.

235 *Line 325.* Figure 11 gives some selected dependences of intensity angle distributions for component  $I_{\parallel}$ . We did not find any correlation of the distribution parameters (*A*,  $I_0$ , *w*) with the height of the selected area inside the cloud. So the selection of cloud portions presented in Fig. 11 is rather subjective. The only requirement is: these portions must have a pronounced dependence on the tilt angle, and there is no signal overflow when the lidar is oriented to zenith.

**240 *Comment* 7**

The symbols â'L'e and âL'e correspond to parallel and orthogonal normally linked to linear polarization. Circular polarization is right handed or left handed or more general it can be described as co-polar and cross-polar. Or at least mark the intensity as a circular polarized component whenever it is used to not confuse the reader with the linear polarization. In general, you should be more careful in distinguishing the linear and circular depolarization ratio throughout the text

245 *(often it is just stated "depolarization ratio").*

**Response**

I suppose that left-handed and right-handed circular polarizations are orthogonal, too. But I agree that the use of a symbol  $I_{\perp}$  may be unreasonable for circular polarization. So I will use  $I_{co}$  and  $I_{cros}$  in the text and figures. Then the depolarization ratio is equal to  $I_{cros}/I_{co}$  both for linear and circular polarization.

**250**

**Comment 8**

The paragraph line 286-292 describing the relation of circular and linear polarization should be placed earlier. The same holds for the information in line 300-303. Till these lines, it remained unclear how you deal with two wavelengths and a quarter wave plate. This has to be mentioned when describing Fig. 4.

**255 Response**

Lines 300-303 are now in Sect.2. Lines 286-292 are shifted to the beginning of Sect. 4.1.

**Revised text**

*Line 111.* We have two sets of quarter-wave plates. One set is designed for 532 nm, another set for 1064 nm. For the wavelength of installed plates (below are the results only for installed 1064 nm plates) we can investigate both linear and

260 circular polarizations. For the second wavelength (532 nm) polarization state when turning 45 degrees is not determined. However, in the position where the axis of the rotating phase plate coincides with the plane of polarization of the transmitter, the radiation remains linearly polarized for any wavelength. So the measurements for the wavelength of  $\lambda = 532$  nm were carried out only for linear polarization of radiation. Of course, in our lidar we can use a quarter-wave plates for 532 nm if such experiments were planned.

265

**Comments 9,10**

Figures and captions.

All figures and captions are updated

**270 Comment 16**

In 89 "to evaluate some elements of BSPM" Which elements? Be more precise

**Revised** text**

Line 94. ... that makes it possible to detect the deviations of BSPM from diagonal shape (Balin et al., 2011).

**275 Comment 18**

In 94 "PP1 with the phase shift of 20 wavelengths is used for  $\_ = 532$  nm, and 9.5 wavelengths for  $\_ = 1064$  nm." What do you mean by this?

**Revised** text**

*Line 98.* For coincidence of the polarization planes, the phase plate *PP1* with the phase shift of 20 wavelengths for  $\lambda = 532$

280 nm and 9.5 wavelengths for  $\lambda = 1064$  nm is installed. The rotation of this plate causes the rotation of the polarization plane for 1064 nm but does not affect the polarization of 532 nm.

**Comment 21**

In 146 Where do you get this value from?

**285 Revised text**

*Line 160.* Accuracy of installation angle is  $\theta=34'=0.0125$  rad. ...  $\theta$  is a small setting angle error.  $\Delta\delta=\theta^2=0.000156\approx0.016\%$

**Comment 22**

In 148-150: 45\_\* 0.68 ms /0.3\_ = 102 ms Using the information you provided, the quarter wave plate would need 102 ms to
turn by 45\_. That would be too slow for a laser repetition rate of 10 Hz. Maybe you just have to report one more significant digit for time?

**Response**

I think, 102 ms is not much different from 100 ms need for 10Hz repetition rate. I have already inserted next lines into the text.

*Line 133.* The rotation of the mirror obturator and platforms with phase plates is synchronized with the external trigger of the laser. So laser pulse frequency is about 10 pulse per second, but its exact value is determined by the obturator controller.

**Comment 23**

In 157-158 Here it would be helpful to already mention Fig. 5. Otherwise, the number of steps seems somehow arbitrary

**300 Response**

This is a very interesting offer.

**Comment 24,25**

24. Fig. 5a Why do you show this plot?

25. In 159-164: The same procedure is done for plate B without plate A, isn't it?

**305 Revised text**

*Line 175.* The rotation angle setting is monitored by the zero position sensors of platforms. When installing the plates in the platform frame, the plate axis can be shifted relative to the sensor at a certain angle, initially unknown. However, a laser pulse must be produced at the moment when the axis of the plates coincides with the reference plane of the lidar. The exact positions of the plates in the frame is set separately for plate A and B. We set one plate (e.g. plate B) in its channel (receiver

for plate B) and turn on the rotation. A section of a homogeneous atmosphere with small aerosol content is selected. Figure 5a shows the lidar signals from two photodetectors ( $P_{co}$  and  $P_{cros}$ ), summarized over all positions of plate B (red and green lines). A height range from 6 to 9 km was chosen, on which the depolarization ratio (blue line) is constant.

For each pulse, the rotation angle of the plate was recorded and the average value of the depolarization ratio over the range of 6–9 km was calculated. These values are shown in Fig. 5b (bottom frame). The dependence averaged over 30

315 minutes is shown in the upper frame in Fig. 5b. Minimal depolarization is observed at the 34th step of the platform. The accuracy of platform setting is  $\pm 1$  step, which corresponds to 0.03% error with respect to depolarization ratio. Similar adjustment of the plate A gives an exact position at the 45th platform step.

A timing diagram in Fig. 4b shows the position of the laser pulses relative to the zero position of the plates. 31 ms interval (45 steps) for plate A and 23 ms (34 steps) for plate B passes before the first laser pulse. The situation repeats every

320 8 pulses. As already mentioned, the frequency of laser pulses is strictly synchronized with the rotation of the plates.

**Comment 29**

In 220 The calibration was made 7 May 2017, the measurements are performed one year later. Did you perform calibration measurements in 2018 as well? What can

325 you say about the stability of such calibration measurements?

**Response**

Of course, Similar calibration procedures were carried out before measurements in April and June 2018. I will clarify the values of these constants.

**Revised** text**

Fig. 7b shows the value of calibration constant *K*, calculated in the height range of 1800-8000 m. In this case, the mean value K=1.91. Similar calibration procedures were carried out before each measurement in April and June 2018. All values *K* did not deviate by more than ±0.05.

**Comment 32**

335 Why do you study the range -1 to 4\_ only? Many lidar systems are operated at an off-zenith angle at 5\_. It would be interesting to extend your tilt angle up to 5\_, even if the change from 4\_ to 5\_ will not be significant.

**Response**

The most interesting is a scanning over  $30^{\circ}$  when the polarization from oriented plates changes abruptly. But due to the inhomogeneities of ice clouds scanning should be done in a short time. We chose a small scanning angle for this reason. I

340 think in future measurements we will use a wider angle. May be the scanning from -5 to +5 will be used, because the symmetry around vertical position, I suppose, can indicate cloud uniformity. And certainly the scanning around 30° is interesting.

**Comment 33**

345 In 250 "Values outside the vertical are close throughout the entire cloud thickness." – Close to what? Response

Values  $\delta^{Circ}$  outside the vertical are close to 100% throughout the entire cloud thickness.

**Comment 36**

350 ln 273 How to determine  $I_0$  (for alpha » 4) if the scanning cycles are only done up to 4?

**Response**

Certainly,  $I_0$  is obtained by fitting the function by the least squares method. But it is better to remove (a>>4) from the text.

**Revised text**

Line 315.  $I_0$  is the offset of dependence determined by a signal without the specular component

**355**

**Comment* 37,47**

37. In 278-279 Can you provide a mean and a standard deviation? Or maybe add it as a dashed line in Fig. 10.

47. Fig. 11 You just show some fitting results without showing the original data points. Can you underlay your fitting curves (in bold) with the data points in the corresponding color (in a light hue). Then, the reader will see the data used for these

**360 *fits*.**

**Response**

I think that the scatter of experimental data for single pulses well demonstrates approximation errors. I added data points to Fig. 11

**365 *Comment 39**

In 305 "thus, the amplitudes of signals are reduced to one value" – How?

**Revised** text**

*Line 345.* The relative sensitivity of photodetectors at 532 and 1064 nm was not calibrated. Therefore the intensities of polarization components for 532 nm (both  $I_{co}^{Lin}$  and  $I_{cros}^{Lin}$ ) were normalized so that the intensity maximum of  $I_{co}^{Lin}$  (0°) in the

370 vertical position coincided with  $I_{co}^{Lin}(0^{\circ})$  for 1064 nm.

**Experiment**

**375 *Comment 1**

*Lines 114 f.: Please, provide more information about the fiber (polarization preserving?) and the shutter (coating?).*

**Revised text**

Line 127. The signal from the receiver of the near range through the optic fiber Fb is fed to the mirror shutter (aluminium

380 coated obturator) MS, by means of which the signals of the near and far ranges are alternately switched. The small receiver is closer to the transmitter, so the transition to range-square mode starts earlier (80-100 m) than for the large receiver (800-900 m). During data processing signals from near range (50-1200 m) and far range (400 m-15 km) fit together at a distance of 800-900 m when range-square mode for large receiver starts. Of course, the silica optical fiber (1 mm diameter) destroys the polarization state of the signal, so near-range data cannot be used for polarization analysis.

**385**

**Comment 2**

How is background scattering suppressed? There seem to be no filters in the setup, is this correct?

**Response**

Of course, we use interference filters before each detector. They are depicted in Fig. 2 as narrow rectangles, but were not mentioned in the text.

**Revised** text**

Line 149. Interference filters (IF) with bandwidth about 1 nm are placed in front of the detectors.

**Comment 3**

395 Lines 149-150: 150 Steps are required for a 45\_-turn, which would take 102 ms (according to the information provided) and thus slightly longer than the time period between the 10-Hz laser pulses. Please, comment.

**Revised** text**

*Line 133* - The rotation of the mirror obturator and platforms with phase plates is synchronized with the external trigger of the laser. So laser pulse frequency is about 10 pulse per second, but its exact value is determined by the obturator controller.

**400**

**Comment 4**

Lines 159-160: 'Only one: : : channel'. Please, provide more details.

**Revised** text**

*Line 177* - The exact positions of the plates is set separately for plate A and B. We set one plate (e.g. plate B) in its channel (receiver for plate B) and turn on the rotation.

**Measurements**

**Comment 1**

Lines 242 ff.: It is not obvious what is meant with 'double lines'.

**410 Revised text**

*Line 283* - In Fig. 7a we saw clearly expressed single line of maximum signal, because the vertical position was the edge position when scanning. When we scan our lidar from  $4^{\circ}$  to  $-1^{\circ}$  and back, vertical positions ( $0^{\circ}$ ) are close to each other. So in Fig. 8a two lines of maximum signals are close to each other and look like a double line.

**415 *Comment 2**

*Line 264: Figs. 7-9 present data from April and June, 2018. Then, suddenly, 1 October is mentioned. Please, provide earlier on in the section an overview of the measurements to be discussed.*

**Response**

It is a mistake. October 1 is not need here. Below we present the data obtained on 2 June.

**420**

**Comment 3**

Paragraph, lines 286 ff.: This information must be provided before the measurement are presented, because otherwise the interested reader is waiting for the linear depolarization ratios to be shown.

**Response**

425 Thank you, I moved this lines to the beginning of Sect. 4.1

**Comment 4**

Paragraph, lines 300 ff.: This information definitely belongs to section 2 or 3!

**Response**

430 This lines are moved to Sect. 2 and clarified

**Revised** text**

*Line 111.* We have two sets of quarter-wave plates. One set is designed for 532 nm, another set for 1064 nm. For the wavelength of installed plates (below are the results only for installed 1064 nm plates) we can investigate both linear and circular polarizations. For the second wavelength (532 nm) polarization state when turning 45 degrees is not determined.

435 However, in the position where the axis of the rotating phase plate coincides with the plane of polarization of the transmitter, the radiation remains linearly polarized for any wavelength. So the measurements for the wavelength of  $\lambda = 532$  nm were carried out only for linear polarization of radiation. Of course, in our lidar we can use a quarter-wave plates for 532 nm if such experiments were planned.

**Phrasing**

**Comment 4**

Lines 186-188: 'If: : : sounding.'

**Revised** text**

Line 211. The channel calibration problem is solved in most devices for polarization measurements. An exception is devices

445 where signals of different polarization are sequentially directed to one photodetector (Platt, 1977; Eloranta and Piironen, 1994; McCullough et al., 2017).

**Comment 6**

Line 305: 'thus: : : value.'

**450 Revised text**

*Line 345.* The relative sensitivity of photodetectors at 532 and 1064 nm was not calibrated. Therefore the intensities of polarization components for 532 nm (both  $I_{co}^{Lin}$  and  $I_{cros}^{Lin}$ ) were normalized so that the intensity maximum of  $I_{co}^{Lin}$  (0°) in the vertical position coincided with  $I_{co}^{Lin}$  (0°) for 1064 nm.

**455**

**Typos**

**Comment 3**

Line 66 'The authors', full stop missing

**Response**

We did not quite understand what does this means, so I've replaced the sentence:

**460 Revised text**

Line 67. However, we did not find references to such direct experiments in the literature.

**Figures.**

I substantially reworked all the figures and their captions in accordance with the comments of the reviewers.

465

**References and typos.**

Thank you for your comments. Your remarks were very helpful. We tried to take everything into account in the revised text.

**Comment 1**

Provide a full description of the various symbols in Figure 2, in the caption of the figure.

**Response**

I expanded the caption to the figure

**475 *Revised text**

Figure 2. Optical circuit of the LOSA-M3 lidar. PP: phase plates; GP: Glan prism; BE: achromatic beam expander; AL: achromatic lens 40 mm diameter; CL: Cassegrain mirror lens *CL* 200 mm diameter; VC: video camera; FS1, FS2: iris diaphragms; Fb: optic fiber; MS: mirror shutter; L: lense; WP: Wollaston prism; BS: beamsplitters; APD: avalanche photodiodes; PMT: photomultiplier tubes.

**480**

**Comments 2,3**

2. Provide a full description of the measurement sequence, in terms of the measurements at near and far zones, measurements at different wavelengths, measurements with linearly- and circularly-polarized emission (and corresponding detection), so the sequence of the measurements and their time resolution is clear. The use of a new figure to provide this

485 sequence visually would help.

3. The system relies heavily on its rotating parts, but in the text there is not much information about their synchronization. Please provide your comments on this and/or the tests you performed to check for this.

**Response**

I think that the addition of a new figure is difficult, since there are already 13 figures. Some information about the synchronization of laser pulses and plate rotation is shown in Figs. 4, 6.

I have reworked some pieces of the text related to the obturator and phase plates where the alternation of near and far ranges is described.

**Revised** text**

Line 134. Shutter controller sets the obturator rotation frequency, the rotation speed of the phase plates and externally

495 triggers the laser. So laser pulse frequency is about 10 pulse per second, but its exact value is synchronized with the rotation of the mirror obturator and platforms with phase plates.

There may be times when we are only interested in distant objects, such as high-level clouds. In this case, the obturator's rotation speed doubles, and the laser only starts when the shutter is open. The frequency of the laser pulses remains the same (10 Hz), but only far range signals are recorded.

500 *Line 195.* The diagrams indicated in Figs. 4, 6 refer to the cases of registration only the far range signals. If we register both near and far range signals, the plates rotate through the angle 22.5° between laser pulses. Intermediate positions correspond to the near range signals and have an undefined polarization.

**Some more comments:**

**505 *Comment 1**

Make Fig. 4a and 4b two different figures. It is confusing to be in the same figure, because the first refers to the rotation of the phase plates and the second refers to the definition of their initial position.

**Response**

This is a reasonable offer. Moreover, in the last version of the text (uploaded 28 Oct) I first refer to Fig. 4a, then Fig 5, and 510 then 4b. Therefore now Fig. 4b will be Fig. 6.

**Comment 2**

Change caption of Fig. 5a to "lidar signal used to mount the plates at their initial position"

**Response**

515 Now the caption of Fig. 5 is:

**Revised** text**

Figure 5. (a) Lidar signals from two photodetectors ( $P_{co}$  and  $P_{cros}$ ), summarized over all positions of plate B (red and green lines). A height range 6-9 km with constant depolarization ratio (blue line) is chosen to adjust the plates. (b) Depolarization ratio for each pulse (bottom) and averaged over a 30 minute record (top).

520 I believe that the new caption takes into account the need to show the lidar signal in the figure 5a.

**Scanning polarization lidar LOSA-M3: opportunity for researchthe possibility of studying the crystalline particle orientation in the iceupper layers clouds**

Grigorii P. Kokhanenko, Yurii -S. Balin, Marina G. Klemasheva, Sergei V. Nasonov, Mikhail M. Novoselov, Iogannes E. Penner, Svetlana V. Samoilova

V.E.Zuev Institute of Atmospheric Optics, SB RAS, Tomsk, Russia

530 Correspondence to: Grigorii Kokhanenko (kokh@jao.ru)

> Abstract. The article describes a scanning polarization lidar LOSA-M3, developed at the V.E. Zuev Institute of Atmospheric Optics, the Siberian Branch of Russian Academy of Sciences (IAO SB RAS). The first results of studying investigation of the crystalline particles orientation by means of carried out with this lidar are presented herein. The main features of LOSA-M3 lidar are the following: 1) an- automatic scanning devicedrive, which allows to change the sensing

- 535 direction of sounding within the upper hemisphere at the a speed of up to 1.5 degrees degrees per second with the accuracy of the set angle measurement setting at least no worse than 1 arc angle minute; 2) - the separation of the polarization components of the received radiation is carried out directly behind the receiving telescope, without installing the elements distorting the polarization, such as of the elements - dichroic mirrors and beamsplitters; and 3)- continuous alternation from pulse to pulse of the initial polarization state (linear - circular) from pulse to pulse that which makes it possible to evaluate some elements of the scattering matrix.
- 540

525

For testing lidar performance several series of measurements of the iceupper layer crystalline cloud polarization structure in the zenith scan mode were carried out in Tomsk in April-JuneOctober 2018. The results show that the degree of horizontal orientation of the particles can vary significantly in different parts of the cloud. The dependence of the signal intensity on the tilt angle of inclination reflects the distribution of the particle deflection relative to the horizontal plane, and is well described by the exponential dependence. The values of cross-polarized component in most cases showshows a weak

545

decline of intensity with the angle. However, these variations, but its change are smaller than the measurement errors. We can conclude that it is practically independent of on the tiltinelination angle. In most cases the The scattering intensity at the wavelength of 532 nm has a wider distribution than at 1064 nm.

**1** Introduction**

550 Cirrus clouds cover a significant part of the earth's surface. Thus, they They have a significant impact on the radiation balance and climate, primarily due to the effects of radiation attenuation and reflection of radiation-(Liou, 1986; Sassen et al., 1989). In many cases, crystalline particles of cirrus clouds have a pronounced orientation in space. This leads to optical

anisotropy, manifested in various forms of the solar halo. Anisotropy affects the passage and reflection of radiation from clouds and, for example, leads to a dependence of the reflectivity on the zenith angle of the sun (Lavigne et al., 2008; Klotzsche and Macke. 2006: Lavigne et al., 2008).

555

The most well-known phenomenon is a the-predominant orientation of the crystals in the horizontal plane. It can be caused by aerodynamic forces arising from the free fall of particles (Kaul and Samokhvalov, 2005). Sections Sites with the a horizontal orientation of the particles manifestare manifested in the occurrence of sun glare when observing cloud cover from space (Chepfer et al., 1999; Masuda and Ishimoto, 2004). Analysis of the glare width of the glare shows the 560 correspondence to the Gaussian distribution of the crystal inclinationslopes with the a-half-width of about 0.4 degrees (Lavigne et al., 2008; Breon and Dubrulle, 2004; Lavigne et al., 2008). Presence). The presence of horizontally oriented crystals is also detected when observing specular spots from the a-spotlight on the clouds (Borovoi et al., 2008); flutter width is also estimated at 0.4°. Lidar However, lidar observations of the ice crystalline clouds provide the basic information about the particle orientation (Platt et al., 1978; Chen et al., 2002; Noel and Sassen, 2005). <del>Chen et al., 2002).</del>

565

Unlike water clouds, ice erystalline clouds cause greater depolarization of the backscattered radiation (Sassen and Benson, 2001). Most often, the depolarization ratio is within  $\delta = 0.3 - 0.6$  in the cloud areas of a cloud with randomly oriented particles, the depolarization ratio is within  $\delta = 0.3 \cdot 0.6$  (Noel et al., 2002; You et al., 2006). The magnitude of the depolarization is undoubtedly related to the shape of the particles and the phase composition of the cloud, which are is-taken into account when analyzing the observations of cirrus clouds (Noel et al., 2002; Hoareau et al., 2013; CampbellStillwell et 570 al., 20152018; Haarig et al., 2016; StillwellCampbell et al., 20182015).

Starting from the works of Sassen, 1977 and Platt et al., 1978 and Sassen, 1977, numerous observations show that with a the vertical orientation of the lidar, the horizontally oriented particles cause a specular reflection, manifested in the absence of depolarization and increased backscattering (Sassen and Benson, 2001; Sassen, 1991; Platt, 1978; Thomas et al., 1990; Sassen, 1991; Sassen and Benson, 2001). Data analysis of the polarization lidar in the CALIPSO experiment (Cho et al.,

- 575 2008; Noel and Chepfer, 2010; Yoshida et al., 2010) shows that a significant fraction of horizontally oriented particles in midmiddle latitudes is observed in the temperature range from -35°C to -5°C. Deviation of the lidar from the vertical position eliminates the effect of mirror reflection. For example, the CALIPSO lidar is deflected at 3° deviated from the nadir to eliminate this effect (Hunt et al., 2009). The angular dependence of the depolarization may vary for clouds with different temperatures (Sassen and Benson, 2001; Noel and Sassen, 2005; Sassen and Benson, 2001). According to the data from of these works, the dependence of the signal amplitude on the lidar tilt angle corresponds to the Gaussian distribution.
- 580

Another effect caused by the horizontally oriented plates and columns is the a-corner reflection. It appears when the lidar is tilted at a significant angle. Depolarization of backscattered radiation has a maximum at a lidar tilt of inclined 30 degrees for oriented plates and about 60 degrees for columns (Borovoi et al., 2000). A more detailed simulations show that when angular reflection is taken into account, the element  $a_{44}$  of the light backscattering phase matrix (BSPM) is most informative in determining of the fluttero from the zenith (Konoshonkin-et al., 2016). For the plates, an abrupt change in the  $a_{44}$  occurs

at a tilt angle of 30 degrees, while for the columns this element begins to grow smoothly at 30 degrees and reaches a maximum at 60° (Konoshonkin et al., 2016). Some experiments with tilted lidars were carried out (Experiments with tilt angles of 30 43° were carried out (Hunt et al., 2009; Del Guasta et al., 2006; Hayman et al., 2012, 2014; Neely et al., 2013; Veselovskii et al., 2017; Neely et 1., 2013) and showed a high probability of the presence of oriented particles. such an effect.

590

Particle orientations are promoted not only by aerodynamic forces, but also by forces of a different nature, such as wind shifts and electric fields. Kaul, 2000 supposed that in such conditions, crystalline particles can have. A number of works on polarization sounding of cirrus clouds (Chepfer et al., 1999; Noel and Sassen, 2005; Hayman et al., 2012; Kau et al., 2004; Balin et al., 2011; Borovoi et al., 2014; Hayman et al., 2014) showed that crystalline particles often demonstrate not only a preferential orientation relative to the horizon (zenith orientation) but also a preferential orientation in the horizontal plane

- 595 (azimuthal orientation). The probable reasons for this orientation are wind shifts and electric fields. It is obvious that the direction of preferential orientation is connected with the direction of action of these forces (Kaul, 2000). The basis for such a-conclusion is the observed non-invariance of the light backscattering phase matrix (BSPM) with respect to the rotation of the coordinate system (Kaul, 2000; Kaul et al., 2004; Hayman et al., 2014; Samokhvalov et al., 2014). If we use linear polarized radiation and rotate the lidar around or the cloud as a vertical axis, signal energy depends on the angle of lidar
- 600 rotation relative to the plane of preferential orientation of the particles if whole). For example, according to a large array of experimentally measured BSPM (Kaul et al., 2004) the zero value of the element  $m_{14}$  (Kokhanenko et al., 2018). According to a large array of experimentally measured BSPM, the distribution of relative frequencies for  $m_{12}$  shows that the value of  $m_{12}$  lies in the interval [-0.6, -0.3] with the probability about 30% (Kaul et al., 2004; Balin et al., 2009). Therefore we would have to observe the modulation of the signal very often. However, we did not find references to such direct experiments in 605 the literature is most likely,  $m_{14} = 0 \pm 0.05$ , whereas  $m_{12} = -0.22 \pm 0.2$ . This means that in 30% of cases the measured depolarization and amplitude of the signal depend on the orientation of the lidar reference plane relative to the direction of the preferential orientation of the particles. In other words, when a linear polarized lidar rotates around the vertical axis, the characteristics of the backscattered signal (amplitude, depolarization) should change cyclically. However, such direct
- 610

measurements are unknown to the authors.

It should be mentioned that in-most of the works used 532 nmthe polarization characteristics of the signal are measured at a wavelength for measurements. of 532 nm. Due to technical difficulties, only a small number of works use the first harmonic of the Nd: YAG laser (1064 nm) that is optimal for recording aerosol layers (Haarig et al., 2016; Veselovskii et al., 2017; McGill et al., 2002; Burton et al., 2015; Haarig et al., 2016; Haarig et al., 2017; Veselovskii et al., 2017). The assumption of independence of crystalline particles scattering of by crystalline particles on the wavelength is not always justified 615 (TaoBorovoi et al., 20082014; Vaughan et al., 2010; BorovoiTao et al., 20142008). Therefore, - a comparison of the

polarization and amplitude (color ratio) of signals at two wavelengths can provide the additional information about the properties of crystalline particles.

The optical properties of ice clouds, including their effect on the radiation balance, are determined by both the microphysical properties of crystalline particles and the presence of their orientation. These properties are in turn determined

620 by complex meteorological processes in the clouds. Therefore, The article describes a scanning polarization lidar LOSA-M3 developed at the IAO SB-RAS. The first results of investigation of the crystalline particles orientation carried out with this lidar are presented. The main purpose of this lidar is the study of the orientation of crystalline particles provides significant information about these processes. The article describes a scanning polarization lidar LOSA-M3, developed at the IAO SB RAS. The first results of studying the crystalline particles orientation by means of this lidar are given herein. The main

625

kAS. The first results of studying the crystalline particles of entation by means of this fidar are given herein. The main purpose of this lidar is to study the optical characteristics of the mid- and high-level ice clouds<del>crystal clouds of the upper and middle layers</del> at two wavelengths - 532 and 1064 nm. The design, optical scheme and principle of operation of the scanning polarization lidar are described in Sect. 2. Methods of instrument setup and calibration of the polarization channels are described in Sect. 3. Examples of observations of horizontally oriented particles in the cirrus clouds are given in Sect. 4.

**2 Lidar description**

- Scanning polarizing lidar LOSA-M3 is a continuation of a series of small-sizesized lidars LOSA-MS and LOSA-M2, developed and operated in the IAO SB RAS (Bairashin et al., 2005, 2011) in the -Laboratory of Optical Sensing of the Atmosphere –(LOSA). All of these lidars are intended primarily for use in fieldexpeditionary conditions, which imposeimposes certain restrictions on the weight-dimensional characteristics and the energy potential of the device. The main features of LOSA-M3 lidar are the following: 1) —automatic scanning devicedrive, which allows to change the direction of sensingsounding within the upper hemisphere at the a-speed of up to 1.5 degrees degree per second with the accuracy of the set-angle measurement setting at least no worse than-1 arc angle minute; 2) ) - the separation of the polarization components of the received radiation –,  $I_{\parallel}$ , which coincides with the original radiation, and the component  $I_{\perp}$ orthogonal to it, is carried out directly behind the receiving telescope, without installing the elements distorting the
- 640 impulse of the initial polarization state (linear circular) from pulse to pulse that which makes it possible to detect the deviationsevaluate some elements of BSPM from diagonal shape (Balin et al., 2011). A photograph. The appearance of the lidar system is shown in the Fig. 1.

The optical scheme of the lidar is shown in Fig. 2. The Q-Smart 850 (Quantel) laser with a fundamental harmonic energy of 850 mJ is used with, the repetition frequency of is-10 Hz (externally triggered). Radiation. The radiation is linearly polarized,

polarization, such as -of the elements -- dichroic mirrors and beamsplitters; and 3)- continuous alternation from impulse to

but the plane of polarization plane of the second harmonic (532 nm) is rotated by 45-° relative to the first. For coincidence of the To reconcile the polarization planes, the a-phase plate *PP1* with the having a-phase shift of 20 wavelengths for  $\lambda = 532$ nm and 9.5 wavelengths for  $\lambda = 1064$  nm is installed. The rotation of this plate causes the rotation of used. Turning the plate achieves the coincidence of the planes of polarization plane for 1064 nm but does not affect the polarization of 532 nm. Turning the plate helps achieve the coincidence of the polarization planes for two harmonics. The Glan prism *GP* improves 650 the polarization contrast of the radiation. The quarter wave  $\lambda/4$  phase plate *PP2* serves to transform the polarization state (linear-circular).

Rotation The rotation of the phase plates (the analogous plate *PP3* is placed in front of the analyzer) is performed by means of the a-rotating platform 8RU-M (Standa) in synchronism with the sending-laser pulses frequency. Thus, each subsequent pulse the phase plate can rotate by be rotated at 45° between pulses;°, at the same time, the state of polarization will consistently change from linear to circular and vice versa. Lidar Immediately, we note that the lidar signals are separately recorded for each position of the plates *PP2* and *PP3* and can be summed up (accumulated) in during-further processing for a certain period of time. Usually it takes, usually from ten seconds to one a-minute. Thus, a the synchronous rotation of two plates – *PP2* in the transmission channel and *PP3* in the receiver - allows measuringyou to measure the polarization characteristics (e.g. depolarization ratio- $\delta = I_{\perp}/I_{\parallel}$ ) simultaneously for both linear and circular polarizations.

660 This makes it possible to exclude the variability of elements of the scattering matrix during the observation time.

We have two sets of quarter-wave plates. One set is designed for 532 nm, another set for 1064 nm. For the wavelength of installed plates (below are the results only for installed 1064 nm plates) we can investigate both linear and circular polarizations. For the second wavelength (532 nm) polarization state when turning 45 degrees is not determined. However, in the position where the axis of the rotating phase plate coincides with the plane of polarization of the transmitter, the radiation remains linearly polarized for any wavelength. So the measurements for the wavelength of  $\lambda = 532$  nm were carried out only for linear polarization. Of course, in our lidar we can use a quarter-wave plates for 532 nm if such experiments were planned. A plate rotation algorithm is discussed more detailed below in Sect. 3.2.

- The beam is collimated by the 7-fold achromatic expander *BE*, designed at the IAO SB RAS (Kochanenko et al., 2012). Two receivers are used an achromatic lens *AL* with the 40 mm diameter and 200 mm focus for the near range, and
  Cassegrain mirror lens *CL* with the 200 mm diameter and 1000 mm focus for the far range. The iris diaphragms *FS1*, *FS2* in the focal plane of each lens determine the telescope field of view. A special feature of the Cassegrain lens design is installation of a video camera *VC* behind the secondary mirror, which is getting radiation through an annular diaphragm in the outer area of the secondary mirror (Simonova et al., 2015). The camera has a global shutter and is synchronized along with laser pulses. This camera setup allows observing the image of the laser spot on the objects around without parallax.
  It The beam is collimated by a 7-fold achromatic expander *BE*, developed in the IAO SB RAS (Kochanenko-et al., 2012). Two receivers are used an achromatic lens *AL* with a diameter of 40 mm and a focus of 200 mm for the near zone, and Cassegrain mirror lens *CL* with a diameter of 200 mm and a focus of 1000 mm for the far zone. Iris diaphragms *FS1*, *FS2* in the focal plane of each lens determine the field of view of telescopes. A special feature of Cassegrain lens design is the installation of the video camera *VC* behind the secondary mirror, which is emitted through an annular diaphragm in the outer
- 680

655

area of the secondary mirror (Simonova et al., 2015). The camera has a global shutter and is synchronized by laser pulses. This camera setup allows us to observe without parallax the image of the radiation spot at the object to which the laser

radiation is directed. This is especially important for settingtuning the lidar field of view and for excludingto exclude the possibility of lidar orientation towards residential buildings.

The signal from the receiver of the near range<del>zone</del> through the optic fiber *Fb* is fed to the mirror shutter (aluminium coated 685 obturator) MS, by means of which the signals of the near and far ranges<del>zones</del> are alternately switched. The near range radiation is reflected from the mirror obturator, the far zone radiation passes directly with the open position of the obturator. The small receiver is closer to the transmitter, so the transition to range-square mode starts earlier (80-100 m) than for the large receiver (800-900 m). During data processing signals from near range (50-1200 m) and far range (400 m-15 km) fit together at a distance of 800-900 m when range-square mode for large receiver starts. Of course, the silica optical fiber (1 690 mm diameter) destroys the polarization state of the signal, so near-range data cannot be used for polarization analysis.

Shutter controller sets the obturator rotation frequency, the rotation speed of the phase plates and externally triggers the laser. So laser pulse frequency is about 10 pulse per second, but its exact value is synchronized with the rotation of the mirror obturator and platforms with phase plates.

There may be times when we are only interested in distant objects, such as high-level clouds. In this case, the obturator's 695 rotation speed doubles, and the laser only starts when the shutter is open. The frequency of the laser pulses remains the same (10 Hz), but only far range signals are recorded.

The lens L forms a quasi-parallel beam that enters through the phase plate PP3 (similar to the plate PP2) to the Wollaston prism WP. The prism divides the beam into two components with orthogonal polarization, which are further divided along the wavelengths by dichroic beamsplitters. Unlike the schemes in which the wavelength division is carried out before the 700 separation of polarization components, there is no distortion of the polarization state when reflected from dichroic elements and there is no need to apply laborious calculations of the instrument vector and correction of the measured polarization (Di et al., 2016).

AIn a beam corresponding to cross polarization, a beamsplitter BS1 (Di-757, Semrock) is placed in the cross-polarization channel, transmitting radiation of 1064 nm and reflecting 532 nm. In the channel, corresponding to the initial polarization, 705 beamsplitters BS2 (transmittingtransmits 1064 nm) and BS3 (reflectingreflects 532 nm and transmittingtransmits 607 -nm) are installed. The radiation is detected by photodetectors in the analog mode: the avalanche photodiodes APD1, APD2 for 1064 nm (C30956EH-TC Perkin Elmer, 3 mm diameter of the receiving area-3 mm), photoelectric multipliers *PM1*, *PM2* (H11506 Hamamatsu) for 532 nm. Weak signals of Raman scattering at 607 nm are recorded in the photon counting mode with the photomultiplier PM3 (H11706P Hamamatsu) in the photon counting mode. Part of ). A part of the 532-nm radiation 710 is removed by the glass plate BS4 on the photomultiplier PM4 (H11706P), operating which operates in the photon counting mode, which allows comparing the signals at 607 and 532 nm within one dynamic range. Interference filters (IF) with bandwidth about 1 nm are placed in front of the detectors.

The signals from the photodetectors are processed by 12-bit ADCs LA-n12USB (Rudnev-Shilyaev) in the case of analogeurrent signals or by a 200 MHzMeps photon counter (IOA SB RAS) for Raman signals and input to the computer.

715 Peculiarity of lidar Lidar's feature is the alternation of the transceiver transceiver's parameters from pulse to pulse:

changing the change of signals from the near and far rangeszones, the angle of rotation of the phase plates' rotation, and the rotation angle of the lidar during scanning. as a whole. This eliminates a the possibility of accumulating signals directly after digitizing. In our case, each signal is assigned A scheme has been adopted in which a digital code, corresponding to the position of the-lidar elements, is assigned to each signal in digital form and the signals are sorted already during computer processing.

720

**3 Tuning of lidar optical elements**

**3.1 Coinciding the polarization Polarization planes**

One of the main sources of errors in polarization measurements is a the-discrepancy between the polarization planes of the emitting and receiving channels. The -Glan prism (GP in Fig. 2) is installed in a rotating frame and allows aligning you to 725 align the planes of the emitter and receiver. Accuracy The accuracy of the installation angle is  $\theta=34'=0.0125$  rad. Figure 3 shows the measured value of the cross-polarized component (normalized to the minimum = =1), depending on the angle of the prism rotation-of the head. A signal from a uniform aerosol layer with a constant value of  $\delta \sim 1\%$  was recorded with averaging in over 4.2-5 km range and 3 minutes interval.

Let us have the signal components  $P_{\parallel}$  and  $P_{\perp}$ , and a the-true depolarization ratio  $\delta = P_{\perp}/P_{\parallel}$ . Signal at the cross-

**polarization receiver, when the prism position is inaccurate, will be $P'_{\perp} = P_{\perp} \cos^2 \alpha + P_{\parallel} \sin^2 \alpha \approx P_{\perp} + P_{\parallel} \alpha^2$ ( $\theta \alpha$ is a 730**

small settingan installation angle error). Hence, the measured depolarization ratio will be  $\delta' = \frac{P'_{\perp}}{P_{\parallel}} = \delta + \alpha^2$ , and the error of measured depolarization ratio (due to inaccurate prism position) is  $\Delta \delta = \theta^2 e^2 = 0.000156 \approx 0.016\%$ .

**3.2 Phase plates setup**

Phase quarter-wave plates are mounted on a rotating platform driven by a stepper motor, working 1200 steps per revolution. One step of the platform is  $0.3^{\circ}$  and takes 0.68 ms of time, which allows the plate to rotate 45 degrees induring the time 735 period between two laser pulses (about 100 ms). During For-one revolution of the platform 8 laser shots are made. Positions The positions of the plates A (transmitter) and B (receiver) for four consecutive pulses are shown in Fig. 4a. The bold line shows the direction of a the-fast axis of the plates. With such an arrangement of the axes, the signals of the crosspolarized component with both linear and circular polarization are recorded by the same photodetector.

The rotation angle setting is monitored by the zero position sensors of the platforms. When installing the plates in the platform frame, of the plate platform, the axis of the plate can be shifted relative to the sensor atby a certain angle, initially unknown. However, a laser pulse must be produced at the moment when the axis of the plates coincides with the reference plane of the lidar. The exact positions of the plates in the frame is set separately for plate A and B. We set Fig. 4b shows a

740

situation where, for plates A and B, from the instant of triggering the sensors to the laser pulse, passes 31 ms (45 steps) and

745 23 ms (34 steps), respectively. The situation repeats every 8 pulses.

To install the plates, a part of a homogeneous atmosphere with a small aerosol content is selected. Only one plate (e.g. plate B) in its is inserted into the channel (receiver for plate B) and turn on the rotation. A section of a homogeneous atmosphere with small aerosol content is selected. Figure 5a shows the lidar signals from two photodetectors ( $P_{co}$  and  $P_{cros}$ ), summarized over all positions of plate B (red and green lines). A height range -A section-from 6 to 9 km was chosen, on which the depolarization ratio (blue line) is constant.

750

755

For each pulse, the rotation angle of the plate was recorded and the average value of the depolarization ratio over the range of 6–9 km was calculated. These values are shown in Fig. 5b (bottom frame). The dependence averaged over 30 minutes is shown in the upper frame in Fig. 5b. Fig. 5b shows the depolarization ratio for each pulse (below) and averaged over a 30 minute record (top). Minimal depolarization is observed at the 34th step of the platform. The accuracy of platform setting installation accuracy is  $\pm 1$  step, which corresponds to 0.03% an error of 0.03% with respect to depolarization ratio. Similar adjustment of the plate A gives an exact position at the 45th platform step.

A timing diagram in Fig. 6 shows the position of the laser pulses relative to the zero position of the plates. 31 ms interval (45 steps) for plate A and 23 ms (34 steps) for plate B passes before the first laser pulse. The situation repeats every 8 pulses. As already mentioned, the frequency of laser pulses is strictly synchronized with the rotation of the plates.

760

The diagrams indicated in Figs. 4, 6 refer to the cases of registration only the far range signals. If we register both near and far range signals, the plates rotate through the angle 22.5° between laser pulses. Intermediate positions correspond to the near range signals and have an undefined polarization.

**3.3 Calibration of the polarization channels**

Measurements of the polarization of backscattered radiation require careful consideration of polarization distortions in the receiving paths and the sensitivity of the photodetectors in the channels of the original and cross-polarization (Freudenthaler, 2016; Belegante et al., 2018; McCullough et al., 2018;; Freudenthaler,2016). The task of observations in lidar networks is to monitor the optical and microphysical properties of the aerosol, for which requires restoringit is necessary to restore not only the backscattering coefficient, but also the lidar ratio and attenuation. Therefore, a large number of lidars are designedereated as aerosol-Raman (Althausen et al., 2000; Whiteman et al., 2007; Reichardt et al., 2012; Groß et al., 2015; Haarig et al., 2017; Madonna et al., 2018; Whiteman et al., 2007; Groß et al., 2015) or multiwave high spectral resolution (HSRL) lidars (HSRL) (Eloranta, 2005; Burton et al., 2015; Haarig et al., 2017; Althausen et al., 2000; Eloranta, 2005). Most of these lidars use dichroic beamsplitters as wavelength dividers, and polarizing elements (film polarizers) are installed after the beambeam splitters and deflecting mirrors (NottMeCullough et al., 20122018; Engelmann et al., 2016; MeCulloughNott et al., 2018; 2018; 2012). This leads to distortions in recordingthe registration of polarization components, which require complex

775 procedures for determining the eigenvectors of polarization of the lidar and taking them into account when restoring the

polarization state of the scattered radiation (Alvarez et al., 2006; Haymann et al., 2012; Di et al., 2016; Freudenthaler, 2016; Freudenthaler et al., 2009; Bravo-Aranda et al., 2016; DiAlvares et al., 2016; Freudenthaler, 20162006).

Measurements of the polarization characteristics of backscattered radiation simultaneously for several wavelengths were carried out by different groups: 347+697 nm (Pal and Carswell, 1978), 355+532 nm (Groß et al., 2015; Althausen et al.,

780

- 785 therefore, there is no need to use laborious calculations for<del>of</del> the instrumental vector of the transmitting-receiving channel and correction of correcting the measured polarization. In this case, to measure the magnitude of the depolarization ratio, it is sufficient to determine the relative sensitivity of the detectors in both lidar channels to measure the magnitude of the depolarization ratio. The channel calibration problem is solved in most devices for polarization measurements. An exception is devices where signals lidar. If we ignore the devices that used a sequential change of different polarization are sequentially 790 directed to polarizations on one photodetector (Platt, 1977; Eloranta and Piironen, 1994; McCullough et al., 2017)), this
  - problem was solved in all devices for polarization sounding.

The relative sensitivity of photodetectors can be determined by observing the a-source with a known polarization — it this can be a non-polarized source (Sassen and Benson, 2001), or an atmospheric layer with purely molecular scattering (Biele et al., 2000Noel and Sassen, 2005; Noel et al., 2002; Volkov et al., 2002; Kaul et al., 2004; Noel and Sassen, 795 2005<del>2004; Biele et al., 2000; Volkov et al., 2002</del>). However, the depolarization ratio for in-molecular scattering depends significantly depends on the bandwidth of the used interference filter used, since rotational Raman lines can contribute to the signal (She, 2001; Young, 1980; She, 2001). For pure Rayleigh scattering, the depolarization ratio is  $\delta$ =0.00365, with a wide filter,  $\delta$ =0.015. It This leads to an ambiguous lidar calibration.

The most common calibration is associated with the rotation of the separation plane for<del>of the</del> polarization components 800 component by 90° (Freudenthaler, 2016). In this case, the photodetectors change places with respect to the polarization components. Turning of polarization. Rotation by 90° can be carried out physically -by rotating the entire photodetector unit (Yoshida et al., 2010; Strawbridge, 2013), or by rotating the half-wave phase plate, which can be installed both in the transmitter channel (Spinhirne et al., 1982) - and the receiver (McGill et al., 2002; Reichardt et al., 2012). In the previous lidars of the LOSA series (Balin et al., 2009), a-mechanical rotation of the photodetector unit by 90° was used. The intensity 805 ratios in both channels were taken into account and - which made it possible to eliminate possible changes in the object brightness of the object during turning, the turn. When calibrating with this method, you can choose any stable aerosol-cloud formation as a scattering object, which is characterized by a noticeable (>10%) and constant depolarization ratio during the measurement period.

2000; Engelmann et al., 2016; Summa et al., 2013; Groß et al., 2015; Engelmann et al., 2016), and 355+532+1064 nm (Haarig et al., 2016; Burton et al., 2015; Haarig et al., 2017; Hu et al., 2019). The). Lidar LOSA-M3 lidar measures polarization components at wavelengths of 532 and 1064 nm. At the same time, the polarization components are separated distinguished directly behind the receiving telescope, before the radiation is separated according to by wavelength. This In this scheme has there are no distortions of the polarization state upon reflection from dichroic mirrors reflection;

815

To measure the depolarization ratio of the backscattered radiation, the relative sensitivity of detectors must be known. Thus, in In-our lidar we offer a new method for a of-continuous rotation of  $\lambda/2$  phase plate, temporarily installed in the receiver module instead of  $\lambda/4$  plate *PP3* in Fig. 2. For the light backscattering phase matrix (BSPM), we take a  $4 \times 4$  matrix **M** relating the Stokes vectors of radiation scattered in the direction toward the source  $\mathbf{S} = [I, Q, U, V]^T$  with the Stokes vector  $\mathbf{S}_0$  of radiation, incident on an ensemble of particles. Polarization components are expressed as  $I_{\parallel} = (I+Q)/2$ ,  $I_{\perp} = (I-Q)/2$ . Let us assume that laser radiation is linearly polarized (the initial vector is  $\mathbf{S}_0 = [1,1,0,0]^T$ ). Then the scattered Stokes vector is  $\mathbf{S} = \mathbf{MS}_0^L = [m_{11} + m_{12}, m_{12} + m_{22}, -m_{13} - m_{23}, m_{14} + m_{24}]^T$ . If  $\begin{pmatrix} 1 & 0 & 0 & 0 \\ 0 & 0 & 0 \end{pmatrix}$

we install set the  $\lambda/2$  phase plate before the polarization prism with matrix  $\mathbf{L} = \begin{bmatrix} 1 & 0 & 0 & 0 \\ 0 & \cos 4\varphi & \sin 4\varphi & 0 \\ 0 & \sin 4\varphi & -\cos 4\varphi & 0 \\ 0 & 0 & 0 & 0 \end{bmatrix}$  before the

polarization prism in the receiver, where,  $\varphi$  is the rotation angle, the Stokes vector will be  $\mathbf{S} = \mathbf{LMS}_{0}^{L} = \begin{pmatrix} m_{11} + m_{12} \\ (m_{12} + m_{22})C_{4} - (m_{13} + m_{23})S_{4} \\ (m_{12} + m_{22})S_{4} + (m_{13} + m_{23})C_{4} \\ m_{14} + m_{24} \end{pmatrix}, \text{ where } C_{4} = \cos 4\varphi, \ S_{4} = \sin 4\varphi \text{ Polarization components } I_{\parallel}, I_{\perp}$

are is different:  $2I_{\perp}^{\parallel} = m_{11} + m_{12} \pm (m_{12} + m_{22})C_4 \mp (m_{13} + m_{23})S_4$  (the upper sign is for  $I_{\parallel}$ , the lower for  $I_{\perp}$ ). However, But the integral for a complete turn of the plate over the angle  $\varphi \div \int_{2\pi} I_{\parallel,\perp} d\varphi = \pi (m_{11} + m_{12})$  is the same for each polarization component, regardless of the values of matrix elements. The ratio of the measured values of the integral for two components  $K = \int I_{\perp} d\varphi / \int I_{\parallel} d\varphi$  gives us the value of the relative sensitivity of the polarization channels.

825

820

Calibration procedure is done made-separately for 532 and 1064 nm. The method works for any initial state of polarization and for any BSPM of aerosol layer. Unlike the  $\Delta 90$  method, there is no need to ensure extremely accurate setting of the plate rotation angle, but during the turn of the plate turning (about 20 seconds), the change in the scattering properties of the object should be minimal. Fig. 7 shows 6 depicts one of the calibration records made held on 07 May .05.2017, demonstrating the errors of the applied method. The upper Upper part (7a6a) shows the record of a the signal (1064 nm) integrated by the plate rotation angle of the plate, components  $I_{\parallel}$  (green) and  $I_{\perp}$  (redblue). Integration was carried out during four for 4-revolutions of the plate. Fig. 7b shows 6b depicts the value of calibration constant *K*, calculated in the 1800-

830  $\frac{8000 \text{ m}}{\text{ height range of } 1800-8000 \text{ m}}$ . In this case, the mean value K=1.91. Similar calibration procedures were carried out before each measurement in April and June 2018. All values K did not deviate by more than  $\pm 0.05$ .

**4 Observations of icecrystalline clouds during lidar zenith scanning**

In this paper, we present the results of observations of the ice<del>crystalline</del> cloud cover under the lidar zenith scanning. Measurements were made in Tomsk (56°28'N, 85°E) in April-June 2018. Scanning was carried out at the a-rate of 0.5 degrees per second, which corresponds to three arc minutes a-shift of sensing in the direction of sounding by 3 minutes between laser pulses. The angle is measured from the vertical position of the radiation beam, which was set established with an error of  $\pm$  5 arc minutes. Moreover, in some cases, the scanning was carried out with the lidar axis passing through the zenith (ranging from -1 ° to +-4-°), which made it possible to control the accuracy of the lidar vertical setting. The measurement data array is not large and is mainly used here for testing the lidar performance.

**840 4.1 Zenith scanning at 1064 nm wavelength**

Below we present the data for polarized components at 1064 nm related to radiation with initial circular polarization. The depolarization of linearly polarized radiation is generally less than that of circularly polarized radiation. For randomly distributed particles, the linear depolarization is two times less than for circular (Mishchenko and Hovenier, 1995; Gimmestad, 2008). For oriented particles the difference is smaller, since the BSPM element Figure 7 shows the sounding data on April 6, 2018, 10:25 local time. 7a depicts the component  $I_{\parallel}$  is not equal to zero (Balin et al., 2011). Low value of depolarization results in that the cross-polarized component -(signal intensity, mV), 7b depicts the depolarization ratio  $\delta^{Cire} = I_{\perp}/I_{\parallel}$  very often does not stand out against the background of the photodetector noise in the specular reflection. Therefore, the dependences of polarized components for 1064 nm are presented for initial circular polarization of laser beam. Dependences -(%), measured for the circle initial polarization of the laser beam, 7e - the zenith scanning angle (minutes).

Date of weather sounding (7d) are taken from the site of the University of Wyoming (http://weather.uwyo.edu).

850

The duration of the recording is 300 seconds. In the interval from 120 to 250 seconds, scanning was carried out in the range from 0 to 2 degrees. The behavior of the two cloud layers is clearly different. The characteristics of the upper layer (8-10 km) do not change when scanning, which indicates the chaotic orientation of the particles in the cloud. In the lower layer (6.5-7 km), a pronounced modulation is observed, characteristic for a predominantly horizontal orientation of the particles. The maximum intensity of the signal with the vertical sounding direction corresponds to a minimum of the depolarization ratio. The extremely low value  $\delta^{Circ}$  for linear and circular polarizations for scan angles 0-4° do not differ within the error limits.

Figure 8 shows the lidar data observed on 6 April 2018. The recording starts at 10:25 local time (UTC+7), time from the start is shown on X-axis. Fig. 8a gives the component -in the zenith direction  $\delta_{Zen}^{Circ} \approx 4-5\%$  (signal intensity, mV); 8b - the depolarization ratio  $\delta^{Circ} = I_{cros}^{Circ} / I_{co}^{Circ}$ , %; 8d - the zenith scanning angle (arc minutes), time axis is aligned with Fig. 8a 860 and 8b. Data of weather sounding (8c) are taken from the site of the University of Wyoming (http://weather.uwyo.edu). Radiozonde sounding was carried out at Novosibirsk station, about 250 km from Tomsk to SW, two records were made at 07:00 and 19:00 LST. Of course, due to the distant location of the station and the rare launches of probes (once every twelve hours) these data do not describe the fine structure of the ice cloud.

865

Duration of recording is 300 seconds. In the interval from 120 to 250 seconds, scanning was carried out in the range from 0 to 2 degrees. A high-level cloud consists of two layers. The thin bottom layer (6600-7000 m) has a temperature minus 26-31°C, the top layer (7800-9800 m) temperature is 37-52°C below zero. The behavior of these layers is significantly different. Characteristics of the top layer do not change when scanning, which indicates the chaotic orientation of particles in the cloud. In the bottom layer a pronounced modulation of signal intensity and polarization is observed, characteristic for a 870 predominantly horizontal orientation of particles. The maximum intensity of signal with the vertical sensing direction corresponds to the minimum of depolarization ratio. An extremely low value  $\delta^{Circ}$  in the zenith direction  $\delta^{Circ}_{Zen} \approx 4-5\%$ corresponds to the specular reflection.

875

The other situation is observed on 2 June 2018, 09:55 LST (Fig. 9). Duration of this recording is 550 seconds. The maximum inclination was 4°, the beam passed through the vertical by 1° while scanning. In Fig. 8a we saw clearly expressed single line of maximum signal, because the vertical position was the edge position when scanning. When we scan our lidar from 4° to -1° and back, vertical positions (0°) are close to each other. So in Fig. 9a two lines of maximum signals are close to each other and look like a double line.

**-corresponds to the specular reflection.**

Another situation is observed on June 02, 09:55 LST (Fig. 8). The duration of this recording is 550 seconds. The 880 maximum slope was 4°, while scanning the beam passed through the vertical by 1° (this leads to the appearance of double lines of maximum intensity in Fig.8a). A cirrus cloud with a complex structure occupies the layer 8 - 11.7 km. A cirrus cloud with a complex structure extends in a layer of 8 - 11.7 km. The temperature in the cloud varies from -34°C to -60°C. Pronounced pronounced modulation of the signal intensity and depolarization ratio areis observed throughout the entire height of the cloud. As in the previous case, the maximum intensity of the signal corresponds to the a-minimum of depolarization ratio. However, the minimum values  $\delta_{Zen}^{Circ}$  differare significantly different in various different parts of the 885 cloud. In the lower part of the cloud, the values  $\delta_{Zen}^{Circ} \approx 0.6$ , which indicates the predominance of chaotically oriented particles and a small proportion of horizontally oriented particles. In the upper part (11.2-11.7 km)  $\delta_{Zen}^{Circ} < 5\%$  . This is characteristic of mirror reflection and indicates a pronounced horizontal orientation of particles in this part of the cloud.

Values  $\delta^{Circ}$  The values outside the vertical are close to 100% throughout the entire cloud thickness of the cloud. This suggests that particles in the cloud differ only in the degree of their horizontal orientation.

**4.2 Dependence of the signal intensity on the lidar tilt angle**

The ice clouds of the upper layers never constitute a formation uniform in height and constant in time. Pronounced In the structure of crystalline clouds, pronounced layers with the a-thickness of the order of hundreds of meters are regularly observed in the structure of ice clouds. They differ in the state of depolarization and signal intensity from the higher and lower regions and are supposedly homogeneous in composition and degree of particle orientation. However, the signal intensity and the value of the depolarization ratio do not remain constant, but vary with time rather quickly change with time. The height of the layers also varies changes. Weak values of the backscatter signals lead to the need of for averaging the signals over the height of the selected layer and over time about of the order of 3-5 minutes. As a result, to measure the dependence of the characteristics of the ceho signal on the angle of inclination, it is necessary to pre-first select sections of the cloud, characterized by an approximately constant value of the intensity and depolarization ratio, and lasting for several scan cycles to measure the dependence of echo signal characteristics on the tilt angle. An example of such a procedure for selecting the studied cloud sections to be studied is shown in Fig. 109.

In the given 5-minute recording, 6 sections were selected (marked with rectangles) with thea duration from of three to seven scan cycles. The rest of remaining cloud portions on this record were not analyzed on this record. In total, about 20 records were selected during observations on 2 the observation on June 2, 2018 and October 1, 2018, in which have there is a pronounced dependence on the tilt angle, of the lidar and there is no signal overflow when the lidar is oriented to zenith. The following are typical examples of the dependences of the intensity of the polarization components on the angle of inclination.

Figure 1110 indicates the typical dependences dependencies of signal intensities of parallel  $I_{\parallel}^{circ}$  (circular initial polarization) and cross-polarized  $I_{cros}^{circ}$   $I_{\perp}^{circ}$  components on the tilt angle  $\alpha$ . The record shown in Fig. 11a10a was registered at 12:00 LST with averaging of the characteristics over the layer from 11470m to 11600m. The record shown in Fig. 11b10b was made at 10:00 LST with averaging of the characteristics over the layer from 10980 to 11270m. The signal was accumulated during 10 minutes. These -obtained in-measurements and all data, obtained in June and October 2018, are-are well described satisfactorily by the following exponential dependencedistribution

$$-I(\alpha) = I_0 + A \exp(-|\alpha - \alpha_0|/w)$$
(1)

(red line in Fig. 1110), where I is the signal intensity,  $I_0$  is the offset of dependence determined by a signal intensity without the specular component ( $\alpha > 4^\circ$ ), A is the a-constant, depending on the contribution of specular reflection in the-total intensity,  $\alpha$  is the tiltan inclination angle, w determines the width of the distribution, and  $\alpha_0$  indicates the error of the lidar targeting to the vertical. For Fig. 11a10a w=42 arcangle minutes, for Fig. 11b10b w=82'. The Gaussian function used in Noel

920 and Sassen, 2005 Ref. 11-(blue line in Fig. 1 is noticeably wider and 10) poorly describes the observed distribution, perhaps because it does not take into consideration the intensity. The main difference is determined by the underestimation of the sharp peak at  $\alpha$ =0°.

Cross-polarized signals The values  $I_{\perp}$  have random variations from pulse to pulse comparable with the average value. In most cases the values of -in most cases shows a weak decline with the angle, but its change are smaller than the measurement errors. We can conclude that  $I_{\perp}$  show a weak decline of intensity with the angle, but these variations do not exceed instrumental errors (about 1% for depolarization ratio). -is practically independent on the inclination angle.

Figure 11a shows the variations even less the 0.5%. Moreover, a slightly noticeable maximum at  $\alpha$ =0° looks like signal penetration from parallel to perpendicular channel. Therefore we think, that linear trend is not statistically reliable. We can conclude that  $I_{cros}^{circ}$  is practically independent on the tilt angle.

Figure 12 gives 11 depicts some selected dependences dependencies of intensity angle distributions for component- $I_{\parallel}$ . We did not find any correlation of the distribution parameters  $(A, I_0, w)$  with the height of the selected area inside the cloud. So the selection of cloud portions presented in Fig. 12 is rather subjective. The only requirement is: these portions must have a pronounced dependence on the tilt angle, and there is no signal overflow when the lidar is oriented to zenith. The dependences are normalized to the value  $I_0$  obtained by fitting according to the Eq. (1). Squares indicate the tilt angles, corresponding to the distribution width w. For all measurements the value w is within 40-160 arc150 angle minutes. The shift  $a_0$  of the curve maximum from 0° is less than 2 minutes, which indicates a good lidar orientation of the lidar to the zenith. Since the signal intensity offset  $I_0$  for  $a>>4^\circ$  is determined by all-particles with any orientation (both random and horizontal), the ratio  $I_{\parallel}(0^\circ)/I_0$  may reflect shows the contribution proportion of mirrorhorizontally oriented particles to minutes. The ratio  $I_{core}^{C}(4^\circ)/I_0$  is less then 1.1. The right frame shows cases with  $I_{core}^{Circ}(4^\circ)/I_0 > 1.1$ , among these cases there are distributions with a large (up to 159 arc min) width given section of the cloud.

945

925

Previous data related to radiation with initially circular polarization. The degree of depolarization of linearly polarized radiation is generally less than that of circularly polarized radiation. For chaotically oriented particles, the depolarization is less than half (Mishchenko and Hovenier, 1995; Gimmestad, 2008), for oriented particles the difference is smaller, since the BSPM element  $m_{12}$ -is not equal to zero (Balin et al., 2011). The low value of depolarization leads to the fact that in specular reflection the cross polarized component  $I_{\perp}^{Lin}$  very often does not stand out against the background of the noise of the photodetector. Therefore, the dependences  $I_{\perp}(\alpha)$  were investigated for circular polarization. The dependences  $I_{\parallel}(\alpha)$  for linear and circular polarizations for scan angles 0–4° do not differ within the error limits.

**4.3 Angular distributions for green and infrared wavelengths**

- The obtainedresulting dependence  $I_{\parallel}(\alpha)$  in general terms should correspond to the distribution of specularly reflecting particles along the angles of deviation from the horizontal plane (flutter). However, due to the phenomenon of diffraction of backscattered radiation diffraction on the particle's contour (Borovoi et al., 2008), the distribution is wider than the distribution of particles along the tilt angles of inclination. In addition, the ratio of backscattering coefficients (color ratio) for  $\alpha >> 4^{\circ}$  (and for randomly oriented particles in any direction) is not equal to unity (Tao(Borovoi et al., 2008<del>2014</del>;
- 955 Vaughan et al., 2010; BorovoiTao et al., 20142008). Therefore, the distribution  $I_{\parallel}(\alpha)$  may depend on the radiation wavelength.

As mentioned in the Sect. 2 the In our lidar, measurements for the a-wavelength of  $\lambda = 532$  nm were are-carried out only for linear polarization of radiation, as we used since the quarter-wave plates in the transceiver for the wavelength are made for a wavelength of  $\lambda = 1064$  nm. However, in the position where the axis of the rotating phase plate coincides with the plane of polarization of the emitter, the radiation remains linearly polarized for any wavelength. These pulses are used for measurements at a wavelength of 532 nm.

Two cases with the maximum angular differences are shown in Fig. 13.12. The relative sensitivity of photodetectors at 532 and 1064 nm was not calibrated. Therefore the intensities of polarization components for 532 nm (both  $I_{co}^{Lin}$  and  $I_{cros}^{Lin}$ ) were normalized so that the intensity maximum of  $I_{co}^{Lin}$  (0°) in the vertical position coincided with  $I_{co}^{Lin}$  (0°) for 1064 nm.5 therefore, the amplitudes of the signals are shown to be the same. In the left figureframe the angle distributions are the same, in the right figureframe the distribution for 532 nm is much wider ( $I_{532}(4^\circ)/I_{1064}(4^\circ) = 1.5$ ). In all other cases we have intermediate states: the distribution width *w* for 532 nm is equal or bigger thanmore then for 1064 nm. The reason for suchthis behavior of dependences is not yet clear for the authors of the article.

**5** Conclusions**

- 970 Scanning polarizing lidar LOSA-M3 is developed and operated in the IAO SB RAS in the Laboratory of Optical Sensing of the Atmosphere (LOSA) at the IAO SB RAS.). The main purpose of this lidar is to the study of the optical characteristics of the mid- and high-level iccerystal clouds of the upper and middle layers at two wavelengths - 532 and 1064 nm. Lidar allows a smoothyou to smoothly change of the tilt angle of inclination from the vertical with simultaneous conical (azimuthal) scanning. Such a-measurement scheme makes it possible to study in detail-the preferential orientation of icc crystals in the
- 975 clouds in detail. The lidar simultaneously measures the polarization characteristics of signals for linear and circular initial polarizations, which allows obtaining to obtain additional information about the anisotropy of scattering particles. Since the circular polarization signal does not depend on the rotation of the lidar relative to the direction of particle orientation, this polarization can be used as a reference for investigation, including exploring the azimuthal orientation of the particles. At the

same time, the a-lidar, like all of the LOSA series-aerosol-Raman lidars, has a Raman scattering channel (607 nm band, for night observations) and a system for combining near and far rangeszones, which allows it to be used for observingto observe aerosol fields in the troposphere distances of 5050 m - 1515 km.

During 2018, several series of measurements of the structure of the iccerystalline clouds of the upper layers in the zenith scan mode were carried out. The results show that the contribution of horizontally oriented particles, giving a specular reflection, can vary significantly in different parts of the cloud. The dependence of the signal intensity on the tilt angle of inclination reflects the distribution of the particles deviation of particles relative to the horizontal plane, and is well described by thean exponential dependence. Cross-polarized component  $I_{\perp}$  in most cases shows a weak decline with the angle, but its variationschange are smaller than the measurement errors. We can conclude that  $I_{\perp}$  is practically independent on the tiltinclination angle. The angle distribution of for-the radiation for 532 nm in all experiments is equal or wider than then for 1064 nm. The reason for such this behavior of dependences dependencies is not yet clear for the authors of the article.

[revised manuscript text omitted]